# Discovering non-additive heritability using additive GWAS summary statistics

**Samuel Pattillo Smith**[1,2,3,4†], **Gregory Darnell**[1,5†], **Dana Udwin**[6], **Julian Stamp**[1], **Arbel Harpak**[3,4], **Sohini Ramachandran**[1,2,7‡], **Lorin Crawford**[1,6,8*‡]

[1]Center for Computational Molecular Biology, Brown University, Providence, United States; [2]Department of Ecology and Evolutionary Biology, Brown University, Providence, United States; [3]Department of Integrative Biology, The University of Texas at Austin, Austin, United States; [4]Department of Population Health, The University of Texas at Austin, Austin, United States; [5]Institute for Computational and Experimental Research in Mathematics, Brown University, Providence, United States; [6]Department of Biostatistics, Brown University, Providence, United States; [7]Data Science Institute, Brown University, Providence, United States; [8]Microsoft, Cambridge, United States

**Abstract** LD score regression (LDSC) is a method to estimate narrow-sense heritability from genome-wide association study (GWAS) summary statistics alone, making it a fast and popular approach. In this work, we present interaction-LD score (i-LDSC) regression: an extension of the original LDSC framework that accounts for interactions between genetic variants. By studying a wide range of generative models in simulations, and by re-analyzing 25 well-studied quantitative phenotypes from 349,468 individuals in the UK Biobank and up to 159,095 individuals in BioBank Japan, we show that the inclusion of a *cis*-interaction score (i.e. interactions between a focal variant and proximal variants) recovers genetic variance that is not captured by LDSC. For each of the 25 traits analyzed in the UK Biobank and BioBank Japan, i-LDSC detects additional variation contributed by genetic interactions. The i-LDSC software and its application to these biobanks represent a step towards resolving further genetic contributions of sources of non-additive genetic effects to complex trait variation.

**\*For correspondence:** lcrawford@microsoft.com

[†]These authors contributed equally to this work
[‡]These authors also contributed equally to this work

**Competing interest:** The authors declare that no competing interests exist.

## Editor's evaluation

This study provides a valuable investigation into whether phenotypic variance due to interactions between genetic variants can be measured using genome-wide association summary statistics. The authors present a convincing method, i-LDSC, that uses statistics on the correlations between genotypes at different loci (linkage disequilibrium) to estimate the phenotypic variance explained by both additive genetic effects and pairwise interactions.

## Introduction

Heritability is defined as the proportion of phenotypic trait variation that can be explained by genetic effects (*Bulik-Sullivan et al., 2015b, Bulik-Sullivan et al., 2015a, Shi et al., 2016*). Until recently, studies of heritability in humans have been reliant on typically small sized family studies with known relatedness structures among individuals (*Zaitlen et al., 2013*; *Polderman et al., 2015*). Due to advances in genomic sequencing and the steady development of statistical tools, it is now possible to obtain reliable heritability estimates from biobank-scale data sets of unrelated individuals (*Bulik-Sullivan et al., 2015b*; *Shi et al., 2016*; *Hou et al., 2019*; *Pazokitoroudi et al., 2020*). Computational

and privacy considerations with genome-wide association studies (GWAS) in these larger cohorts have motivated a recent trend to estimate heritability using summary statistics (i.e. estimated effect sizes and their corresponding standard errors). In the GWAS framework, additive effect sizes and standard errors for individual single nucleotide polymorphisms (SNPs) are estimated by regressing phenotype measurements onto the allele counts of each SNP independently. Through the application of this approach over the last two decades, it has become clear that many traits have a complex and polygenic basis—that is, hundreds to thousands of individual genetic loci across the genome often contribute to the genetic basis of variation in a single trait (*Yengo et al., 2018*).

Many statistical methods have been developed to improve the estimation of heritability from GWAS summary statistics (*Bulik-Sullivan et al., 2015b*, *Shi et al., 2016*, *Speed and Balding, 2019*, *Song et al., 2022*). The most widely used of these approaches is linkage disequilibrium (LD) score regression and the corresponding LDSC software (*Bulik-Sullivan et al., 2015b*), which corrects for inflation in GWAS summary statistics by modeling the relationship between the variance of SNP-level effect sizes and the sum of correlation coefficients between focal SNPs and their genomic neighbors (i.e. the LD score of each variant). The formulation of the LDSC framework relies on the fact that the expected relationship between chi-square test statistics (i.e. the squared magnitude of GWAS allelic effect estimates) and LD scores holds when complex traits are generated under the infinitesimal (or polygenic) model which assumes: (*i*) all causal variants have the same expected contribution to phenotypic variation and (*ii*) causal variants are uniformly distributed along the genome. Initial simulations in Bulik-Sullivan et al. showed that violations of these assumptions can be tolerated to a point, but begin to affect the estimation of narrow-sense heritability once a certain proportion of variants have nonzero effects. Importantly, the estimand of the LDSC model is the proportion of phenotypic variance attributable to additive effects of genotyped SNPs. The main motivation behind the LDSC model is that, for polygenic traits, many marker SNPs tag nonzero effects. This may simply arise because some of these SNPS are in LD with causal variants (*Bulik-Sullivan et al., 2015b*) or because their statistical association is the product of a confounding factor such as population stratification.

As of late, there have been many efforts to build upon and improve the LDSC framework. For example, recent work has shown that it is possible to estimate the proportion of phenotypic variation explained by dominance effects (*Palmer et al., 2023*) and local ancestry (*Chan et al., 2023*) using extensions of the LDSC model. One limitation of LDSC is that, in practice, it only uses the diagonal elements of the squared LD matrix in its formulation which, while computationally efficient, does not account for information about trait architecture that is captured by the off-diagonal elements. This tradeoff helps LDSC to scale genome-wide, but it has also been shown to lead to heritability estimates with large standard error (*Ning et al., 2020*, *Zhang et al., 2021*, *Song et al., 2022*). Recently, newer approaches have attempted to reformulate the LDSC model by using the eigenvalues of the LD matrix to leverage more of the information present in the correlation structure between SNPs (*Shi et al., 2016*, *Song et al., 2022*).

In this paper, we show that the LDSC framework can be extended to estimate greater proportions of genetic variance in complex traits (i.e. beyond the variance that is attributable to additive effects) when a subset of causal variants is involved in a gene-by-gene (G×G) interaction. Indeed, recent association mapping studies have shown that G×G interactions can drive heterogeneity of causal variant effect sizes (*Patel et al., 2022*). Importantly, non-additive genetic effects have been proposed as one of the main factors that explains 'missing' heritability—the proportion of heritability not explained by the additive effects of variants (*Eichler et al., 2010*).

The key insight we highlight in this manuscript is that SNP-level GWAS summary statistics can provide evidence of non-additive genetic effects contributing to trait architecture if there is a nonzero correlation between individual-level genotypes and their statistical interactions. We present the 'interaction-LD score' regression model or i-LDSC: an extension of the LDSC framework which recovers 'missing' heritability by leveraging this 'tagged' relationship between linear and nonlinear genetic effects. To validate the performance of i-LDSC in simulation studies, we focus on synthetic trait architectures that have been generated with contributions stemming from second-order and *cis*-acting statistical SNP-by-SNP interaction effects; however, note that the general concept underlying i-LDSC can easily be extended to other sources of non-additive genetic effects (e.g. gene-by-environment interactions). The main difference between i-LDSC and LDSC is that the i-LDSC model includes an additional set of '*cis*-interaction' LD scores in its regression model. These scores measure the amount

of phenoytpic variation contributed by genetic interactions that can be explained by additive effects. In practice, these additional scores are efficient to compute and require nothing more than access to a representative pairwise LD map, same as the input required for LD score regression.

Through extensive simulations, we show that i-LDSC recovers substantial non-additive heritability that is not captured by LDSC when genetic interactions are indeed present in the generative model for a given complex trait. More importantly, i-LDSC has a calibrated type I error rate and does not overestimate contributions of genetic interactions to trait variation in simulated data when only additive effects are present. While analyzing 25 complex traits in the UK Biobank and BioBank Japan, we illustrate that pairwise interactions are a source of 'missing' heritability captured by additive GWAS summary statistics—suggesting that phenotypic variation due to non-additive genetic effects is more pervasive in human phenotypes than previously reported. Specifically, we find evidence of tagged genetic interaction effects contributing to heritability estimates in all of the 25 traits in the UK Biobank, and 23 of the 25 traits we analyzed in the BioBank Japan. We believe that i-LDSC, with our development of a new *cis*-interaction score, represents a significant step towards resolving the true contribution of genetic interactions.

## Results

### Overview of the interaction-LD score regression model

Interaction-LD score regression (i-LDSC) is a statistical framework for estimating heritability (i.e. the proportion of trait variance attributable to genetic variance). Here, we will give an overview of the i-LDSC method and its corresponding software, as well as detail how its underlying model differs from that of LDSC (***Bulik-Sullivan et al., 2015b***). We will assume that we are analyzing a GWAS dats set $\mathcal{D} = \{\mathbf{X}, \mathbf{y}\}$ where $\mathbf{X}$ is an $N \times J$ matrix of genotypes with $J$ denoting the number of SNPs (each of which is encoded as $\{0, 1, 2\}$ copies of a reference allele at each locus $j$) and $\mathbf{y}$ is an $N$-dimensional vector of measurements of a quantitative trait. The i-LDSC framework only requires summary statistics of individual-level data: namely, marginal effect size estimates for each SNP $\widehat{\beta}$ and a sample LD matrix $\mathbf{R}$ (which can be provided via reference panel data).

We begin by considering the following generative linear model for complex traits

$$\mathbf{y} = b_0 + \mathbf{X}\beta + \mathbf{W}\theta + \varepsilon, \qquad \varepsilon \sim \mathcal{N}(\mathbf{0}, (1 - H^2)\mathbf{I}), \tag{1}$$

where $b_0$ is an intercept term; $\beta = (\beta_1, \ldots, \beta_J)$ is a $J$-dimensional vector containing the true additive effect sizes for an additional copy of the reference allele at each locus on $\mathbf{y}$; $\mathbf{W}$ is an $N \times M$ matrix of (pairwise) *cis*-acting SNP-by-SNP statistical interactions between some subset of causal SNPs, where columns of this matrix are assumed to be the Hadamard (element-wise) product between genotypic vectors of the form $\mathbf{x}_j \circ \mathbf{x}_k$ for the $j$-th and $k$-th variants; $\theta = (\theta_1, \ldots, \theta_M)$ is an $M$-dimensional vector containing the interaction effect sizes; $\varepsilon$ is a normally distributed error term with mean zero and variance scaled according to the proportion of phenotypic variation not explained by genetic effects (***Bulik-Sullivan et al., 2015b***), which we will refer to as the broad-sense heritability of the trait denoted by $H^2$; and $\mathbf{I}$ denotes an $N \times N$ identity matrix. For convenience, we will assume that the genotype matrix (column-wise) and the trait of interest have been mean-centered and standardized (***Strandén and Christensen, 2011***; ***de Los Campos et al., 2013***; ***Zhou et al., 2013***). Lastly, we will let the intercept term $b_0$ be a fixed parameter and we will assume that the effect sizes are each normally distributed with variances proportional to their individual contributions to trait heritability (***Yang et al., 2010***; ***Wu et al., 2011***; ***Zhou et al., 2013***; ***Crawford et al., 2017***)

$$\beta_j \sim \mathcal{N}(0, \varphi_\beta^2/J), \qquad \theta_m \sim \mathcal{N}(0, \varphi_\theta^2/M). \tag{2}$$

Effectively, we say that $\mathbb{V}[\mathbf{X}\beta] = \varphi_\beta^2$ is the proportion of phenotypic variation contributed by additive SNP effects under the generative model, while $\mathbb{V}[\mathbf{W}\theta] = \varphi_\theta^2$ makes up the proportion of phenotypic variation contributed by genetic interactions. While the appropriateness of treating genetic effects as random variables in analytical derivations has been questioned (***de Los Campos et al., 2015***), later, we will justify the theory presented here with simulation results showing that i-LDSC accurately recovers non-additive genetic variance in ***Equation 1*** under a broad range of conditions.

There are two key takeaways from the generative model specified above. First, *Equation 2* implies that the additive and non-additive components in *Equation 1* are orthogonal to each other. In other words, $\mathbb{E}[\beta^{\mathsf{T}}\mathbf{X}^{\mathsf{T}}\mathbf{W}\theta] = \mathbb{E}[\beta^{\mathsf{T}}]\mathbf{X}^{\mathsf{T}}\mathbf{W}\mathbb{E}[\theta] = 0$. This is important because it means that there is a unique partitioning of genetic variance when studying a trait of interest. The second key takeaway is that the genotype matrix $\mathbf{X}$ and the matrix of genetic interactions $\mathbf{W}$ themselves are correlated despite being linearly independent (see Materials and methods). This property stems from the fact that the pairwise interaction between two SNPs is encoded as the Hadamard product of two genotypic vectors in the form $\mathbf{w}_m = \mathbf{x}_j \circ \mathbf{x}_k$ (which is a nonlinear function of the genotypes).

A central objective in GWAS studies is to infer how much phenotypic variation can be explained by genetic effects. To achieve that objective, a key consideration involves incorporating the possibility of non-additive sources of genetic variation to be explained by additive effect size estimates obtained from GWAS analyses (*Hill et al., 2008*). If we assume that the genotype and interaction matrices are correlated, then $\mathbf{X}$ and $\mathbf{W}$ are not completely orthogonal (i.e. such that $\mathbf{X}^{\mathsf{T}}\mathbf{W} \neq 0$) and the following relationship between the moment matrix $\mathbf{X}^{\mathsf{T}}\mathbf{y}$, the observed marginal GWAS summary statistics $\widehat{\beta}$, and the true coefficient values $\beta$ from the generative model in *Equation 1* holds in expectation (see Materials and methods)

$$\mathbb{E}[\mathbf{X}^{\mathsf{T}}\mathbf{y}] = (\mathbf{X}^{\mathsf{T}}\mathbf{X})\beta + (\mathbf{X}^{\mathsf{T}}\mathbf{W})\theta \qquad \overset{\approx}{\Longleftrightarrow} \qquad \mathbb{E}[\widehat{\beta}] = \mathbf{R}\beta + \mathbf{V}\theta \qquad (3)$$

where $\mathbf{R}$ is a sample estimate of the LD matrix, and $\mathbf{V}$ represents a sample estimate of the correlation between the individual-level genotypes $\mathbf{X}$ and the span of genetic interactions between causal SNPs in $\mathbf{W}$. Intuitively, the term $\mathbf{V}\theta$ can be interpreted as the subset of pairwise interaction effects that are tagged by the additive effect estimates from the GWAS study. Note that, when (*i*) non-additive genetic effects do not contribute to the overall architecture of a trait (i.e. such that $\theta = \mathbf{0}$) or (*ii*) the genotype and interaction matrices $\mathbf{X}$ and $\mathbf{W}$ are uncorrelated, the equation above simplifies to a relationship between LD and summary statistics that is assumed in many GWAS studies and methods (*Hormozdiari et al., 2014*; *Nakka et al., 2016*; *Zhu and Stephens, 2017*; *Zhang et al., 2018*; *Zhu and Stephens, 2018*; *Cheng et al., 2020*; *Demetci et al., 2021*).

The goal of i-LDSC is to increase estimates of genetic variance by accounting for sources of non-additive genetic effects that can be explained by additive GWAS summary statistics. To do this, we extend the LD score regression framework and the corresponding LDSC software (*Bulik-Sullivan et al., 2015b*). Here, according to *Equation 3*, we note that $\widehat{\beta} \sim \mathcal{N}(\mathbf{R}\beta + \mathbf{V}\theta, \lambda\mathbf{R})$ where $\lambda$ is a scale variance term due to uncontrolled confounding effects (*Guan and Stephens, 2011*; *Song et al., 2022*). Next, we condition on $\Theta = (\beta, \theta)$ and take the expectation of chi-square statistics $\chi^2 = N\widehat{\beta}\widehat{\beta}^{\mathsf{T}}$ to yield

$$
\begin{aligned}
\mathbb{E}[\hat{\beta}\hat{\beta}^{\mathsf{T}}] = \mathbb{E}\left[\mathbb{E}\left[\hat{\beta}\hat{\beta}^{\mathsf{T}} \mid \Theta\right]\right] \quad &= \mathbb{E}\left[\mathbb{V}\left[\hat{\beta} \mid \Theta\right] + \mathbb{E}\left[\hat{\beta} \mid \Theta\right]\mathbb{E}\left[\hat{\beta} \mid \Theta\right]^{\mathsf{T}}\right] \\
&= \mathbb{E}\left[\lambda\mathbf{R} + (\mathbf{R}\beta + \mathbf{V}\theta)(\mathbf{R}\beta + \mathbf{V}\theta)^{\mathsf{T}}\right] \\
&= \mathbb{E}\left[\lambda\mathbf{R} + \mathbf{R}\beta\beta^{\mathsf{T}}\mathbf{R} + 2\mathbf{R}\beta\theta^{\mathsf{T}}\mathbf{V}^{\mathsf{T}} + \mathbf{V}\theta\theta^{\mathsf{T}}\mathbf{V}^{\mathsf{T}}\right] \\
&= \lambda\mathbf{R} + \left(\frac{\varphi_\beta^2}{J}\right)\mathbf{R}^2 + \left(\frac{\varphi_\theta^2}{M}\right)\mathbf{V}^2.
\end{aligned}
\qquad (4)
$$

We define $\ell_j = \sum_k r_{jk}^2$ as the LD score for the additive effect of the $j$-th variant (*Bulik-Sullivan et al., 2015b*), and $f_j = \sum_m v_{jm}^2$ represents the '*cis*-interaction' LD score which encodes the pairwise interaction between the $j$-th variant and all other variants within a genomic window that is a pre-specified number of SNPs wide (*Crawford et al., 2017*), respectively. By considering only the diagonal elements of LD matrix in the first term, similar to the original LDSC approach (*Bulik-Sullivan et al., 2015b*; *Song et al., 2022*), we get the following simplified regression model

$$\mathbb{E}[\chi^2] \propto \mathbf{1} + \boldsymbol{\ell}\tau + \mathbf{f}\vartheta \qquad (5)$$

where $\chi^2 = (\chi_1^2, \dots, \chi_J^2)$ is a $J$-dimensional vector of chi-square summary statistics, and $\boldsymbol{\ell} = (\ell_1, \dots, \ell_J)$ and $\mathbf{f} = (f_1, \dots, f_J)$ are $J$-dimensional vectors of additive and *cis*-interaction LD scores, respectively. Furthermore, we define the variance components $\tau = N\varphi_\beta^2/J$ and $\vartheta = N\varphi_\theta^2/M$ as the additive and non-additive regression coefficients of the model, and 1 is the intercept meant to model the bias factor due to uncontrolled confounding effects (e.g. cryptic relatedness structure). In practice, we efficiently

compute the *cis*-interaction LD scores by considering only a subset of interactions between each *j*-th focal SNP and SNPs within a *cis*-proximal window around the *j*-th SNP. In our validation studies and applications, we base the width of this window on the observation that LD decays outside of a window of 1 centimorgan (cM); therefore, SNPs outside the 1 cM window centered on the *j*-th SNP will not significantly contribute to its LD scores. Note that the width of this window can be relaxed in the i-LDSC software when appropriate. We fit the i-LDSC model using weighted least squares to estimate regression parameters and derive p-values for identifying traits that have significant statistical evidence of tagged *cis*-interaction effects by testing the null hypothesis $H_0 : \vartheta = 0$. Importantly, under the null model of a trait being generated by only additive effects, the i-LDSC model in *Equation 5* reduces to an infinitesimal model (*Fisher, 1999*) or, in the case some variants have no effect on the trait, a polygenic model.

Lastly, we want to note the empirical observation that the additive ($\ell$) and interaction ($f$) LD scores are lowly correlated. This is important because it indicates that the presence of *cis*-interaction LD scores in the model specified in *Equation 5* has little-to-no influence over the estimate for the additive coefficient $\tau$. Instead, the inclusion of $f$ creates a multivariate model that can identify the proportion of variance explained by both additive and non-additive effects in summary statistics. In other words, we can interpret $\widehat{\vartheta}$ as an estimate of the phenotypic variation explained by tagged *cis*-acting interaction effects. The concept of additive genetic effects partially explaining non-additive variation has also described in various studies from quantitative genetics (*Hill et al., 2008*; *Hivert et al., 2021*; *Mäki-Tanila and Hill, 2014*). Under Hardy-Weinberg equilibrium, it can be shown that the additive variance explained by $J$ SNPs takes on the following form (Materials and methods) (*Falconer and Mackay, 1983*)

$$\sigma_A^2 = \sum_{j=1}^{J} 2p_j(1 - p_j) \left[ \beta_j + 2 \sum_{k \neq j}^{J} p_k \theta_{jk} \right]^2. \tag{6}$$

The expression for the additive variance $\sigma_A^2$ in *Equation 6* is important because it represents the theoretical upper bound on the proportion of total phenotypic variance that can be recovered from GWAS summary statistics using the i-LDSC framework. As a result, we use the sum of coefficient estimates $\widehat{\tau} + \widehat{\vartheta} \leq \sigma_A^2$ to construct i-LDSC heritability estimates. A full derivation of the *cis*-interaction regression framework and details about its corresponding implementation in our software i-LDSC can be found in Materials and Methods.

## Detection of tagged pairwise interaction effects using i-LDSC in simulations

We illustrate the power of i-LDSC across different genetic trait architectures via extensive simulation studies (Materials and methods). We generate synthetic phenotypes using real genome-wide genotype data from individuals of self-identified European ancestry in the UK Biobank. To do so, we first assume that traits have a polygenic architecture where all SNPs have a nonzero additive effect. Next, we randomly select a set of causal *cis*-interaction variants and divide them into two interacting groups (Materials and methods). One may interpret the SNPs in group #1 as being the 'hubs' in an interaction map (*Crawford et al., 2017*), whereas SNPs in group #2 are selected to be variants within some kilobase (kb) window around each SNP in group #1. We assume a wide range of simulation scenarios by varying the following parameters:

- heritability: $H^2$ = 0.3 and 0.6;
- proportion of phenotypic variation that is generated by additive effects: $\rho$ = 0.5, 0.8, and 1;
- percentage of SNPs selected to be in group #1: 1%, 5%, and 10%;
- genomic window used to assign SNPs to group #2: ± 10 and ± 100 kb.

We also varied the correlation between SNP effect size and minor allele frequency (MAF; as discussed in *Schoech et al., 2019*). All results presented in this section are based on 100 different simulated phenotypes for each parameter combination.

*Figure 1* demonstrates that i-LDSC robustly detects significant tagged non-additive genetic variance, regardless of the total number of causal interactions genome-wide. Instead, the power of i-LDSC depends on the proportion of phenotypic variation that is generated by additive versus interaction

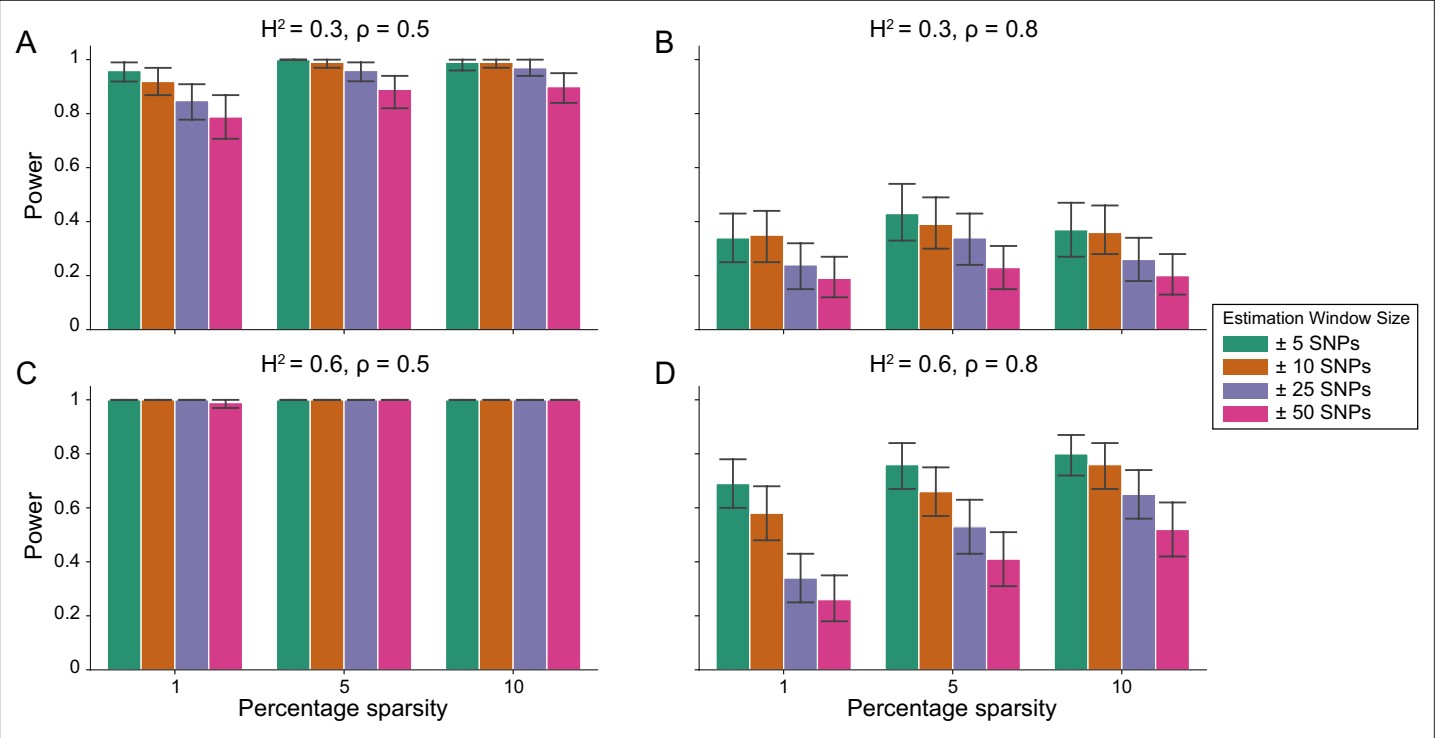

**Figure 1.** Power of the i-LDSC framework to detect tagged pairwise genetic interaction effects on simulated data. Synthetic trait architecture was simulated using real genotype data from individuals of self-identified European ancestry in the UK Biobank. All SNPs were considered to have at least an additive effect (i.e. creating a polygenic trait architecture). Next, we randomly select two groups of interacting variants and divide them into two groups. The group #1 SNPs are chosen to be 1%, 5%, and 10% of the total number of SNPs genome-wide (see the x-axis in each panel). These interact with the group #2 SNPs which are selected to be variants within a ± 10 kilobase (kb) window around each SNP in group #1. Coefficients for additive and interaction effects were simulated with no minor allele frequency dependency $\alpha = 0$ (see Materials and methods). Panels (**A**) and (**B**) are results with simulations using a heritability $H^2 = 0.3$, while panels (**C**) and (**D**) were generated with $H^2 = 0.6$. We also varied the proportion of heritability contributed by additive effects to (**A, C**) $\rho = 0.5$ and (**B, D**) $\rho = 0.8$, respectively. Here, we are blind to the parameter settings used in generative model and run i-LDSC while computing the *cis*-interaction LD scores using different estimating windows of ± 5 (green), ± 10 (orange), ± 25 (purple), and ± 50 (pink) SNPs. Results are based on 100 simulations per parameter combination and the horizontal bars represent standard errors. Generally, the performance of i-LDSC increases with larger heritability and lower proportions of additive variation. Note that LDSC is not shown here because it does not search for tagged interaction effects in summary statistics.

The online version of this article includes the following figure supplement(s) for figure 1:

**Figure supplement 1.** Power calculations for the i-LDSC framework to detect tagged pairwise genetic interaction effects on simulated data using a ± 10 kilobase (kb) window to generate *cis*-interactions around a focal SNP with a moderate minor allele frequency dependency $\alpha = -0.5$ for effect sizes.

**Figure supplement 2.** Power calculations for the i-LDSC framework to detect tagged pairwise genetic interaction effects on simulated data using a ± 10 kilobase (kb) window to generate *cis*-interactions around a focal SNP with a strong minor allele frequency dependency $\alpha = -1$ for effect sizes.

**Figure supplement 3.** Power calculations for the i-LDSC framework to detect tagged pairwise genetic interaction effects on simulated data using a ± 10 kilobase (kb) window to generate *cis*-interactions around a focal SNP with no minor allele frequency dependency $\alpha = 0$ for effect sizes.

**Figure supplement 4.** Power calculations for the i-LDSC framework to detect tagged pairwise genetic interaction effects on simulated data using a ± 100 kilobase (kb) window to generate *cis*-interactions around a focal SNP with a moderate minor allele frequency dependency $\alpha = -0.5$ for effect sizes.

**Figure supplement 5.** Power calculations for the i-LDSC framework to detect tagged pairwise genetic interaction effects on simulated data using a ± 100 kilobase (kb) window to generate *cis*-interactions around a focal SNP with a strong minor allele frequency dependency $\alpha = -1$ for effect sizes.

effects ($\rho$), and its power tends to scale with the window size used to compute the *cis*-interaction LD scores (see Materials and methods). i-LDSC shows a similar performance for detecting tagged *cis*-interaction effects when the effect sizes of causal SNPs depend on their minor allele frequency and when we varied the number of SNPs assigned to be in group #2 within 10 kb and 100 kb windows, respectively (*Figure 1—figure supplements 1–5*).

Importantly, i-LDSC does not falsely identify putative non-additive genetic effects in GWAS summary statistics when the synthetic phenotype was generated by only additive effects ($\rho = 1$). *Figure 2* illustrates the performance of i-LDSC under the null hypothesis $H_0 : \vartheta = 0$, with the type I

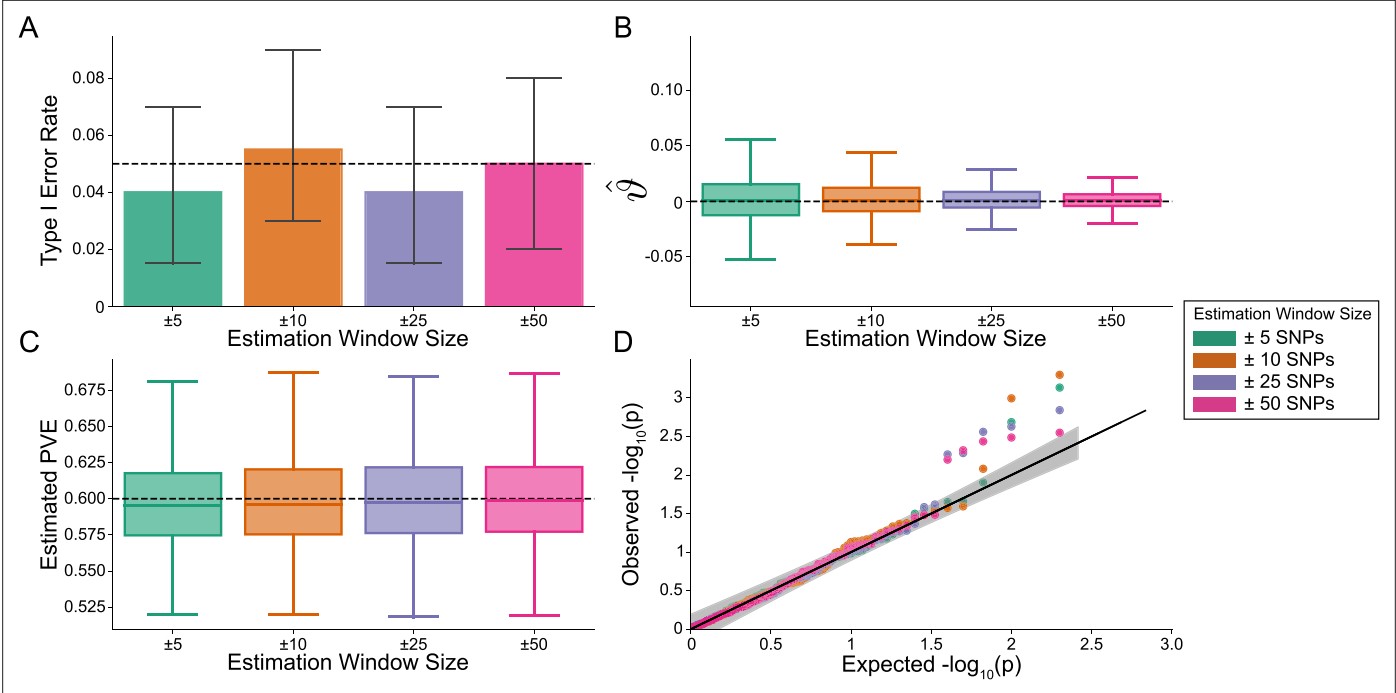

**Figure 2.** The i-LDSC framework is well-calibrated under the null hypothesis and does not identify evidence of tagged non-additive effects when polygenic traits are generated by only additive effects. In these simulations, synthetic trait architecture is made up of only additive genetic variation (i.e. $\rho = 1$). Coefficients for additive and interaction effects were simulated with no minor allele frequency dependency $\alpha = 0$ (see Materials and methods). Here, we are blind to the parameter settings used in generative model and run i-LDSC while computing the *cis*-interaction LD scores using different estimating windows of ± 5 (green), ± 10 (orange), ± 25 (purple), and ± 50 (pink) SNPs. (**A**) Mean type I error rate using the i-LDSC framework across an array of estimation window sizes for the *cis*-interaction LD scores. This is determined by assessing the p-value of the *cis*-interaction coefficient ($\vartheta$) in the i-LDSC regression model and checking whether $p < 0.05$. (**B**) Estimates of the *cis*-interaction coefficient ($\vartheta$). Since traits were simulated with only additive effects, these estimates should be centered around zero. (**C**) Estimates of the proportions of phenotypic variance explained (PVE) by genetic effects (i.e. estimated heritability) where the true additive variance is set to $H^2\rho = 0.6$. (**D**) QQ-plot of the p-values for the *cis*-interaction coefficient ($\vartheta$) in i-LDSC. Results are based on 100 simulations per parameter combination and the horizontal bars represent standard errors.

The online version of this article includes the following figure supplement(s) for figure 2:

**Figure supplement 1.** The i-LDSC framework is well-calibrated under the null hypothesis and does not identify evidence of tagged non-additive effects when polygenic traits are generated by only additive effects and a moderate minor allele frequency dependency $\alpha = -0.5$ for effect sizes.

**Figure supplement 2.** The i-LDSC framework is well-calibrated under the null hypothesis and does not identify evidence of tagged non-additive effects when polygenic traits are generated by only additive effects and a strong minor allele frequency dependency $\alpha = -1$ for effect sizes.

error rates for different estimation window sizes of the *cis*-interaction LD scores highlighted in panel A. Here, we also show that, when no genetic interaction effects are present, i-LDSC unbiasedly estimates the *cis*-interaction coefficient in the regression model to be $\hat{\vartheta} = 0$ (*Figure 2B*), robustly estimates the heritability (*Figure 2C*), and provides well-calibrated p-values when assessed over many traits (*Figure 2D*). This behavior is consistent across different MAF-dependent effect size distributions, and p-value calibration is not sensitive to misspecification of the estimation windows used to generate the *cis*-interaction LD scores (*Figure 2—figure supplements 1–2*).

One of the innovations that i-LDSC offers over the traditional LDSC framework is increased heritability estimates after the identification of non-additive genetic effects that are tagged by GWAS summary statistics. Here, we applied both methods to the same set of simulations in order to understand how LDSC behaves for traits generated with *cis*-interaction effects. *Figure 3* depicts boxplots of the heritability estimates for each approach and shows that, across an array of different synthetic phenotype architectures, LDSC captures less of phenotypic variance explained by all genetic effects. It is important to note that i-LDSC can yield upwardly biased heritability estimates when the *cis*-interaction scores are computed over genomic window sizes that are too small; however, these estimates become more accurate for larger window size choices (*Figure 3—figure supplement 1*). In contrast to LDSC, which aims to capture phenotypic variance attributable to the additive effects of

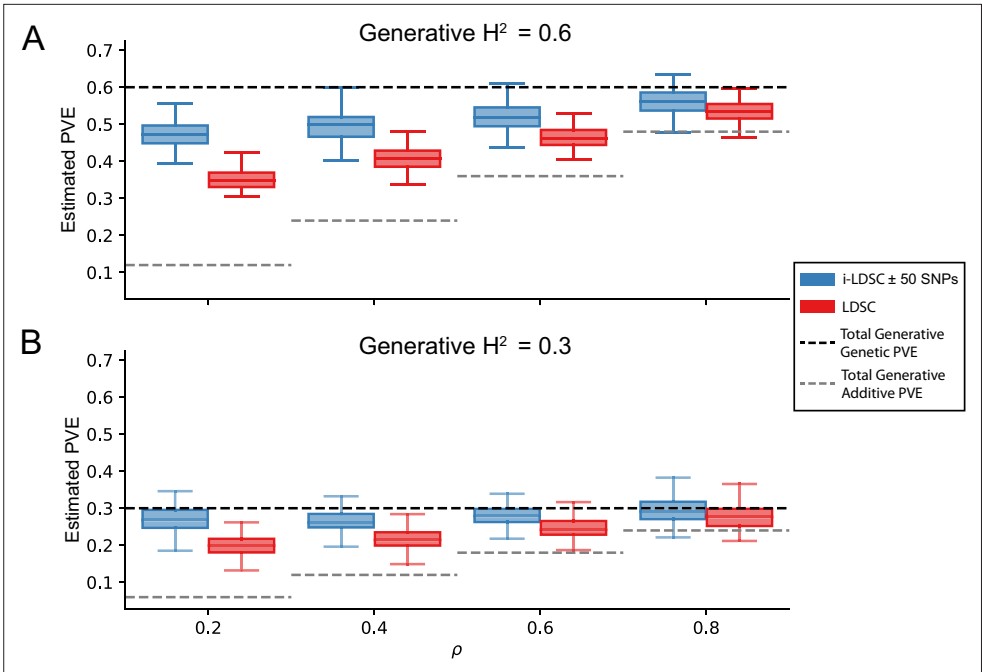

**Figure 3.** i-LDSC robustly and accurately estimates the proportions of phenotypic variance explained (PVE) by genetic effects (i.e. estimated heritability) in simulations in polygenic traits, compared to LDSC, due to our accounting for interaction effects tagged in additive GWAS summary statistics. Synthetic trait architecture was simulated using real genotype data from individuals of self-identified European ancestry in the UK Biobank (Materials and Methods). All SNPs were considered to have at least an additive effect (i.e. creating a polygenic trait architecture). Next, we randomly select two groups of interacting variants and divide them into two groups. The group #1 SNPs are chosen to be 10% of the total number of SNPs genome-wide. These interact with the group #2 SNPs which are selected to be variants within a ± 100 kilobase (kb) window around each SNP in group #1. Coefficients for additive and interaction effects were simulated with no minor allele frequency dependency $\alpha = 0$ (see Materials and methods). Here, we assume a heritability (**A**) $H^2 = 0.3$ or (**B**) $H^2 = 0.6$ (marked by the black dotted lines, respectively), and we vary the proportion contributed by additive effects with $\rho = \{0.2, 0.4, 0.6, 0.8\}$. The grey dotted lines represent the total contribution of additive effects in the generative model for the synthetic traits ($H^2\rho$). i-LDSC outperforms LDSC in recovering heritability across each scenario. Results are based on 100 simulations per parameter combination.

The online version of this article includes the following figure supplement(s) for figure 3:

**Figure supplement 1.** i-LDSC robustly and accurately estimates the proportions of phenotypic variance explained (PVE) by genetic effects in polygenic traits by accounting for interaction effects tagged by GWAS summary statistics.

**Figure supplement 2.** Performance of LDSC and i-LDSC on simulated polygenic traits with architectures that are determined by additive, *cis*-interaction, and gene-by-environment (G×E) effects.

**Figure supplement 3.** Performance of LDSC and i-LDSC on simulated polygenic traits with architectures that are determined by additive, *cis*-interaction, and gene-by-ancestry (G×Ancestry) effects with principal components (PCs) included in the GWAS model to correct for additional structure.

**Figure supplement 4.** Performance of LDSC and i-LDSC on simulated polygenic traits with architectures that are determined by additive, *cis*-interaction, and gene-by-ancestry (G×Ancestry) effects without correcting for the additional structure in the GWAS analysis.

**Figure supplement 5.** Performance of LDSC and i-LDSC on simulated polygenic traits with architectures that are determined by only additive and gene-by-environment (G×E) effects.

**Figure supplement 6.** Performance of LDSC and i-LDSC on simulated polygenic traits with architectures that are determined by only additive and gene-by-ancestry (G×Ancestry) effects with principal components (PCs) included in the GWAS model to correct for additional structure.

**Figure supplement 7.** Performance of LDSC and i-LDSC on simulated polygenic traits with architectures that are determined by only additive and gene-by-ancestry (G×Ancestry) effects without correcting for the additional structure in the GWAS analysis.

*Figure 3 continued on next page*

*Figure 3 continued*

**Figure supplement 8.** Performance of LDSC and i-LDSC on simulated traits with sparse architectures that are determined by only additive effects.

**Figure supplement 9.** The non-additive component estimates in i-LDSC are robust to unobserved additive effects in a haplotype.

**Figure supplement 10.** The i-LDSC framework protects against the false discovery of non-additive genetic variance when causal interacting SNPs are unobserved and the proportion of genetic variance explained by additive effects is equal to $\rho = 0.5$.

**Figure supplement 11.** The i-LDSC framework protects against the false discovery of non-additive genetic variance when causal interacting SNPs are unobserved and the proportion of genetic variance explained by additive effects is equal to $\rho = 0.8$.

**Figure supplement 12.** Bias in LDSC and i-LDSC estimates when the additive and interaction effect sizes in the generative model of complex traits are correlated.

**Figure supplement 13.** Bias in LDSC and i-LDSC estimates when interaction effect sizes in the generative model of complex traits are a linear or squared function of the the additive effects.

---

genotyped SNPs, i-LDSC accurately partitions genetic effects into additive versus *cis*-interacting components, which in turn generally leads the ability of i-LDSC to capture more genetic variance. The mean absolute error between the true generative heritability and heritability estimates produced by i-LDSC and LDSC are shown in *Supplementary files 1 and 2*, respectively. Generally, the error in heritability estimates is higher for LDSC than it is for i-LDSC across each of the scenarios that we consider.

Next, we perform an additional set of simulations where we explore other common generative models for complex trait architecture that involve non-additive genetic effects. Specifically, we compare heritability estimates from LDSC and i-LDSC in the presence of additive effects, *cis*-acting interactions, and a third source of genetic variance stemming from either gene-by-environment (G×E) or or gene-by-ancestry (G×Ancestry) effects. Details on how these components were generated can be found in Materials and Methods. In general, i-LDSC underestimates overall heritability when additive effects and *cis*-acting interactions are present alongside G×E (*Figure 3—figure supplement 2*) and/or G×Ancestry effects when PCs are included as covariates (*Figure 3—figure supplement 3*). Notably, when PCs are not included to correct for residual stratification, both LDSC and i-LDSC can yield unbounded heritability estimates greater than 1 (*Figure 3—figure supplement 4*). Also interestingly, when we omit *cis*-interactions from the generative model (i.e. the genetic architecture of simulated traits is only made up of additive and G×E or G×Ancestry effects), i-LDSC will still estimate a nonzero genetic variance component with the *cis*-interaction LD scores (*Figure 3—figure supplements 5–7*). Collectively, these results empirically show the important point that *cis*-interaction scores are not enough to recover missing genetic variation for all types of trait architectures; however, they are helpful in recovering phenotypic variation explained by statistical interaction effects. Recall that the linear relationship between (expected) $\chi^2$ test statistics and LD scores proposed by the LDSC framework holds when complex traits are generated under the polygenic model where all causal variants have the same expected contribution to phenotypic variation. When *cis*-interactions affect genetic architecture (e.g. in our earlier simulations in *Figure 3*), these assumptions are violated in LDSC, but the inclusion of the additional nonlinear scores in i-LDSC help recover the relationship between the expectation of $\chi^2$ test statistics and LD.

As a further demonstration of how i-LDSC performs when assumptions of the original LD score model are violated, we also generated synthetic phenotypes with sparse architectures using the spike-and-slab model (*Zhou et al., 2013*). Here, traits were simulated with solely additive effects, but this time only variants with the top or bottom $\{1, 5, 10, 25, 50, 100\}$ percentile of LD scores were given nonzero effects (see Materials and methods). Breaking the relationship assumed under the LDSC framework between LD scores and chi-squared statistics (i.e. that they are generally positively correlated) led to unbounded estimates of heritability in all but the (polygenic) scenario when 100% of SNPs contributed to the phenotypic variation (*Figure 3—figure supplement 8*).

Finally, we performed a set of polygenic simulations to assess if i-LDSC estimates of non-additive genetic variance could be spuriously inflated due to either (*i*) unobserved additive effects (see, for example, *Hemani et al., 2014*), (*ii*) unobserved SNPs that are involved in genetic interactions, or by (*iii*) nonzero correlation between the additive and interaction effect sizes in the

generative model (i.e. breaking the independence assumption in *Equation 2*). In the first setting, we observed that, across a range of both minor allele frequencies and effect sizes, the omission of causal haplotypes had a negligible effect on the estimated value of the coefficients in i-LDSC (*Figure 3—figure supplement 9*). We hypothesize this is due to the fact that the simulations were done for polygenic architectures where all SNPs have at least an additive effect. As a result, not observing a small subset of SNPs does not hinder the ability of i-LDSC to estimate genetic variance because the effect size of each SNP is small. If these simulations were conducted for sparse architectures, we would have likely seen a greater impact on i-LDSC; although, we have already shown the LD score regression framework to be uncalibrated for traits with sparse genetic architectures (again see *Figure 3—figure supplement 8*). In the second setting, we observed that the i-LDSC framework protects against the false discovery of non-additive genetic effects and underestimates the variance component $\vartheta$ when causal variants involved in pairwise interactions were unobserved (*Figure 3—figure supplements 10 and 11*). As a direct comparison, estimates of the additive variance component $\tau$ in i-LDSC were not affected by the unobserved interacting variants. Lastly, in the third setting, we observed that the mean estimate of the genetic variance in both LDSC and i-LDSC had a slight upward bias as the correlation between additive and interaction effect sizes in the generative model increased; however, the median of these bias estimates was still near zero across all simulated scenarios and their corresponding replicates (*Figure 3—figure supplements 12 and 13*).

## Application of i-LDSC to the UK Biobank and BioBank Japan

To assess whether pairwise interaction genetic effects are significantly affecting estimates of heritability in empirical biobank data, we applied i-LDSC to 25 continuous quantitative traits from the UK Biobank and BioBank Japan (*Supplementary file 3*). Protocols for computing GWAS summary statistics for the UK Biobank are described in the Materials and methods; while pre-computed summary statistics for BioBank Japan were downloaded directly from the consortium website (https://pheweb.jp/downloads). We release the *cis*-acting SNP-by-SNP interaction LD scores used in our analyses on the i-LDSC GitHub repository from two reference groups in the 1000 Genomes: 489 individuals from the European superpopulation (EUR) and 504 individuals from the East Asian (EAS) superpopulation (see also *Supplementary files 4 and 5*).

In each of the 25 traits, we analyzed in the UK Biobank, we detected significant proportions of estimated genetic variation stemming from tagged pairwise *cis*-interactions (*Table 1*). This includes many canonical traits of interest in heritability analyses: height, cholesterol levels, urate levels, and both systolic and diastolic blood pressure. Our findings in *Table 1* are supported by multiple published studies identifying evidence of non-additive effects playing a role in the architectures of different traits of interest. For example, *Li et al., 2020* found evidence for genetic interactions that contributed to the pathogenesis of coronary artery disease. It was also recently shown that non-additive genetic effects plays a significant role in body mass index (*Song et al., 2022*). Generally, we find that the traditional LDSC produces lower estimates of trait heritability because it does not consider the additional sources of genetic signal that i-LDSC does (*Table 1*). In BioBank Japan, 23 of the 25 traits analyzed had a significant nonlinear component detected by i-LDSC — with HDL and triglyceride levels being the only exceptions.

For each of the 25 traits that we analyzed, we found that the i-LDSC heritability estimates are significantly correlated with corresponding estimates from LDSC in both the UK Biobank ($r^2 = 0.988$, $P = 5.936 \times 10^{-24}$) and BioBank Japan ($r^2 = 0.849$, $P = 6.061 \times 10^{-11}$) as shown in *Figure 4A*. Additionally, we found that the heritability estimates for the same traits between the two biobanks are highly correlated according to both LDSC ($r^2 = 0.848$, $P = 7.166 \times 10^{-11}$) and i-LDSC ($r^2 = 0.666$, $P = 6.551 \times 10^{-7}$) analyses as shown in *Figure 4B*. After comparing the i-LDSC heritability estimates to LDSC, we then assessed whether there was significant difference in the amount of phenotypic variation explained by the non-additive genetic effect component in the GWAS summary statistics derived from the the UK Biobank and BioBank Japan (i.e. comparing the estimates of $\vartheta$; see *Figure 4—figure supplement 1A*). We show that, while heterogeneous between traits, the phenotypic variation explained by genetic interactions is relatively of the same magnitude for both biobanks ($r^2 = 0.372$, $P = 0.0119$). Notably, the trait with the most significant evidence of tagged *cis*-interaction effects in GWAS summary statistics is height which is known to have a highly polygenic architecture.

**Table 1.** i-LDSC heritability estimates and p-values highlighting statistically significant contributions of tagged pairwise genetic interaction effects for 25 traits in the UK Biobank and BioBank Japan.

Here, LDSC heritability estimates are included as a baseline. The difference between the approaches is that the i-LDSC heritability estimates include proportions of phenotypic variation that are explained by tagged non-additive variation (see columns with estimates of $\vartheta$). Note that all 25 traits analyzed in the UK Biobank and 23 of the 25 traits analyzed in BioBank Japan have a statistically significant amount of tagged non-additive genetic effects as detected by the *cis*-interaction LD score (p < 0.05). The two traits without significant tagged non-additive genetic effects in BioBank Japan were HDL (p = 0.081) and Triglyceride (p = 0.110). These traits are indicated by *. The i-LDSC p-values are related to the estimates of the $\vartheta$ coefficients which are also displayed in *Figure 4*.

| Trait | UKB (LDSC) | UKB (i-LDSC) | UKB $\widehat{\vartheta}$ | UKB p-value | BBJ (LDSC) | BBJ (i-LDSC) | BBJ $\widehat{\vartheta}$ | BBJ p-value |
|---|---|---|---|---|---|---|---|---|
| Basophil | 0.0250 | 0.0315 | 0.0065 | $1.572\times 10^{-12}$ | 0.0684 | 0.1548 | 0.0864 | 0.025 |
| BMI | 0.1757 | 0.2349 | 0.0592 | $3.083\times 10^{-84}$ | 0.1667 | 0.2656 | 0.0989 | $2.438\times 10^{-18}$ |
| Cholesterol | 0.0954 | 0.0974 | 0.0020 | $1.821\times 10^{-16}$ | 0.0629 | 0.1268 | 0.0639 | $2.740\times 10^{-4}$ |
| CRP | 0.0354 | 0.0414 | 0.0060 | $9.845\times 10^{-12}$ | 0.0202 | 0.1625 | 0.1423 | 0.020 |
| DBP | 0.0940 | 0.1203 | 0.0263 | $1.118\times 10^{-65}$ | 0.0605 | 0.1267 | 0.0662 | $1.675\times 10^{-7}$ |
| EGFR | 0.1521 | 0.1999 | 0.0478 | $1.187\times 10^{-46}$ | 0.1010 | 0.1225 | 0.0215 | $4.232\times 10^{-5}$ |
| Eosinophil | 0.1055 | 0.1375 | 0.0320 | $1.230\times 10^{-18}$ | 0.0785 | 0.1973 | 0.1188 | 0.001 |
| HBA1C | 0.0906 | 0.1083 | 0.0177 | $1.578\times 10^{-26}$ | 0.1057 | 0.1308 | 0.0251 | 0.031 |
| HDL* | 0.1599 | 0.1768 | 0.0169 | $9.636\times 10^{-37}$ | 0.1590 | 0.1838 | 0.0248 | 0.081 |
| Height | 0.3675 | 0.4815 | 0.1140 | $1.038\times 10^{-64}$ | 0.3941 | 0.7336 | 0.3395 | $7.433\times 10^{-33}$ |
| Hematocrit | 0.1078 | 0.1352 | 0.0274 | $2.479\times 10^{-25}$ | 0.0752 | 0.0928 | 0.0176 | $3.689\times 10^{-5}$ |
| Hemoglobin | 0.1177 | 0.1433 | 0.0256 | $4.284\times 10^{-27}$ | 0.0702 | 0.0752 | 0.0050 | $9.037\times 10^{-4}$ |
| LDL | 0.0802 | 0.0859 | 0.0057 | $5.087\times 10^{-13}$ | 0.0745 | 0.1438 | 0.0693 | 0.018 |
| Lymphocyte | 0.0402 | 0.0501 | 0.0099 | $4.906\times 10^{-19}$ | 0.0844 | 0.1757 | 0.0913 | $5.479\times 10^{-5}$ |
| MCH | 0.1361 | 0.1597 | 0.0236 | $1.785\times 10^{-25}$ | 0.1536 | 0.2831 | 0.1295 | $1.042\times 10^{-5}$ |
| MCHC | 0.0317 | 0.0364 | 0.0047 | $3.730\times 10^{-12}$ | 0.0571 | 0.0650 | 0.0079 | 0.027 |
| MCV | 0.1630 | 0.1902 | 0.0272 | $1.180\times 10^{-29}$ | 0.1530 | 0.2818 | 0.1288 | $1.042\times 10^{-5}$ |
| Monocyte | 0.0788 | 0.0955 | 0.0167 | $5.257\times 10^{-18}$ | 0.0888 | 0.1549 | 0.0661 | 0.004 |
| Neutrophil | 0.1102 | 0.1391 | 0.0289 | $1.777\times 10^{-33}$ | 0.1191 | 0.2114 | 0.0923 | $5.050\times 10^{-5}$ |
| Platelet | 0.1992 | 0.2447 | 0.0455 | $2.303\times 10^{-37}$ | 0.1565 | 0.2436 | 0.0871 | $7.724\times 10^{-9}$ |
| RBC | 0.1574 | 0.1933 | 0.0359 | $3.292\times 10^{-31}$ | 0.1203 | 0.2068 | 0.0865 | $5.972\times 10^{-8}$ |
| SBP | 0.0954 | 0.1201 | 0.0247 | $8.660\times 10^{-75}$ | 0.0769 | 0.1604 | 0.0835 | $9.075\times 10^{-10}$ |
| Triglycerides* | 0.1061 | 0.1204 | 0.0143 | $1.410\times 10^{-26}$ | 0.1171 | 0.2670 | 0.1499 | 0.110 |
| Urate | 0.1217 | 0.1550 | 0.0333 | $9.642\times 10^{-38}$ | 0.1395 | 0.3462 | 0.2067 | 0.015 |
| WBC | 0.0962 | 0.1250 | 0.0288 | $9.866\times 10^{-34}$ | 0.1024 | 0.2266 | 0.1242 | $1.346\times 10^{-8}$ |

The intercepts estimated by LDSC and i-LDSC are also highly correlated in both the UK Biobank and the BioBank Japan (*Figure 4—figure supplement 1B*). Recall that these intercept estimates represent the confounding factor due to uncontrolled effects. For LDSC, this does include phenotypic variation that is due to unaccounted for pairwise statistical genetic interactions. The i-LDSC intercept estimates tend to be correlated with, but are generally different than, those computed with LDSC — empirically indicating that non-additive genetic variation is partitioned away and is missed when using the standard LD score alone. This result shows similar patterns in both the UK Biobank ($r^2 = 0.888$, $P = 1.962 \times 10^{-12}$) and BioBank Japan ($r^2 = 0.813$, $P = 7.814 \times 10^{-10}$).

Lastly, we performed an additional analysis in the UK Biobank where the *cis*-interaction scores are included as an annotation alongside 97 other functional categories in the stratified-LD score

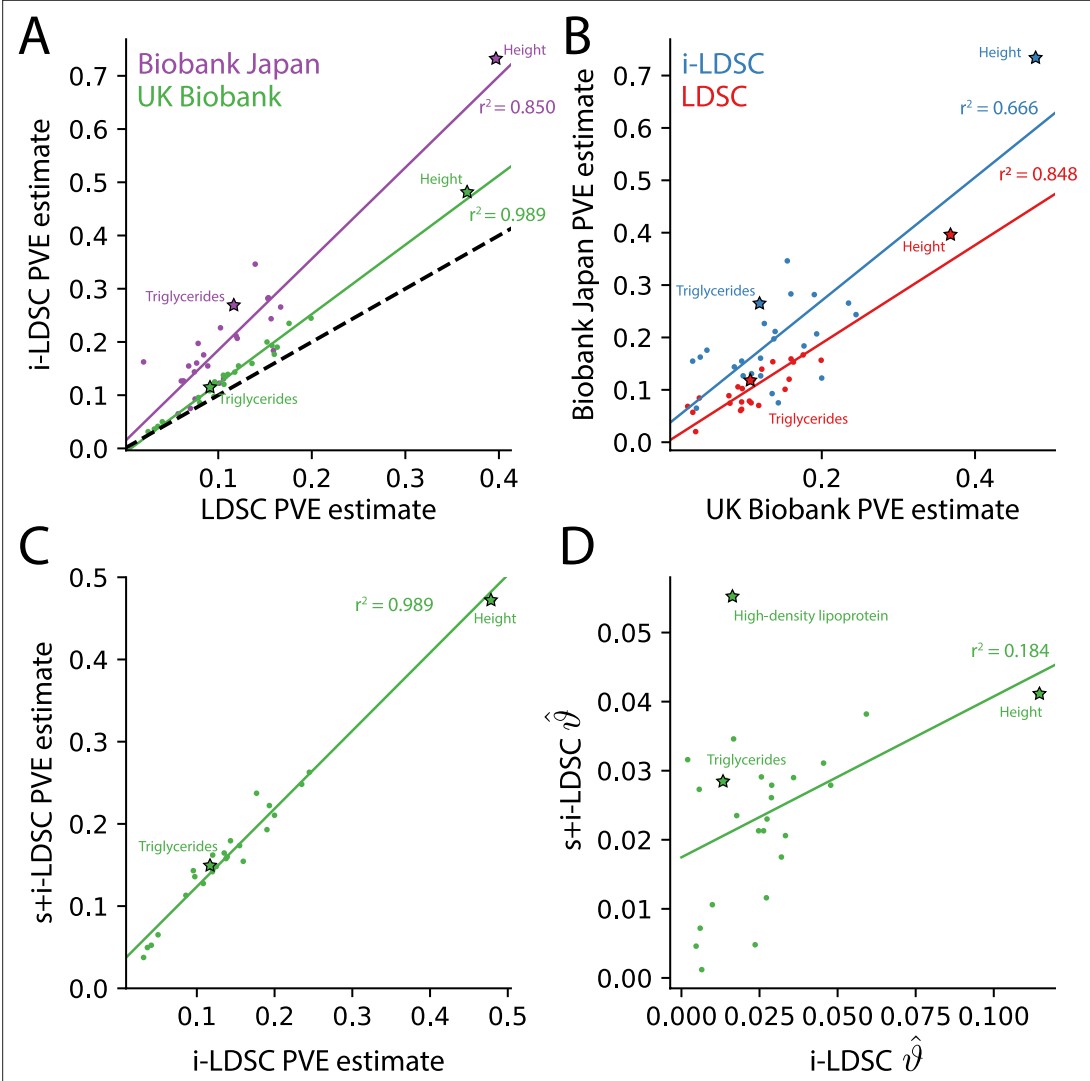

**Figure 4.** The i-LDSC framework recovers heritability and provides estimates of tagged *cis*-interactions in GWAS summary statistics ($\vartheta$) for 25 quantitiative traits in the UK Biobank and BioBank Japan. (**A**) In both the UK Biobank (green) and BioBank Japan (purple), estimates of phenotypic variance explained (PVE) by genetic effects from i-LDSC and LDSC are highly correlated for 25 different complex traits. The Spearman correlation coefficient between heritability estimates from LDSC and i-LDSC for the UK Biobank and BioBank Japan are $r^2 = 0.989$ and $r^2 = 0.850$, respectively. The $y = x$ dotted line represents the values at which estimates from both approaches are the same. (**B**) PVE estimates from the UK Biobank are better correlated with those from the BioBank Japan across 25 traits using LDSC (Spearman $r^2 = 0.848$) than i-LDSC (Spearman $r^2 = 0.666$). (**C**) Both the original and stratified LDSC models recover the same amount of PVE when the *cis*-interaction LD score is included as an additional component in the UK Biobank analysis (Spearman $r^2 = 0.989$). These models are listed as i-LDSC and s+i-LDSC, respectively. For s+i-LDSC, we included 97 functional annotations from Gazal et al. to estimate heritability. (**D**) Estimates of non-additive variance components in i-LDSC versus s+i-LDSC (Spearmen $r^2 = 0.184$). While not statistically significant in the stratified analysis with the additional annotations, the non-additive component still makes nonzero contributions to the PVE estimation for all 25 traits in the UK Biobank (see *Tables 1 and 2*).

The online version of this article includes the following figure supplement(s) for figure 4:

**Figure supplement 1.** Additional results from applying LDSC and i-LDSC for 25 quantitiative traits in the UK Biobank and BioBank Japan.

regression framework and its software s-LDSC (*Gazal et al., 2017*; Materials and methods). Here, s-LDSC heritability estimates still showed an increase with the interaction scores versus when the publicly available functional categories were analyzed alone, but albeit at a much smaller magnitude (*Table 2*). The contributions from the pairwise interaction component to the overall estimate of genetic variance ranged from 0.005 for MCHC ($P = 0.373$) to 0.055 for HDL ($P = 0.575$; *Figure 4C and D*). Furthermore, in this analysis, the estimates of the non-additive components were no longer statistically significant for any of the traits in the UK Biobank (*Table 2*). Despite this, these results highlight

**Table 2.** Comparison of s-LDSC and i-LDSC estimates of phenotypic variance explained (PVE) by genetic effects for 25 complex traits in the UK Biobank.

Here, we use stratified LD score regression (s-LDSC) to partition heritability across different genomic elements (*Finucane et al., 2015*). We used 97 functional annotations from Gazal et al. to estimate heritability in 25 traits. We then appended *cis*-interaction LD scores as an additional annotation to obtain heritability estimates (this method is referred to as s+i-LDSC in the table). p-values for the s+i-LDSC model detailing the contributions of tagged non-additive genetic effects for 25 traits are provided in the last column. Note that, while not statistically significant in this stratified analysis with the additional annotations, the non-additive component still makes nonzero contributions to the PVE estimation for all 25 traits.

| Trait | UKB PVE (s-LDSC) | UKB PVE (s+i-LDSC) | s+i-LDSC p-value |
|---|---|---|---|
| Basophil | 0.0363 | 0.0375 | 0.4728 |
| BMI | 0.2100 | 0.2482 | 0.8126 |
| Cholesterol | 0.1042 | 0.1358 | 0.6202 |
| CRP | 0.0452 | 0.0524 | 0.6483 |
| DBP | 0.1228 | 0.1441 | 0.6125 |
| EGFR | 0.1826 | 0.2105 | 0.8507 |
| Eosinophil | 0.1403 | 0.1578 | 0.1867 |
| HBA1C | 0.1040 | 0.1275 | 0.6917 |
| HDL | 0.1820 | 0.2373 | 0.5754 |
| Height | 0.4315 | 0.4726 | 0.5224 |
| Hematocrit | 0.1416 | 0.1646 | 0.3956 |
| Hemoglobin | 0.1504 | 0.1795 | 0.2299 |
| LDL | 0.0858 | 0.1131 | 0.8812 |
| Lymphocyte | 0.0545 | 0.0651 | 0.1453 |
| MCH | 0.1497 | 0.1545 | 0.0968 |
| MCHC | 0.0450 | 0.0496 | 0.3728 |
| MCV | 0.1814 | 0.1930 | 0.1530 |
| Monocyte | 0.1085 | 0.1431 | 0.5421 |
| Neutrophil | 0.1320 | 0.1599 | 0.2499 |
| Platelet | 0.2317 | 0.2628 | 0.7371 |
| RBC | 0.1933 | 0.2223 | 0.3197 |
| SBP | 0.1206 | 0.1419 | 0.1100 |
| Triglycerides | 0.1335 | 0.1621 | 0.5301 |
| Urate | 0.1530 | 0.1736 | 0.1177 |
| WBC | 0.1221 | 0.1482 | 0.5155 |

the ability of the i-LDSC framework to identify sources of 'missing' phenotypic variance explained in heritability estimation. Importantly, moving forward, we suggest using the *cis*-interaction scores with additional annotations whenever they are available as it provides more conservative estimates of the role of non-additive effects on trait architecture.

## Discussion

In this paper, we present i-LDSC, an extension of the LD score regression framework which aims to recover missing heritability from GWAS summary statistics by incorporating an additional score

that measures the non-additive genetic variation that is tagged by genotyped SNPs. Here, we demonstrate how i-LDSC builds upon the original LDSC model through the development of new 'cis-interaction' LD scores which help to investigate signals of cis-acting SNP-by-SNP interactions (*Figure 1* and *Figure 1—figure supplements 1–5*). Through extensive simulations, we show that i-LDSC is well-calibrated under the null model when polygenic traits are generated only by additive effects (*Figure 2* and *Figure 2—figure supplements 1–2*), we highlight that i-LDSC provides greater heritability estimates over LDSC when traits are indeed generated with cis-acting SNP-by-SNP inter-action effects (*Figure 3* and *Figure 3—figure supplement 1*, and *Supplementary files 1 and 2*), and we tested the robustness of i-LDSC on phenotypes where assumptions of the original LD score model are violated (*Figure 3—figure supplements 2–13*). Finally, in real data, we show examples of many traits with estimated GWAS summary statistics that tag cis-interaction effects in the UK Biobank and BioBank Japan (*Figure 4* and *Figure 4—figure supplement 1*, *Tables 1 and 2*, and *Supplementary files 3-5*). We have made i-LDSC a publicly available command line tool that requires minimal updates to the computing environment used to run the original implementation of LD score regression. In addition, we provide pre-computed cis-interaction LD scores calculated from the European (EUR) and East Asian (EAS) reference populations in the 1000 Genomes phase 3 data (see Data and Software Availability under Materials and Methods).

The current implementation of the i-LDSC framework offers many directions for future development and applications. First, an area of future work would be to explore how the relationship between cis-interaction LD scores and interaction effect sizes from the generative model of complex traits might bias heritability estimates provided by i-LDSC (e.g., similar to the relationship we explored between the standard LD scores and linear effect sizes in *Figure 3—figure supplement 8*). Second, as we showed with our simulation studies (*Figure 3—figure supplements 2–8*), the cis-interaction LD scores that we propose are not always enough to recover explainable non-additive genetic effects for all types of trait architectures. While we focus on pairwise cis-acting SNP-by-SNP statistical inter-actions in this work, the theoretical concepts underlying i-LDSC can easily be adapted to other types of interactions as well. Third, in our analysis of the UK Biobank and BioBank Japan, we showed that the inclusion of additional categories via frameworks such as stratified LD score regression (*Finucane et al., 2015*) can be used to provide more refined heritability estimates from GWAS summary statistics while accounting for linkage (see results in *Table 1* versus *Table 2*). A key part of our future work is to continue to explore whether considering functional annotation groups would also improve our ability to identify tagged non-additive genetic effects. Lastly, we have only focused on analyzing one phenotype at a time in this study. However, many previous studies have extensively shown that modeling multiple phenotypes can often dramatically increase power (*Runcie et al., 2020*; *Stamp et al., 2022*). Therefore, it would be interesting to extend the i-LDSC framework to multiple traits to study nonlinear genetic correlations in the same way that LDSC was recently extended to uncover additive genetic correlation maps across traits (*Naqvi et al., 2021*).

## Materials and methods
### Generative statistical model for complex traits

Our goal in this study is to reanalyze summary statistics from genome-wide association studies (GWAS) and estimate heritability while accounting for both additive genetic associations and tagged interaction effects. We begin by assuming the following generative linear model for complex traits which can be seen as an extended view of *Equation 1* in the main text

$$\mathbf{y} = b_0 + \mathbf{X}\beta + \mathbf{X}_D\boldsymbol{\omega} + \mathbf{W}\theta + \varepsilon, \qquad \varepsilon \sim \mathcal{N}(\mathbf{0}, (1-H^2)\mathbf{I}), \tag{7}$$

where $\mathbf{y}$ denotes an $N$-dimensional vector of phenotypic states for a quantitative trait of interest measured in $N$ individuals; $b_0$ is an intercept term; $\mathbf{X}$ is an $N \times J$ matrix of genotypes, with $J$ denoting the number of single nucleotide polymorphism (SNPs) encoded as $\{0, 1, 2\}$ copies of a reference allele at each locus; $\beta = (\beta_1, \dots, \beta_J)$ is a $J$-dimensional vector containing the true additive effect sizes for an additional copy of the reference allele at each locus on $\mathbf{y}$; $X_D$ is an $N \times J$ matrix that represents the dominance for each genotype encoded as $\{0, 1, 1\}$ with corresponding effect sizes $\boldsymbol{\omega}$; $\mathbf{W}$ is an $N \times M$ matrix of genetic interactions; $\theta = (\theta_1, \dots, \theta_M)$ is an $M$-dimensional vector containing the interaction effect sizes; $\varepsilon$ is a normally distributed error term with mean zero and variance scaled according

to the proportion of phenotypic variation not explained by the broad-sense heritability of the trait, denoted by $H^2$; and $\mathbf{I}$ denotes an $N \times N$ identity matrix. Note that the encoding for dominance in $\mathbf{X}_D$ was chosen because it imposes orthogonality with the genotype encoding in $\mathbf{X}$ (*Purcell et al., 2007*; *Vitezica et al., 2017*; *Palmer et al., 2023*).

For convenience, we will assume that the genotype matrix (column-wise), the dominance matrix (also column-wise), and trait of interest have all been standardized (*Strandén and Christensen, 2011*; *de Los Campos et al., 2013*; *Zhou et al., 2013*). Furthermore, while the matrix $\mathbf{W}$ could encode any source of non-additive genetic interactions (e.g. gene-by-environmental effects) in theory, we limit our focus in this study to trait architectures that have been generated with contributions stemming from *cis*-acting statistical SNP-by-SNP (or pairwise) interactions. To that end, we assume that the columns of $\mathbf{W}$ are the Hadamard (element-wise) product between genotypic vectors of the form $\mathbf{x}_j \circ \mathbf{x}_k$ for the $j$-th and $k$-th variants. We also want to point out that the generative formulation of *Equation 7* can also be easily extended to accommodate other fixed effects (e.g. age, sex, or genotype principal components), as well as other random effects terms that can be used to account for sample non-independence due to other environmental factors.

As a final set of assumptions, we will let the intercept term $b_0$ be a fixed parameter while allowing the other coefficients to follow independent Gaussian distributions with variances proportional to their individual contributions to the trait heritability (*Yang et al., 2010*; *Wu et al., 2011*; *Zhou et al., 2013*; *Jiang and Reif, 2015*; *Crawford et al., 2017*)

$$\beta_j \sim \mathcal{N}(0, \varphi_\beta^2/J), \qquad \omega_j \sim \mathcal{N}(0, \varphi_\omega^2/J), \qquad \theta_m \sim \mathcal{N}(0, \varphi_\theta^2/M), \tag{8}$$

for $j = 1, \ldots, J$ and $m = 1, \ldots, M$. The broad-sense heritability of the trait is defined as $H^2 = \varphi_\beta^2 + \varphi_\omega^2 + \varphi_\theta^2$. Under the generative model in *Equation 7*, we then say that $\mathbb{V}[\mathbf{X}\beta] = \varphi_\beta^2$ is the proportion of phenotypic variation contributed by additive SNP effects, $\mathbb{V}[\mathbf{X}_D\omega] = \varphi_\omega^2$ is the proportion of phenotypic variation contributed by dominance effects, and the set of interactions involving some subset of causal SNPs contribute the remaining proportion to the heritability $\mathbb{V}[\mathbf{W}\theta] = \varphi_\theta^2$. As we mentioned in the main text, we recognize that the appropriateness of treating genetic effects as random variables in analytical derivations has been questioned (*de Los Campos et al., 2015*), but our simulation studies show that i-LDSC accurately recovers non-additive genetic variance in *Equation 7* under a broad range of conditions.

## Orthogonality between additive and non-additive genetic effects

Assuming that the effect sizes $\{\beta, \omega, \theta\}$ in *Equation 8* follow independent and zero mean Gaussian distributions leads to orthogonality between the additive and non-additive components in *Equation 7*. Since the genotypes $\mathbf{X}$ and the dominance values $\mathbf{X}_D$ are fixed orthogonal matrices, it is straightforward to show that $\mathrm{Cov}[\mathbf{X}\beta, \mathbf{X}_D\omega] = 0$ (*Vitezica et al., 2017*; *Palmer et al., 2023*). The same relationship can be shown for the additive and the pairwise interaction genetic effects where

$$
\begin{aligned}
\mathrm{Cov}[\mathbf{X}\beta, \mathbf{W}\theta] \quad &= \mathbb{E}[\beta^\intercal \mathbf{X}^\intercal \mathbf{W}\theta] - \mathbb{E}[\beta^\intercal \mathbf{X}^\intercal]\mathbb{E}[\mathbf{W}\theta] \\
&= \mathbb{E}\left[\sum_{rs} \beta_r \left(\mathbf{X}^\intercal \mathbf{W}\right)_{rs} \theta_s\right] - \mathbb{E}[\beta^\intercal]\mathbf{X}^\intercal \mathbf{W}\mathbb{E}[\theta] \\
&= \sum_{rs} \left(\mathbf{X}^\intercal \mathbf{W}\right)_{rs} \mathbb{E}[\beta_r \theta_s] - \mathbf{0}^\intercal \mathbf{X}^\intercal \mathbf{W}\mathbf{0} \\
&= \sum_{rs} \left(\mathbf{X}^\intercal \mathbf{W}\right)_{rs} \mathbb{E}[\beta_r]\mathbb{E}[\theta_s] \\
&= 0
\end{aligned}
\tag{9}
$$

with $\mathbf{x}_j$ and $\mathbf{w}_m$ denoting the $j$-th and $m$-th column of the individual-level genotype matrix $\mathbf{X}$ and the interaction matrix $\mathbf{W}$, respectively. Note that a similar derivation to *Equation 9* can also be done for the dominance and pairwise genetic interaction effects. This concept of orthogonality is important because we want to preserve a unique partitioning of genetic variance when modeling a trait of interest.

## Genotypes and their interactions are correlated despite being linearly independent

The design matrices $\mathbf{X}$ and $\mathbf{W}$ in *Equation 7* are not linearly dependent because the pairwise interactions between two SNPs are encoded as the Hadamard product of two genotypic vectors in the form

$\mathbf{x}_j \circ \mathbf{x}_k$ (which is a nonlinear function). Linear dependence would have implied that one could find a transformation between a SNP and an interaction term in the form $\mathbf{w}_m = c \times \mathbf{x}_j$ for some constant $c$. However, despite their linear independence, $\mathbf{X}$ and $\mathbf{W}$ are themselves not orthogonal and still have a nonzero correlation. This implies that the inner product between genotypes and their interactions is nonzero $\mathbf{X}^\mathsf{T}\mathbf{W} \neq 0$. To see this, we focus on a focal SNP $\mathbf{x}_j$ and consider three different types of interactions:

- Scenario I: Interaction between a focal SNP with itself ($\mathbf{x}_j \circ \mathbf{x}_j$).
- Scenario II: Interaction between a focal SNP with a different SNP ($\mathbf{x}_j \circ \mathbf{x}_k$).
- Scenario III: Interaction between a focal SNP with a pair of different SNPs ($\mathbf{x}_k \circ \mathbf{x}_l$).

The following derivations rely on the fact that: (1) we assume that genotypes have been mean-centered and scaled to have unit variance, and (2) under Hardy-Weinberg equilibrium, SNPs marginally follow a binomial distribution $\mathbf{x}_j \sim \mathrm{Bin}(2, p)$ where $p$ represents the minor allele frequency (MAF) (*Wray et al., 2007*; *Lippert et al., 2013*).

## Scenario I

The covariance between a focal SNP and an interaction with itself is $\mathrm{Cov}[\mathbf{x}_j, \mathbf{x}_j\mathbf{x}_j] = \mathbb{E}[\mathbf{x}_j^3] - \mathbb{E}[\mathbf{x}_j]\mathbb{E}[\mathbf{x}_j^2]$. With mean-centered SNPs, this is proportional to $\mathbb{E}[\mathbf{x}_j^3] = (q - p)/\sqrt{2pq}$ which is the skewness of the binomial distribution where, again, $p =$ MAF and $q = $ 1-MAF of the $j$-th SNP.

## Scenario II

Assume that we have two SNPs, $\mathbf{x}_j \sim \mathrm{Bin}(2, p_j)$ and $\mathbf{x}_k \sim \mathrm{Bin}(2, p_k)$ where $p_j$ and $p_k$ represent their respective minor allele frequencies. We want to compute the correlation between $\mathbf{x}_j$ and the interaction $\mathbf{x}_j\mathbf{x}_k$ where $\mathrm{Cov}[\mathbf{x}_j, \mathbf{x}_j\mathbf{x}_k] = \mathbb{E}[\mathbf{x}_j^2\mathbf{x}_k] - \mathbb{E}[\mathbf{x}_j]\mathbb{E}[\mathbf{x}_j\mathbf{x}_k]$. Again, with the mean-centered assumption, the covariance is proportional to the expectation $\mathbb{E}[\mathbf{x}_j^2\mathbf{x}_k]$. Here, with SNPs taking on values $\{0, 1, 2\}$, the joint distribution between $\mathbf{x}_j^2$ and $\mathbf{x}_k$ can be written out as the following *Kang and Jung, 2001*:

|  | $\mathbf{x}_j^2 = 0$ | $\mathbf{x}_j^2 = 1$ | $\mathbf{x}_j^2 = 4$ |
|---|---|---|---|
| $\mathbf{x}_k = 0$ | $u_{jk}^2$ | $2u_{jk}(1 - p_k - u_{jk})$ | $(1 - p_k - u_{jk})^2$ |
| $\mathbf{x}_k = 1$ | $2u_{jk}(1 - p_j - u_{jk})$ | $2u_{jk}(u_{jk} + p_j + p_k - 1)+$ <br> $2(1 - p_j - u_{jk})(1 - p_k - u_{jk})$ | $2(u_{jk} + p_j + p_k - 1)(1 - p_k - u_{jk})$ |
| $\mathbf{x}_k = 2$ | $(1 - p_j - u_{jk})^2$ | $2(u_{jk} + p_j + p_k - 1)(1 - p_j - u_{jk})$ | $(u_{jk} + p_j + p_k - 1)^2$ |

where $u_{jk} = (1 - p_j)(1 - p_k) + r_{jk}\sqrt{p_j p_k (1 - p_j)(1 - p_k)}$ and $r_{jk}$ is the Pearson correlation or linkage disequilibrium (LD) between the $j$-th and $k$-th SNPs.

## Scenario III

The covariance between a focal SNP and an interaction with a pair of different SNPs $\mathrm{Cov}[\mathbf{x}_j, \mathbf{x}_k\mathbf{x}_l]$ will be nonzero if the $j$-th SNP is correlated with either variant (i.e., $r_{jk} \neq 0$ or $r_{jl} \neq 0$).

## Traditional estimation of additive GWAS summary statistics

As previously mentioned, the key to this work is that SNP-level GWAS summary statistics can also tag non-additive genetic effects when there is a nonzero correlation between individual-level genotypes and their interactions (as defined in *Equation 7*). Throughout the rest of this section, we will use $\mathbf{X}^\mathsf{T}\mathbf{X}/N$ to denote the LD or pairwise correlation matrix between SNPs. We will then let $\mathbf{R}$ represent an LD matrix empirically estimated from external data (e.g. directly from GWAS study data, or using a pairwise LD map from a population that is representative of the samples analyzed in the GWAS study). The important property here is the following

$$\mathbb{E}[\mathbf{X}^\mathsf{T}\mathbf{X}] \approx N\mathbf{R}, \qquad \mathbb{E}[\mathbf{x}_j^\mathsf{T}\mathbf{x}_j] \approx N, \qquad \mathbb{E}[\mathbf{x}_j^\mathsf{T}\mathbf{x}_k] \approx Nr_{jk} \tag{10}$$

where the term $r_{jk}$ is again defined as the Pearson correlation coefficient between the $j$-th and $k$-th SNPs, respectively.

In traditional GWAS studies, summary statistics of the true additive effects $\beta = (\mathbf{X}^\mathsf{T}\mathbf{X})^{-1}\mathbf{X}^\mathsf{T}\mathbf{y}$ in *Equation 7* are typically derived by computing a marginal least squares estimate with the observed data

$$\widehat{\beta}_j = (\mathbf{x}_j^\mathsf{T}\mathbf{x}_j)^{-1}\mathbf{x}_j^\mathsf{T}\mathbf{y} \qquad \Longleftrightarrow \qquad \widehat{\beta} = \mathrm{diag}(\mathbf{X}^\mathsf{T}\mathbf{X})^{-1}\mathbf{X}^\mathsf{T}\mathbf{y}. \tag{11}$$

There are two key identities that may be taken from *Equation 11*. The first uses *Equation 10* and is the approximate relationship (in expectation) between the moment matrix $\mathbf{X}^\mathsf{T}\mathbf{y}$ and the linear effect size estimates $\widehat{\beta}$:

$$\mathbb{E}[\mathbf{X}^\mathsf{T}\mathbf{y}] = \mathbb{E}[\mathrm{diag}(\mathbf{X}^\mathsf{T}\mathbf{X})\widehat{\beta}] \approx N\widehat{\beta}. \tag{12}$$

The second key point combines *Equations 10 and 12* to describe the asymptotic relationship between the observed marginal GWAS summary statistics $\widehat{\beta}$ and the joint coefficient values $\beta$ where (in expectation)

$$\mathbb{E}[\beta] = \mathbb{E}[(\mathbf{X}^\mathsf{T}\mathbf{X})^{-1}\mathbf{X}^\mathsf{T}\mathbf{y}] \approx (N\mathbf{R})^{-1}N\widehat{\beta} = \mathbf{R}^{-1}\widehat{\beta}. \tag{13}$$

After some algebra, the above mirrors a high-dimensional regression model (in expectation) where $\widehat{\beta} = \mathbf{R}\beta$ with the estimated summary statistics as the response variables and the empirically estimated LD matrix acting as the design matrix (*Hormozdiari et al., 2014*; *Hormozdiari et al., 2016*; *Zhang et al., 2018*; *Cheng et al., 2020*; *Demetci et al., 2021*). Theoretically, the resulting coefficients output from this high-dimensional model are the desired true effect size estimates used to generate the phenotype of interest.

## Additive GWAS summary statistics with tagged interaction effects

When interactions contribute to the architecture of complex traits (i.e. $\theta \neq 0$), the marginal GWAS summary statistics derived using least squares in *Equation 11* will also explain non-additive variation when there is a nonzero correlation between genotypes and their interactions. To see this, we use the concept of 'omitted variable bias' (*Barreto and Howland, 2005*) where the fitted model aims to estimate the true additive coefficients $\beta$ but does not account for contributions from the non-additive components which also contribute to trait architecture. In this case, we get the following

$$\begin{aligned} \widehat{\beta} &= \mathrm{diag}(\mathbf{X}^\mathsf{T}\mathbf{X})^{-1}\mathbf{X}^\mathsf{T}\mathbf{y} \\ &= \mathrm{diag}(\mathbf{X}^\mathsf{T}\mathbf{X})^{-1}\mathbf{X}^\mathsf{T}\left[\mathbf{X}\beta + \mathbf{X}_D\omega + \mathbf{W}\theta + \varepsilon\right]. \end{aligned} \tag{14}$$

Since we assume that the genotypes are orthogonal to both the dominance effects in *Equation 7*, we know that $\mathbf{X}^\mathsf{T}\mathbf{X}_D = 0$. This simplifies the above to be the following

$$\widehat{\beta} = \mathrm{diag}(\mathbf{X}^\mathsf{T}\mathbf{X})^{-1}\mathbf{X}^\mathsf{T}\mathbf{X}\beta + \mathrm{diag}(\mathbf{X}^\mathsf{T}\mathbf{X})^{-1}\mathbf{X}^\mathsf{T}\mathbf{W}\theta + \mathrm{diag}(\mathbf{X}^\mathsf{T}\mathbf{X})^{-1}\mathbf{X}^\mathsf{T}\varepsilon \tag{15}$$

where the matrix $\mathbf{X}^\mathsf{T}\mathbf{W}$(which we showed to be nonzero) can be interpreted as the sample correlation between individual-level genotypes and the *cis*-interactions between causal SNPs. By taking the expectation using *Equations 10 and 12*, we get the following alternative (approximate) relationship between the observed marginal GWAS summary statistics $\widehat{\beta}$ and the true coefficient values $\beta$

$$\mathbb{E}[\widehat{\beta}] = \mathbf{R}\beta + \mathbf{V}\theta, \tag{16}$$

which results from our initial assumption that the residuals are normally distributed with mean zero $\mathbb{E}[\varepsilon] = 0$ in *Equation 7*. Here, we define $\mathbf{V}$ to represent a sample estimate of the correlation between the individual-level genotypes and the non-additive genetic interaction matrix such that $\mathbb{E}[\mathbf{X}^\mathsf{T}\mathbf{W}] \approx N\mathbf{V}$. Similar to the LD matrix $\mathbf{R}$, the correlation matrix $\mathbf{V}$ is also assumed to be computed from reference panel data. Intuitively, when $\theta \neq 0$ there is additional phenotypic variation contributed by pairwise interactions that can be explained by GWAS effect size estimates. Moreover, when $\mathbf{V}\theta = 0$, then the relationship in *Equation 16* converges onto the conventional asymptotic assumption (in expectation)

between GWAS summary statistics and the true additive coefficients in *Equation 13*; *Hormozdiari et al., 2014*; *Hormozdiari et al., 2016*; *Zhang et al., 2018*; *Cheng et al., 2020*; *Demetci et al., 2021*.

## Connection to quantitative genetics theory

The concept of additive genetic effects partially explaining non-additive variation has also described in classical quantitative genetics (*Hill et al., 2008*; *Hivert et al., 2021*; *Mäki-Tanila and Hill, 2014*). Consider an individual genotyped at $J$ loci each with major and minor alleles A and B, respectively. Let $p_j$ be the allele frequency of A at the $j$-th locus, $a_j$ denote the additive effect, and $[aa]_{jk}$ be the additive-by-additive (pairwise) interaction effect between loci $j$ and $k$, and $[aaa]_{jkl}$ represent a third order interaction between loci $j$, $k$, and $l$. For simplicity in presentation, assume that dominance only makes a small contribution to the genetic variance (*Palmer et al., 2023*; *Pazokitoroudi et al., 2021*; *Zhu et al., 2015*). The population mean is given as the following

$$\mu = 2\sum_{j=1}^{J} p_j a_j + 4\sum_{j=1}^{J}\sum_{k>j}^{J} p_j p_k [aa]_{jk} + 8\sum_{j=1}^{J}\sum_{k>j}^{J}\sum_{l>k>j}^{J} p_j p_k p_l [aaa]_{jkl} + \cdots \tag{17}$$

We follow the assumption that the genetic variation in human complex traits can predominately be explained by additive effects, with the remainder variation being mostly explained by additive-by-additive effects (*Weinreich et al., 2018*; *Jiang and Reif, 2015*; *Fisher, 1919*; *Lynch and Walsh, 1998*). As a result, we will ignore the higher order interaction terms in *Equation 17*. Under Hardy-Weinberg equilibrium, we can find the average effect by taking the first derivative of the population mean with respect to the frequency of the increasing allele (*Mäki-Tanila and Hill, 2014*; *Hivert et al., 2021*). For the $j$-th SNP, the average effect (including terms up to second-order interaction) is given by the following

$$\eta_j = \frac{1}{2}\left(\frac{\partial \mu}{\partial p_j}\right) = a_j + 2\sum_{k \neq j}^{J} p_k [aa]_{jk} + O\left([aaa]_{jkl}\right) \tag{18}$$

which notably contains both the additive effect and a summation of additive-by-additive interactions between pairs of loci. The additive genetic variance for the $j$-th SNP takes on the following form

$$\begin{aligned}\sigma_A^2(j) \quad &= 2p_j(1-p_j)\left[a_j + 2\sum_{k\neq j}^{J} p_k[aa]_{jk}\right]^2 \\ &= 2p_j(1-p_j)\left[a_j^2 + 2a_j\sum_{k\neq j}^{J} p_k[aa]_{jk} + 4\left(\sum_{k\neq j}^{J} p_k[aa]_{jk}\right)^2\right]\end{aligned} \tag{19}$$

which is the product of the square of the average effect in *Equation 18* and the heterozygosity at $j$-th locus $\mathbb{V}[\mathbf{x}_j] = 2p_j(1-p_j)$ (again assuming that SNPs marginally follow a binomial distribution). The total additive variance is then obtained by summing over the $J$ loci such that $\sigma_A^2 = \sum_j \sigma_A^2(j)$ (*Falconer and Mackay, 1983*).

We can derive a parallel construction for additive genetic variance using the generative random effect model presented in *Equation 7*; *Hivert et al., 2021*. Here, we will leverage that with genotype data taken for $N$ individuals, $\sum_i x_{ij}/N = 2p_j$. Ignoring the assumed small contributions from dominance effects, the population mean for a quantitative trait $\mathbf{y}$ can be written as the following

$$\begin{aligned}\mu = \frac{1}{N}\sum_{i=1}^{N} y_i \quad &= \frac{1}{N}\sum_{i=1}^{N}\left[b_0 + \sum_{j=1}^{J} x_{ij}\beta_j + \sum_{j=1}^{J}\sum_{k>j}^{J} x_{ij}x_{ik}\theta_{jk} + \varepsilon_i\right] \\ &= b_0 + 2\sum_{j=1}^{J} p_j\beta_j + 4\sum_{j=1}^{J}\sum_{k>j}^{J} p_j p_k\theta_{jk} + \frac{1}{N}\sum_{i=1}^{N}\varepsilon_i.\end{aligned} \tag{20}$$

To find the average effect for the $j$-th locus, we this time take the first derivative of the population mean in *Equation 20* with respect to the allele frequency such that

$$\eta_j = \frac{1}{2}\left(\frac{\partial \mu}{\partial p_j}\right) = \beta_j + 2\sum_{k \neq j}^{J} p_k\theta_{jk} \tag{21}$$

which, similar to the theoretical form in quantitative genetics, also contains both the additive effect of the $j$-th SNP and additional terms encoding the interaction effect between the $j$-th SNP and all other

variants in the data. Once again, under Hardy-Weinberg equilibrium, the additive variance for the $j$-th SNP is found as taking on the following form

$$
\begin{aligned}
\sigma_A^2(j) \quad &= 2p_j(1-p_j)\left[\beta_j + 2\sum_{k\neq j}^J p_k\theta_{jk}\right]^2 \\
&= 2p_j(1-p_j)\left[\beta_j^2 + 2\beta_j\sum_{k\neq j}^J p_k\theta_{jk} + 4\left(\sum_{k\neq j}^J p_k\theta_{jk}\right)^2\right]
\end{aligned}
\tag{22}
$$

where we can explicitly draw connections between the two frameworks by setting $\beta_j = a_j$ and $\theta_{jk} = [aa]_{jk}$. Note that when there no non-additive effects (such that $\theta = 0$), the above reduces to $\sigma_A^2 = \sum_j 2p_j(1-p_j)\beta_j^2$ which resembles the classical form for the additive genetic variance (**Lynch and Walsh, 1998**).

## Full derivation of interaction LD score regression

In order to derive the interaction LD score (i-LDSC) regression framework, recall that our goal is to recover missing heritability from GWAS summary statistics by incorporating an additional score that measures the non-additive genetic variation that is tagged by genotyped SNPs. To do this, we build upon the LD score regression framework and the LDSC software (**Bulik-Sullivan et al., 2015b**). Here, we assume nonzero contributions from *cis*-acting pairwise interaction effects in the generative model of complex traits as in **Equation 16**, and we use the observed least squares estimates from **Equation 11** to compute chi-square statistics $\chi_j^2 = N\widehat{\beta}_j^2$ for every $j = 1, \ldots, J$ variant in the data. Taking the expectation of these statistics yields

$$
\mathbb{E}[\chi_j^2] = N\mathbb{E}[\widehat{\beta}_j^2] = N\left[\mathbb{V}[\widehat{\beta}_j] + \left(\mathbb{E}[\widehat{\beta}_j]\right)^2\right].
\tag{23}
$$

We can simplify **Equation 23** in two steps. First, by combining the prior assumption in **Equation 8** and the asymptotic approximation in **Equation 16**, we can show that marginal expectation (i.e. when not conditioning on the true coefficients) $\mathbb{E}[\widehat{\beta}_j] = 0$ for all variants. Second, by conditioning on the generative model from **Equation 7**, we can use the law of total variance to simplify $\mathbb{V}[\widehat{\beta}_j]$ where

$$
\begin{aligned}
\mathbb{V}[\widehat{\beta}_j] \quad &= \mathbb{E}[\mathbb{V}[\widehat{\beta}_j \mid \mathbf{X}]] + \mathbb{V}[\mathbb{E}[\widehat{\beta}_j \mid \mathbf{X}]] \\
&\approx \mathbb{E}[\mathbb{V}[\mathbf{x}_j^\mathsf{T}\mathbf{y}/N \mid \mathbf{X}]] + 0 \\
&= \mathbb{E}\left[\frac{1}{N^2}\mathbf{x}_j^\mathsf{T}\left\{\mathbb{V}[\mathbf{y}\mid\mathbf{X}]\right\}\mathbf{x}_j\right] \\
&= \mathbb{E}\left[\frac{1}{N^2}\mathbf{x}_j^\mathsf{T}\left\{\frac{\varphi_\beta^2}{J}\mathbf{X}\mathbf{X}^\mathsf{T} + \frac{\varphi_\omega^2}{J}\mathbf{X}_D\mathbf{X}_D^\mathsf{T} + \frac{\varphi_\theta^2}{M}\mathbf{W}\mathbf{W}^\mathsf{T} + (1-H^2)\right\}\mathbf{x}_j\right] \\
&= \mathbb{E}\left[\frac{1}{N^2}\left\{\frac{\varphi_\beta^2}{J}\mathbf{x}_j^\mathsf{T}\mathbf{X}\mathbf{X}^\mathsf{T}\mathbf{x}_j + \frac{\varphi_\omega^2}{J}\mathbf{x}_j^\mathsf{T}\mathbf{X}_D\mathbf{X}_D^\mathsf{T}\mathbf{x}_j + \frac{\varphi_\theta^2}{M}\mathbf{x}_j^\mathsf{T}\mathbf{W}\mathbf{W}^\mathsf{T}\mathbf{x}_j + N(1-H^2)\right\}\right] \\
&= \mathbb{E}\left[\frac{1}{N^2}\left\{\frac{\varphi_\beta^2}{J}\mathbf{x}_j^\mathsf{T}\mathbf{X}\mathbf{X}^\mathsf{T}\mathbf{x}_j + \frac{\varphi_\theta^2}{M}\mathbf{x}_j^\mathsf{T}\mathbf{W}\mathbf{W}^\mathsf{T}\mathbf{x}_j + N(1-H^2)\right\}\right]
\end{aligned}
$$

since $\mathbf{x}_j^\mathsf{T}\mathbf{X}_D = 0$. Using the same logic from the original LDSC regression framework (**Bulik-Sullivan et al., 2015b**), we can use Isserlis' theorem **Isserlis, 1918** to write the above in terms of more familiar quantities based on sample correlations

$$
\frac{1}{N^2}\mathbf{x}_j^\mathsf{T}\mathbf{X}\mathbf{X}^\mathsf{T}\mathbf{x}_j = \sum_{k=1}^J \widetilde{r}_{jk}^2, \qquad \frac{1}{N^2}\mathbf{x}_j^\mathsf{T}\mathbf{W}\mathbf{W}^\mathsf{T}\mathbf{x}_j = \sum_{m=1}^M \widetilde{v}_{jm}^2
\tag{24}
$$

where $\widetilde{r}_{jk}$ is used to denote the sample correlation between additively-coded genotypes at the $j$-th and $k$-th variants, and $\widetilde{v}_{jm}$ is used to denote the sample correlation between the genotype of the $j$-th variant and the $m$-th genetic interaction on the phenotype of interest (again see **Equation 16**). Furthermore, we can use the delta method (only displaying terms up to $\mathcal{O}(1/N^2)$) to show that (in expectation)

$$
\mathbb{E}[\widetilde{r}_{jk}^2] \approx r_{jk}^2 + (1 - r_{jk}^2)/N, \qquad \mathbb{E}\left[\widetilde{v}_{jm}^2\right] \approx v_{jm}^2 + \left(1 - v_{jm}^2\right)/N.
\tag{25}
$$

Next, we can then approximate the quantities in *Equation 24* via the following

$$\mathbb{E}\left[\sum_{k=1}^{J} \tilde{r}_{jk}^2\right] \approx \ell_j + (J - \ell_j)/N, \qquad \mathbb{E}\left[\sum_{m=1}^{M} \tilde{v}_{jm}^2\right] \approx f_j + \left(M - f_j\right)/N \qquad (26)$$

where $\ell_j$ is the corresponding LD score for the additive effect of the $j$-th variant and $f_j$ represents the "interaction" LD score between the $j$-th SNP and all other variants in the data set (*Crawford et al., 2017*), respectively. Altogether, this leads to the specification of the univariate framework with the $j$-th SNP

$$\mathbb{E}[\chi_j^2] \approx N\left[\left(\frac{\varphi_\beta^2}{J}\right)\ell_j + \left(\frac{\varphi_\theta^2}{M}\right)f_j + \frac{1}{N}(1 - H^2)\right] = \ell_j\tau + f_j\vartheta + 1 \qquad (27)$$

where we define $\tau = N\varphi_\beta^2/J$ as estimates of the additive genetic signal, the coefficient $\vartheta = N\varphi_\theta^2/M$ as an estimate of the proportion of phenotypic variation explained by tagged pairwise interaction effects, and 1 is the intercept meant to model the misestimation due to uncontrolled confounding effects (e.g. cryptic relatedness and population stratification). Similar to the original LDSC formulation, an intercept greater than one means significant bias. Note that the simplification for many of the terms above such as $(1 - H^2)/N \approx 1/N$ results from our assumption that the number of individuals in our study is large. For example, the sample sizes for each biobank-scale study considered in the analyses of this manuscript are at least on the order of $N \geq 10^4$ observations (see *Supplementary file 5*). Altogether, we can jointly express *Equation 27* in multivariate form as

$$\mathbb{E}[\chi^2] \approx \ell\tau + f\vartheta + 1 \qquad (28)$$

where $\chi^2 = (\chi_1^2, \ldots, \chi_J^2)$ is a $J$-dimensional vector of chi-square summary statistics, and $\ell = (\ell_1, \ldots, \ell_J)$ and $f = (f_1, \ldots, f_J)$ are $J$-dimensional vectors of additive and *cis*-interaction LD scores, respectively. It is important to note that, while $\chi^2$ must be recomputed for each trait of interest, both vectors $\ell$ and $f$ only need to be constructed once per reference panel or individual-level genotypes (see next section for efficient computational strategies).

To identify summary statistics that have significant tagged interaction effects, we test the null hypothesis $H_0 : \vartheta = 0$. The i-LDSC software package implements the same model fitting strategy as LDSC. Here, we use weighted least squares to fit the joint regression in *Equation 28* such that

$$\widehat{\vartheta} = (f^{\mathsf{T}}\Psi f)^{-1}f^{\mathsf{T}}\Psi\chi^2, \qquad \psi_{jj} = \left[\ell_j\widehat{\tau} + f_j\widehat{\vartheta} + 1\right]^{-2} \qquad (29)$$

where $\Psi$ is a $J \times J$ diagonal weight matrix with nonzero elements set to values inversely proportional to the conditional variance $\mathbb{V}[\chi_j^2 | \ell_j, f_j] = \psi_{jj}^{-1}$ to adjust for both heteroscedasticity and over-estimation of the summary statistics for each SNP (*Bulik-Sullivan et al., 2015b*). Standard errors for each coefficient estimate are derived via a jackknife over blocks of SNPs in the data (*Finucane et al., 2015*), and we then use those standard errors to derive p-values with a two-sided test (i.e. testing the alternative hypothesis $H_A : \vartheta \neq 0$). It is worth noting that the block-jackknife approach tends to be conservative and yield larger standard errors for hypothesis testing (*Efron, 1982*). As an alternative, we could first run i-LDSC using the block-jackknife procedure over all traits in a study and then use the average of the standard errors to calculate the statistical significance of coefficient estimates; but we do not explore this strategy here and leave that for future work. The quantitative genetics expression for the additive variance $\sigma_A^2$ in *Equation 22* is important because it represents the theoretical upper bound on the proportion of phenotypic variance that can be explained from GWAS summary statistics via i-LDSC. Using this relationship, we can write the following (approximate) inequality

$$\widehat{\tau} + \widehat{\vartheta} \lesssim \sum_{j=1}^{J} 2p_j(1 - p_j)\left[\beta_j + 2\sum_{k \neq j}^{J} p_k\theta_{jk}\right]^2 = \sigma_A^2. \qquad (30)$$

For all analyses in this paper, we estimate proportion of phenotypic variance explained by genetic effects using a sum of the coefficients $\widehat{\tau} + \widehat{\vartheta}$ (i.e. the estimated additive component plus the additional genetic variance explained by the tagged pairwise interaction effects).

## Efficient computation of *cis*-interaction LD scores

In practice, *cis*-interaction LD scores in i-LDSC can be computed efficiently through realizing two key opportunities for optimization. First, given $J$ SNPs, the full matrix of genome-wide interaction effects $\mathbf{W}$ contains on the order of $J(J-1)/2$ total pairwise interactions. However, to compute the *cis*-interaction score for each SNP, we simply can replace the full $\mathbf{W}$ matrix with a subsetted matrix $\mathbf{W}_j$ which includes only interactions involving the $j$-th SNP. Analogous to the original LDSC formulation (*Bulik-Sullivan et al., 2015b*), we consider only interactive SNPs within a *cis*-window proximal to the focal $j$-th SNP for which we are computing the i-LDSC score. In the original LDSC model, this is based on the observation that LD decays outside of a window of 1 centimorgan (cM) (*Bulik-Sullivan et al., 2015b*); therefore, SNPs outside the 1 cM window centered on the $j$-th SNP $j$ will not significantly contribute to its LD score. The second opportunity for optimization comes from the fact that the matrix of interaction effects for any focal SNP, $\mathbf{W}_j$, does not need to be explicitly generated. Referencing *Equation 24*, the i-LDSC scores are defined as $\mathbf{x}_j^{\mathsf{T}}\mathbf{W}_j\mathbf{W}_j^{\mathsf{T}}\mathbf{x}_j/N^2$. This can be re-written as $\mathbf{x}_j^{\mathsf{T}}(\mathbf{D}_j\mathbf{X}^{(j)})(\mathbf{D}_j\mathbf{X}^{(j)})^{\mathsf{T}}\mathbf{x}_j$, where $\mathbf{D}_j = \mathrm{diag}(\mathbf{x}_j)$ is a diagonal matrix with the $j$-th genotype as its nonzero elements (*Crawford et al., 2017*) and $\mathbf{X}^{(j)}$ denotes the subset SNPs within a *cis*-window proximal to the focal $j$-th SNP. This means that the i-LDSC score for the $j$-th SNP can be simply computed as the following

$$f_j \approx \tfrac{1}{N^2}(\mathbf{x}_j^{\mathsf{T}})^2\mathbf{X}^{(j)}\mathbf{X}^{(j)\mathsf{T}}(\mathbf{x}_j)^2. \tag{31}$$

With these simplifications, the computational complexity of generating i-LDSC scores reduces to that of computing LD scores — modulo a vector-by-vector Hadamard product which, for each SNP, is constant factor of $N$ (i.e. the number of genotyped individuals).

## Coefficient estimates as determined by *cis*-interaction window size

When computing *cis*-interaction LD scores, the most important de*cis*ion is choosing the number of interacting SNPs to include in $\mathbf{X}^{(j)}$ (or equivalently $\mathbf{W}_j$ for each $j$-th focal SNP in the calculation of $f_j$ in *Equation 31*). The i-LDSC framework considers different estimating windows to account for our lack of a priori knowledge about the 'correct' non-additive genetic architecture of traits. Theoretically, one could follow previous work *Guan and Stephens, 2011*; *Carbonetto and Stephens, 2012*; *Zhou et al., 2013*; *Zhu and Stephens, 2017*; *Zhu and Stephens, 2018*; *Demetci et al., 2021* by considering an $L$-valued grid of possible SNP interaction window sizes. After fitting a series of i-LDSC regressions with *cis*-interaction LD scores $f^{(l)}$ generated under the $L$-different window sizes, we could compute normalized importance weights using their maximized likelihoods via the following

$$\pi^{(l)} = \frac{\mathcal{L}\left(\ell, f^{(l)}; \widehat{\beta}\right)}{\sum_{l'}\mathcal{L}\left(\ell, f^{(l')}; \widehat{\beta}\right)}, \qquad \sum_{l=1}^{L}\pi^{(l)} = 1. \tag{32}$$

As a final step in the model fitting procedure, we could then compute averaged estimates of the coefficients $\tau$ and $\vartheta$ by marginalizing (or averaging) over the $L$-different grid combinations of estimating windows

$$\widehat{\tau} = \sum_{l=1}^{L}\pi^{(l)}\widehat{\tau}^{(l)}, \qquad \widehat{\vartheta} = \sum_{l=1}^{L}\pi^{(l)}\widehat{\vartheta}^{(l)}. \tag{33}$$

This final step can be viewed as an analogy to model averaging where marginal estimates are computed via a weighted average using the importance weights (*Hoeting et al., 1999*). In the current study, we explore the utility of *cis*-interaction LD scores generated with different window sizes ± 5, ± 10, ± 25, and ± 50 SNPs around each $j$-th focal SNP. In practice, we find that *cis*-interaction LD scores that are calculated using larger windows lead to the most robust estimates of heritability while also not over representing the total phenotypic variation explained by tagged non-additive genetic effects (see *Figure 3—figure supplement 1*). Therefore, unless otherwise stated, we use *cis*-interaction LD scores calculated with a ± 50 SNP interaction window for all simulations and real data analyses conducted in this work. For a direct comparison between choosing a single window size versus the model averaging strategy described above, see *Supplementary files 1 and 2*.

## Relationship between minor allele frequency and effect size

The LDSC software computes LD scores using annotations over equally spaced minor allele frequency (MAF) bins. These annotations enable the per trait relationship between the MAF and the effect size of each variant in the genome to vary based on the discrete category (or MAF bin) it is placed into. This additional flexibility is intended to help LDSC be more robust when estimating heritability. The relationship between MAF and effect size is already implicitly encoded in the LDSC formulation since we assume genotypes are normalized. When normalizing by the variance of each SNP (or equivalently its MAF), we make the assumption that rare variants inherently have larger effect sizes. There exists a true functional relationship between MAF and effect size which is likely to be somewhere between the two extremes of (*i*) normalizing each SNP by its MAF and (*ii*) allowing the variance per SNP to be dictated by its MAF.

Recent approaches have proposed using a single parameter $\alpha$ to better represent the nonlinear relationship between MAF and variant effect size. The main idea is that this $\alpha$ not only provides the same additional flexibility to LDSC as the MAF-based discrete annotations, but it also empirically yields even more precise heritability estimates (*Zabad et al., 2021*). Namely, we use

$$\ell_j(c) := \sum_k L_{jk}(\alpha)a_c(k), \qquad L_{jk}(\alpha) = r_{jk}^2 \mathbb{V}[\mathbf{x}_k]^{1-\alpha} \tag{34}$$

where $a_c(k)$ is the annotation value for the $c$-th categorical bin. The α parameter is unknown in practice and needs to be estimated for any given trait. While standard ranges for α can be used for heritability estimates, we use a restricted maximum likelihood (REML) based method which was recently developed (*Schoech et al., 2019*).

In the i-LDSC software, we use this α construction to handle the relationship between MAF and variant effect size for two specific reasons. First, by constructing the LD scores using α, we more accurately capture the variation in chi-square test statistics due to additive effects (*Zabad et al., 2021*). Second, we note that there is correlation between MAF and (*i*) LD scores, (*ii*) *cis*-interaction LD scores, and (*iii*) trait architecture. To that end, if we do not properly condition on MAF, there becomes additional bias, and we may falsely attribute some amount of variation in the chi-square test statistics to LD or the tagged interaction effects. Therefore, in our formulation, we include an α term on the LD scores to condition on this effect. We demonstrate in simulations that this removes the bias introduced by the relationship between MAF and trait architecture, and it mitigates potential inflation of type I error rates in the i-LDSC test.

## Estimation of allele frequency parameters

In the main text, we analyzed 25 complex traits in both the UK Biobank and BioBank Japan data sets. In order to account for minor allele frequency (MAF) dependent trait architecture, we calculated $\alpha$ values for each trait that had not been analyzed by previous studies (*Schoech et al., 2019*). The α estimates for each of the 25 traits analyzed in this study are shown in *Supplementary file 4*. Intuitively, $\alpha$ parameterizes the weighting of the effects of each individual variant given its frequency in the study cohort and can take on values in the range of [–1,0]. More negative values of $\alpha$ indicate that lower frequency variants contribute more to the observed variation in a trait of interest, whereas values of α closer to zero indicate that common variants contribute a greater amount of variation to observed trait values.

We took α values for 11 traits (again see *Supplementary file 4*) that had previously been calculated from Schoech et al. For the remaining 14 traits analyzed in this study, we followed the estimation protocol described in the same manuscript. Specifically, using the variants passing the quality control step in our pipeline for 25,000 randomly selected individuals in the UK Biobank cohort, we constructed MAF-dependent genetic relatedness matrices for values of $\alpha = \{-1, -0.95, -0.9, \dots, 0\}$ using the GRM-MAF-LD software (*Schoech, 2018*). We then used the GCTA software (*Yang et al., 2011*) to obtain heritability and likelihood estimates using REML for each $\alpha$-trait pairing. We then fit a trait-specific profile likelihood across the range of α values and estimate the maximum likelihood value of $\alpha$ using a natural cubic spline.

## Simulation studies

We used a simulation scheme to generate synthetic quantitative traits and SNP-level summary statistics under multiple genetic architectures using real genome-wide data from individuals of self-identified

European ancestry in the UK Biobank. Here, we consider phenotypes that have some combination of additive effects, *cis*-acting interactions, and a third source of genetic variance stemming from either gene-by-environment (G×E) or gene-by-ancestry (G×Ancestry) effects. For each scenario, we select some set of SNPs to be causal and assume that complex traits are generated via the following general linear model

$$\mathbf{y} = \mathbf{X}\beta + \mathbf{W}\theta + \mathbf{Z}\gamma + \varepsilon, \qquad \varepsilon \sim \mathcal{N}(\mathbf{0}, \delta^2\mathbf{I}), \tag{35}$$

where $\mathbf{y}$ is an $N$-dimensional vector containing all the phenotypes; $\mathbf{X}$ is an $N \times J$ matrix of genotypes encoded as 0, 1, or 2 copies of a reference allele; $\beta$ is a $J$-dimensional vector of additive effect sizes for each SNP; $\mathbf{W}$ is an $N \times M$ matrix which holds all pairwise interactions between the randomly selected subset of the interacting SNPs with corresponding effects $\theta$ is an $N \times K$ matrix of either G×E or G×Ancestry interactions with coefficients $\gamma$; and $\varepsilon$ is an $N$-dimensional vector of environmental noise. The phenotypic variation is assumed to be $\mathbb{V}[\mathbf{y}] = 1$. All additive and interaction effect sizes for SNPs are randomly drawn from independent standard Gaussian distributions and then rescaled so that they explain a fixed proportion of the phenotypic variance $\mathbb{V}[\mathbf{X}\beta] + \mathbb{V}[\mathbf{W}\theta] + \mathbb{V}[\mathbf{Z}\gamma] = H^2$. Note that we do not assume any specific correlation structure between the effect sizes $\beta$, $\theta$, and $\gamma$. We then rescale the random error term such that $\mathbb{V}[\varepsilon] = (1 - H^2)$. In the main text, we compare the traditional LDSC to its direct extension in i-LDSC. For each method, GWAS summary statistics are computed by fitting a single-SNP univariate linear model via least squares where $\widehat{\beta}_j = (\mathbf{x}_j^\mathsf{T}\mathbf{x}_j)^{-1}\mathbf{x}_j^\mathsf{T}\mathbf{y}$ for every $j = 1, \ldots, J$ SNP in the data. These effect size estimates are used to derive the chi-square test statistics $\chi_j^2 = N\widehat{\beta}_j^2$. We implement both LDSC and i-LDSC with the LD matrix $\mathbf{R} = \mathbf{X}^\mathsf{T}\mathbf{X}/N$ and the *cis*-interaction correlation matrix $\mathbf{V} = \mathbf{X}^\mathsf{T}\mathbf{W}/N$ being computed using a reference panel of 489 individuals from the European superpopulation (EUR) of the 1000 Genomes Project (https://mathgen.stats.ox.ac.uk/impute/data_download_1000G_phase1_integrated.html). The resulting matrices $\mathbf{R}$ and $\mathbf{V}$ are used to compute the additive and *cis*-interaction LD scores, respectively.

## Polygenic simulations with *cis*-interactions

In our first set of simulations, we consider phenotypes with polygenic architectures that are made up of only additive and *cis*-acting SNP-by-SNP interactions. Here, we begin by assuming that every SNP in the genome has at least a small additive effect on the traits of interest. Next, when generating synthetic traits, we assume that the additive effects make up $\rho\%$ of the heritability while the pairwise interactions make up the remaining $(1 - \rho)\%$. Alternatively, the proportion of the heritability explained by additivity is said to be $\mathbb{V}[\mathbf{X}\beta] = \rho H^2$, while the proportion detailed by interactions is given as $\mathbb{V}[\mathbf{W}\theta] = (1 - \rho)H^2$. The setting of $\rho = 1$ represents the limiting null case for i-LDSC where the variation of a trait is driven by solely additive effects. Here, we use the same simulation strategy used in Crawford et al. where we divide the causal *cis*-interaction variants into two groups. One may view the SNPs in group #1 as being the 'hubs' of an interaction map. SNPs in group #2 are selected to be variants within some kilobase (kb) window around each SNP in group #1. Given different parameters for the generative model in *Equation 35*, we simulate data mirroring a wide range of genetic architectures by toggling the following parameters:

- heritability: $H^2 = 0.3$ and 0.6;
- proportion of phenotypic variation that is generated by additive effects: $\rho = 0.5$, 0.8, and 1;
- percentage of SNPs selected to be in group #1: 1% (sparse), 5%, and 10% (polygenic);
- genomic window used to assign SNPs to group #2: ± 10 and ± 100 kilobase (kb);
- allele frequency parameter: $\alpha = -1, -0.5$, and 0.

All figures and tables show the mean performances (and standard errors) across 100 simulated replicates.

## Polygenic simulations with gene-by-environmental effects

In our second set of simulations, we continue to consider phenotypes with polygenic architectures that are made up of only additive and *cis*-acting SNP-by-SNP interactions; however, now we also consider each trait to have contributions stemming from nonzero G×E effects. Here, both the additive and *cis*-interaction effects are simulated in the same way as previously described where, for the two groups of interacting variants, 10% of SNPs were selected to be in group #1 and we chose ±10 kb windows

to assign SNPs to group #2. To create G×E effects, we follow a simulation strategy implemented by Zhu et al. and split our sample population in half to emulate two subsets of individuals coming from different environments. We randomly draw the effect sizes for the first environment from a standard Gaussian distribution which we denote as $\gamma_1$. We then selected an amplification coefficient $w$ and set the effect sizes of the G×E interactions in the second environment to be a scaled version of the first environment effects where $\gamma_2 = w\gamma_1$. In this paper, we generate traits with heritability $H^2 = \{0.3, 0.6\}$ and amplification coefficients set to $w = [1.1, 1.2, \ldots, 2]$. For the first set of simulations, we hold the proportion of phenotypic variation explained by the different genetic components constant by fixing:

- $H^2 = 0.3$: $\mathbb{V}[\mathbf{X}\beta] = 0.15$; $\mathbb{V}[\mathbf{W}\theta] = 0.075$; and $\mathbb{V}[\mathbf{Z}\gamma] = 0.075$;
- $H^2 = 0.6$: $\mathbb{V}[\mathbf{X}\beta] = 0.3$; $\mathbb{V}[\mathbf{W}\theta] = 0.15$; and $\mathbb{V}[\mathbf{Z}\gamma] = 0.15$;

where $\mathbf{Z} = [\mathbf{X}_1, \mathbf{X}_2]$ is the set of genotypes split according to environment and $\gamma = [\gamma_1, \gamma_2]$. To test the sensitivity of the *cis*-interaction LD scores to other sources of non-additive variation, we also repeated the same simulations where there were only additive and G×E effects contributing equally to trait architecture:

- $H^2 = 0.3$: $\mathbb{V}[\mathbf{X}\beta] = 0.15$; $\mathbb{V}[\mathbf{W}\theta] = 0$; and $\mathbb{V}[\mathbf{Z}\gamma] = 0.15$;
- $H^2 = 0.6$: $\mathbb{V}[\mathbf{X}\beta] = 0.3$; $\mathbb{V}[\mathbf{W}\theta] = 0$; and $\mathbb{V}[\mathbf{Z}\gamma] = 0.3$.

Again all figures show the mean performances (and standard errors) across 100 simulated replicates.

## Polygenic simulations with gene-by-ancestry effects

In our third set of simulations, we consider phenotypes with polygenic architectures that are made up of additive, *cis*-interactions, and G×Ancestry effects. Here, we follow Sohail et al. and first run a matrix decomposition on the individual-level genotype matrix $\mathbf{X} = \mathbf{U}\mathbf{Q}^{\mathsf{T}}$ where $\mathbf{U}$ is a unitary $N \times K$ score matrix, $\mathbf{Q}$ is a $K \times J$ loadings matrix, and $K$ represents the number of (predetermined) principal components (PCs). To generate G×Ancestry interactions, we then create the matrix $\mathbf{Z}_k = \mathbf{X}\mathbf{q}_k$ where $\mathbf{q}_k$ is a $J$-dimensional vector of SNP loadings for the $k$-th principal component. In this paper, we generate traits with heritability $H^2 = \{0.3, 0.6\}$ and interaction effects taken over $k = 1, \ldots, 10$ principal components. For the first set of simulations, we hold the proportion of phenotypic variation explained by the different genetic components constant by fixing:

- $H^2 = 0.3$: $\mathbb{V}[\mathbf{X}\beta] = 0.15$; $\mathbb{V}[\mathbf{W}\theta] = 0.075$; and $\mathbb{V}[\mathbf{Z}\gamma] = 0.075$;
- $H^2 = 0.6$: $\mathbb{V}[\mathbf{X}\beta] = 0.3$; $\mathbb{V}[\mathbf{W}\theta] = 0.15$; and $\mathbb{V}[\mathbf{Z}\gamma] = 0.15$;

To test the sensitivity of the *cis*-interaction LD scores to other sources of non-additive variation, we also repeated the same simulations where there were only additive and G×E effects contributing equally to trait architecture:

- $H^2 = 0.3$: $\mathbb{V}[\mathbf{X}\beta] = 0.15$; $\mathbb{V}[\mathbf{W}\theta] = 0$; and $\mathbb{V}[\mathbf{Z}\gamma] = 0.15$;
- $H^2 = 0.6$: $\mathbb{V}[\mathbf{X}\beta] = 0.3$; $\mathbb{V}[\mathbf{W}\theta] = 0$; and $\mathbb{V}[\mathbf{Z}\gamma] = 0.3$.

Note that, for each case, we generate summary statistics in two ways: (*i*) including the top 10 PCs as covariates in the marginal linear model to correct for population structure and (*ii*) not correcting for any population structure. Again all figures show the mean performances (and standard errors) across 100 simulated replicates.

## Sparse simulation study design with additive effects

In this set of simulations, we consider phenotypes with sparse architectures (*Zhou et al., 2013*). Here, traits were simulated with solely additive effects such that $\mathbb{V}[\mathbf{X}\beta] = H^2$, but this time only variants with the top or bottom $\{1, 5, 10, 25, 50, 100\}$ percentile of LD scores were given nonzero coefficients (a similar simulation approach was also previously implemented in both *Bulik-Sullivan et al., 2015b* and *Lee et al., 2018*). We once again generate traits with heritability $H^2 = \{0.3, 0.6\}$. We also want to note that, in each of these specific analyses, synthetic trait architectures were generated using all UK Biobank genotyped variants that passed initial preprocessing and quality control (see next section). Since not all of these SNPs are HapMap3 SNPs, some variants were omitted from the LDSC and i-LDSC regression. Overall, as shown in the main text with results taken over 100 replicates, breaking the assumed relationship between LD scores and chi-squared statistics (i.e. that they are

generally positively correlated) led to unbounded estimates of heritability in all but the (more polygenic) scenario when 100% of SNPs contributed to phenotypic variation.

## Polygenic simulations with unobserved additive effects

In this next set of simulations, we consider another extension of the polygenic case where a portion of the variants with only additive genetic effects are not observed due ascertainment or other quality control procedures. It was found in *Hemani et al., 2014*. that an initial set of signals pointing towards evidence of genetic interactions were actually better explained using linear models of unobserved variants in the same haplotype. Here, we test whether the i-LDSC framework is prone to overestimate the non-additive genetic variance when additive effects in the same haplotype are not included in the model. In each simulation, we generated haplotypes that each contain 5000 variants. Next, we select either a single causal variant with only an additive effect or a set of ten causal variants with only additive effects — each having an MAF that is randomly selected between: (*i*) (0.01, 0.1), (*ii*) (0.1, 0.2), (*iii*) (0.2, 0.3), (*iv*) (0.3, 0.4), and (*v*) (0.4, 0.5). The corresponding additive effect size for each causal variant across the haplotype is simulated inversely proportional with its MAF. For this analysis, we measure the difference between i-LDSC coefficient estimates when every variant is included in the model versus when the haplotype causal variants are omitted for two different trait architectures with broad-sense heritability set to $H^2 = 0.3$ and 0.6. Differences in the component estimates between the observed and unobserved single additive variant models are shown in *Figure 3—figure supplement 9A and B*. Similar estimates when the larger number of ten additive variants are unobserved in each haplotype are shown in *Figure 3—figure supplement 9C and D*. If i-LDSC was prone to overestimating the non-additive effects, then the omission of the variants with only significant additive effects would lead to increased estimates of $\tau$ and $\vartheta$. However, across a range of generative broad-sense heritabilities and haplotype architectures we observe that estimates of $\tau$ and $\vartheta$ are robust. Intuitively, this is likely due to the fact that these simulations were done under polygenic trait architectures where, as a result, the omission of a few causal variants with small marginal effect sizes has little impact on the ability to estimate genetic variance.

## Polygenic simulations with unobserved interaction effects

In this set of simulations, we extend the polygenic case to a setting where a portion of the variants involved in genetic interactions are unobserved. Similar to the case with unobserved additive effects, the purpose of these simulations is to assess whether the i-LDSC framework is prone to false discovery of non-additive genetic variance when causal interacting SNPs are not included during the estimation of GWAS summary statistics. In each simulation, we generated haplotypes that each contain 5000 variants. Traits were simulated using the generative model in *Equation (35)* with both additive and interaction effects such that $\mathbb{V}[\mathbf{X}\beta] + \mathbb{V}[\mathbf{W}\theta] = H^2$. Here, every SNP in the genome had at least a small additive effect with a corresponding effect size that was drawn to be inversely proportional to its MAF. Only 1% or 5% of variants within each haplotype had causal non-zero interaction effects. However, when running i-LDSC, only a percentage of the interacting SNPs {1%, 5%, 10%, 25% or 50%} were included in the estimation of $\widehat{\vartheta}$. We once again generate traits with heritability $H^2 = \{0.3, 0.6\}$ such that the proportion of genetic variance explained by additive effects was equal to $\rho = \{0.5, 0.8\}$. As with the other simulation scenarios, all synthetic traits were generated using UK Biobank genotyped variants that passed initial preprocessing and quality control (see next section). Since not all of these SNPs are HapMap3 SNPs, some variants were omitted from the i-LDSC regression analyses. Overall, as discussed in the main text with results taken over 100 replicates, i-LDSC underestimated values of $\widehat{\vartheta}$ when there were unobserved interacting variants (see *Figure 3—figure supplements 10 and 11*). As expected, estimates of the additive variance component $\widehat{\tau}$, on the other hand, were not affected.

## Polygenic simulations with correlated additive and interaction effects

In our last set of simulations, we sought out to better understand how the relationship between the additive ($\beta$) and interaction ($\theta$) coefficients in the generative model of complex traits could potentially bias the additive and non-additive variance component estimates in LDSC and i-LDSC. To that end, we performed a set of simulations where we varied the correlation between the set of effects. Specifically, we first drew a set of additive effect sizes for each variant using the MAF-dependent procedure described above (i.e. $\alpha = -1$). We next selected a subset of the causal variants to be in

cis-interactions. Here, we set the interaction effect sizes to covary with the additive effect size vector in two different ways. In the first, we simply drew the additive and interaction effect sizes from a multivariate normal such that their correlation was equal to $r = \{-1, -0.8, -0.6, \dots, 0.6, 0.8, 1\}$ (see *Figure 3—figure supplement 12*). In the second, we simply amplified the interaction effects to be a linear function $\theta = \beta \times q$ (*Figure 3—figure supplement 13A and C*) or a squared function $\theta = \beta^{2q}$ (*Figure 3—figure supplement 13B and D*) of the additive effects where $q = \{0.1, 0.2, \dots, 0.9, 1\}$. While testing 100 replicates for each value of $q$, we observed that the mean estimate of genetic variance had a slight upward bias as the correlation between the additive and interaction effect sizes in the generative model increased; however, the distribution of these bias estimates covered zero in the first and third quartiles of all results. We evaluated this behavior for multiple broad-sense heritability levels $H^2$ = 0.3 and 0.6.

## Preprocessing for the UK Biobank and BioBank Japan

In order to apply the i-LDSC framework to 25 continuous traits the UK Biobank (*Bycroft et al., 2018*), we first downloaded genotype data for 488,377 individuals in the UK Biobank using the ukbgene tool (https://biobank.ctsu.ox.ac.uk/crystal/download.cgi) and converted the genotypes using the provided ukbconv tool (https://biobank.ctsu.ox.ac.uk/crystal/refer.cgi?id=149660). Phenotype data for the 25 continuous traits were also downloaded for those same individuals using the ukbgene tool. Individuals identified by the UK Biobank as having high heterozygosity, excessive relatedness, or aneuploidy were removed (1,550 individuals). After separating individuals into self-identified ancestral cohorts using data field 21000, unrelated individuals were selected by randomly choosing an individual from each pair of related individuals. This resulted in $N = 349,469$ white British individuals to be included in our analysis. We downloaded imputed SNP data from the UK Biobank for all remaining individuals and removed SNPs with an information score below 0.8. Information scores for each SNP are provided by the UK Biobank (http://biobank.ctsu.ox.ac.uk/crystal/refer.cgi?id=1967).

Quality control for the remaining genotyped and imputed variants was then performed on each cohort separately using the following steps. All structural variants were first removed, leaving only single nucleotide polymorphisms (SNPs) in the genotype data. Next, all AT/CG SNPs were removed to avoid possible confounding due to sequencing errors. Then, SNPs with minor allele frequency less than 1% were removed using the PLINK 2.0 (*Chang et al., 2015*) command `--maf` 0.01. We then removed all SNPs found to be out of Hardy-Weinberg equilibrium, using the PLINK `--hwe` 0.000001 flag to remove all SNPs with a Fisher's exact test p-value $> 10^{-6}$. Finally, all SNPs with missingness greater than 1% were removed using the PLINK `--mind` 0.01 flag.

We then performed a genome-wide association study (GWAS) for each trait in the UK Biobank on the remaining 8,981,412 SNPs. SNP-level GWAS effect sizes were calculated using PLINK and the `--glm` flag (*Chang et al., 2015*). Age, sex, and the first 20 principal components were included as covariates for all traits analyzed (*Sohail et al., 2019*). Principal component analysis was performed using FlashPCA 2.0 (*Abraham et al., 2017*) on a set of independent markers derived separately for each ancestry cohort using the PLINK command `--indep-pairwise` 100 10 0.1. Using the parameters `--indep-pairwise` removes all SNPs that have a pairwise correlation above 0.1 within a 100 SNP window, then slides forward in increments of ten SNPs genome-wide.

In order to analyze data from BioBank Japan, we downloaded publicly available GWAS summary statistics for the 25 traits listed in *Supplementary file 5* from https://pheweb.jp/downloads. Summary statistics used age, sex, and the first ten principal components as confounders in the initial GWAS study. We then used individuals from the East Asian (EAS) superpopulation from the 1000 Genomes Project Phase 3 to calculate paired LDSC and i-LDSC scores from a reference panel. We pruned the reference panel using the PLINK command `--indep-pairwise` 100 10 0.5 to limit the computational time of calculating scores (*Chang et al., 2015*). This resulted in reference scores for 1,164,666 SNPs that are included on the i-LDSC GitHub repository (https://github.com/lcrawlab/i-LDSC). Using summary statistics from BioBank Japan, with scores calculated from the EAS population in the 1000 Genomes, we obtained i-LDSC heritability estimates for each of the 25 traits.

## Acknowledgements

We thank Jeffrey P Spence, Roshni Patel, Matthew Aguirre, Mineto Ota, and our anonymous referees for insightful comments on an earlier version of this manuscript as well as the Harpak, Ramachandran,

and Crawford Labs for helpful discussions. This research was conducted in part using computational resources and services at the Center for Computation and Visualization at Brown University. This research was also conducted using the UK Biobank Resource under Application Numbers 14649 (LC) and 22419 (SR). SP Smith and D Udwin were trainees supported under the Brown University Predoctoral Training Program in Biological Data Science (NIH T32 GM128596). SP Smith was also supported by NIH RF1AG073593. SP Smith and A Harpak were also supported by NIH R35 GM151108 to A Harpak. G Darnell was supported by NSF Grant No. DMS-1439786 while in residence at the Institute for Computational and Experimental Research in Mathematics (ICERM) in Providence, RI. This research was supported in part by an Alfred P Sloan Research Fellowship and a David & Lucile Packard Fellowship for Science and Engineering awarded to L Crawford. This research was also partly supported by US National Institutes of Health (NIH) grant R01 GM118652, NIH grant R35 GM139628, and National Science Foundation (NSF) CAREER award DBI-1452622 to S Ramachandran. Any opinions, findings, and conclusions or recommendations expressed in this material are those of the author(s) and do not necessarily reflect the views of any of the funders.

## Additional information

### Funding

| Funder | Grant reference number | Author |
|---|---|---|
| National Institutes of Health | T32 GM128596 | Samuel Pattillo Smith Dana Udwin |
| National Institutes of Health | RF1 AG073593 | Samuel Pattillo Smith |
| National Institutes of Health | R35 GM151108 | Samuel Pattillo Smith Arbel Harpak |
| National Science Foundation | DMS-1439786 | Gregory Darnell |
| Alfred P. Sloan Foundation | | Lorin Crawford |
| David and Lucile Packard Foundation | | Lorin Crawford |
| National Institutes of Health | R01 GM118652 | Sohini Ramachandran |
| National Institutes of Health | R35 GM139628 | Sohini Ramachandran |
| National Science Foundation | DBI-1452622 | Sohini Ramachandran |

The funders had no role in study design, data collection and interpretation, or the decision to submit the work for publication.

### Author contributions

Samuel Pattillo Smith, Conceptualization, Software, Formal analysis, Funding acquisition, Validation, Investigation, Visualization, Methodology, Writing – original draft, Writing – review and editing; Gregory Darnell, Conceptualization, Software, Formal analysis, Validation, Investigation, Methodology, Writing – original draft, Writing – review and editing; Dana Udwin, Formal analysis, Validation, Investigation, Visualization, Writing – review and editing; Julian Stamp, Investigation, Methodology, Writing – review and editing; Arbel Harpak, Formal analysis, Supervision, Funding acquisition, Investigation, Writing – review and editing; Sohini Ramachandran, Conceptualization, Resources, Data curation, Formal analysis, Supervision, Funding acquisition, Investigation, Methodology, Writing – original draft, Project administration, Writing – review and editing; Lorin Crawford, Conceptualization, Resources, Data curation, Software, Formal analysis, Supervision, Funding acquisition, Investigation, Methodology, Writing – original draft, Project administration, Writing – review and editing

**Author ORCIDs**
Samuel Pattillo Smith ⓘ http://orcid.org/0000-0002-6269-0276
Gregory Darnell ⓘ http://orcid.org/0000-0003-0425-940X
Arbel Harpak ⓘ http://orcid.org/0000-0002-3655-748X
Lorin Crawford ⓘ http://orcid.org/0000-0003-0178-8242

**Decision letter and Author response**
Decision letter https://doi.org/10.7554/eLife.90459.sa1
Author response https://doi.org/10.7554/eLife.90459.sa2

## Additional files

**Supplementary files**

• Supplementary file 1. Comparison of LDSC and i-LDSC estimates of the proportion of phenotypic variance explained (PVE) by genetic effects (i.e., estimated heritability) when the true heritability is set to $H^2 = 0.3$ for polygenic traits. Synthetic trait architecture was simulated using real genotype data from individuals of self-identified European ancestry in the UK Biobank. All SNPs were considered to have at least an additive effect (i.e., creating a polygenic trait architecture). Next, we randomly select two groups of interacting variants and divide them into two groups. The group #1 SNPs are chosen to be 10% of the total number of SNPs genome-wide. These interact with the group #2 SNPs which are selected to be variants within a ± 100 kilobase (kb) window around each SNP in group #1. Coefficients for additive and interaction effects were simulated with no minor allele frequency dependency $\alpha = 0$ (see Materials and Methods). Here, we assume a heritability $H^2 = 0.3$ and vary the proportion contributed by additive effects with $\rho = \{0.2, 0.4, 0.6, 0.8\}$. We run i-LDSC while computing the *cis*-interaction LD scores using different estimating windows of ± 5, ± 10, ± 25, and ± 50 SNPs. The "average" column represents results using model averaging over the different estimating windows (see Materials and Methods). We report the mean estimates of heritability (with standard errors in the parentheses) and use mean absolute error (MAE) to quantify the difference between the two methods. Results are based on 100 simulations per parameter combination. As shown in *Figure 3—figure supplements 3 and 1*, LDSC does not capture the contribution of non-additive genetic effects to trait variation.

• Supplementary file 2. Comparison of LDSC and i-LDSC estimates of the proportion of phenotypic variance explained (PVE) by genetic effects (i.e., estimated heritability) when the true heritability is set to $H^2 = 0.6$. Synthetic trait architecture was simulated using real genotype data from individuals of self-identified European ancestry in the UK Biobank. All SNPs were considered to have at least an additive effect (i.e., creating a polygenic trait architecture). Next, we randomly select two groups of interacting variants and divide them into two interacting groups. The group #1 SNPs are chosen to be 10% of the total number of SNPs genome-wide. These interact with the group #2 SNPs which are selected to be variants within a ± 100 kilobase (kb) window around each SNP in group #1. Coefficients for additive and interaction effects were simulated with no minor allele frequency dependency $\alpha = 0$ (see Materials and Methods). Here, we assume a heritability $H^2 = 0.6$ and vary the proportion contributed by additive effects with $\rho = \{0.2, 0.4, 0.6, 0.8\}$. We run i-LDSC while computing the *cis*-interaction LD scores using different estimating windows of ± 5, ± 10, ± 25, and ± 50 SNPs. The "average" column represents results using model averaging over the different estimating windows (see Materials and Methods). We report the mean estimates of heritability (with standard errors in the parentheses) and use mean absolute error (MAE) to quantify the difference between the two methods. Results are based on 100 simulations per parameter combination. As shown in *Figure 3—figure supplements 3 and 1*, LDSC does not capture the additional contribution of non-additive genetic effects to trait variation.

• Supplementary file 3. Abbreviations used throughout this study for 14 quantitative traits analyzed in this study. The remaining 11 traits analyzed were Basophil count, Cholesterol, Eosinophil count, Height, Hematocrit, Hemoglobin, Lymphocyte count, Monocyte count, Neutrophil count, and Triglyceride levels, respectively. These are not abbreviated in the main text.

• Supplementary file 4. Trait-specific α parameters for each of the 25 traits analyzed. Here, α values are used to weight each variant based on its minor allele frequency to account for frequency dependent architectures in each trait. The * indicates α parameters that were taken directly from Schoech et al. The α parameters for other traits were calculated using the protocol used in that paper. Expansion of trait abbreviations are given in *Supplementary file 3*.

• Supplementary file 5. Number of individuals and total SNPs included in the analysis of each trait in

BioBank Japan.

• MDAR checklist

## Data availability

Source code and tutorials for implementing interaction-LD score regression via the i-LDSC package are written in Python and are publicly available online on GitHub (copy archived at *Crawford and Smith, 2024*). Files of LD scores, cis-interaction LD scores, and GWAS summary statistics used for our analyses of the UK Biobank and BioBank Japan can be downloaded from the Harvard Dataverse. All software for the traditional and stratified LD score regression framework with LDSC and s-LDSC were fit using the default settings, unless otherwise stated in the main text. Source code for these approaches was downloaded from https://github.com/bulik/ldsc (*Bulik-Sullivan et al., 2020*). When applying s-LDSC, we used 97 functional annotations from *Gazal et al., 2017* to estimate heritability. Data from the UK Biobank Resource (*Bycroft et al., 2018*) was made available under Application Numbers 14649 and 22419. Data can be accessed by direct application to the UK Biobank.

The following dataset was generated:

| Author(s) | Year | Dataset title | Dataset URL | Database and Identifier |
|---|---|---|---|---|
| Smith S, Darnell G, Udwin D, Stamp J, Harpak A, Ramachandran S, Crawford L | 2023 | Replication Data for: Discovering non-additive heritability using additive GWAS summary statistics | https://doi.org/10.7910/DVN/W6MA8J | Harvard Dataverse, 10.7910/DVN/W6MA8J |

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
