## [Editor Report]

This study provides a valuable investigation into whether phenotypic variance due to interactions between genetic variants can be measured using genome-wide association summary statistics. The authors present a convincing method, i-LDSC, that uses statistics on the correlations between genotypes at different loci (linkage disequilibrium) to estimate the phenotypic variance explained by both additive genetic effects and pairwise interactions.

---

## [Decision Letter]

**Decision letter after peer review:**

[Editors’ note: the authors submitted for reconsideration following the decision after peer review. What follows is the decision letter after the first round of review.]

Thank you for submitting the paper "Partitioning Tagged Non-Additive Genetic Effects in Summary Statistics Provides Evidence of Pervasive Epistasis in Complex Traits" for consideration by *eLife*. Your article has been reviewed by 2 peer reviewers, and the evaluation has been overseen by a Reviewing Editor and a Senior Editor. The reviewers have opted to remain anonymous.

Comments to the Authors:

We are sorry to say that, after consultation with the reviewers, we have decided that this work will not be considered further for publication by *eLife*. As the manuscript is currently written, it is not clear how the definitions of additive effects and pairwise interaction effects relate to definitions used in quantitative genetics. Further, both reviewers raised concerns about the empirical results comparing LDSC to MELD. We would be willing to consider a resubmission if the authors are able to address all of the reviewer's concerns.

*Reviewer #1 (Recommendations for the authors):*

In this study, Darnell and colleagues propose a new method to quantify the contribution of cis-epistasis (i.e., interactions between nearby genetic variants) to the heritability of complex traits. Their method, MELD, is an extension of the popular LD score regression methodology. The authors perform simulations conditional on real genotypes to assess the consistency of their estimators of SNP-based heritability and apply their methods to 25 complex traits measured in participants of the UK Biobank and Biobank Japan. The authors conclude that unaccounted epistasis biases estimates of (narrow sense) SNP-based heritability.

I have concerns regarding the method and the conclusions of this study.

1. Additive and non-additive effects are not orthogonal in the proposed model. The authors used the generative models in Equations (1) and (2) to define the narrow sense heritability. However, this definition is incorrect because the additive and non-additive components of their model are not orthogonal. A key point to understand is that there is an additive genetic variance even when all genetic effects arise from statistical interactions between loci. This is the core issue discussed in many papers cited by the authors (e.g., Hill, Goddard and Visscher) but this does not seem to be fully grasped in this study. Unfortunately, I think there is an important confusion between what quantitative genetics classically terms as additive genetic variance and what the authors define here. The same distinction applies for dominance variance. Therefore, I don't think that what the authors term as a "bias" is justified here. That being said, it could still be that what MELD estimates (i.e., not the narrow sense heritability) can teach us something interesting about the genetic architecture of complex traits. I'm not just sure what exactly but my next point below may help.

2. Observations are consistent with alternative explanations. The linear relationship between (expected) chi-square statistics and LD scores postulated by the LD score regression methodology holds under a number of assumptions: (i) all variants are causal and (ii) explain the same amount of variance. However, if (i) is violated then the expected chi-square statistic at a given SNP k is an affine function of the squared correlation between SNP k and all causal variants (in the vicinity). As consequence, if causal variants are enriched in low or high LD regions of the genome, then extra (non-linear) terms could be needed to represent the relationship between chi-square and LD. I believe that the real data application of MELD is affected by the non-random distribution of causal variants wrt LD. The authors could test this prediction by sampling causal variants as a function of their LD scores, simulate a trait, and apply MELD on their resulting GWAS summary statistics. Another related explanation could be if causal variants are poorly imputed. This should also create a signal detectable by MELD.

3. Estimates of narrow sense heritability from MELD are biased. Figure S8 shows that MELD estimates of narrow sense heritability are biased. The authors propose a model averaging approach, which is claimed to work. However, I remain sceptical that this is the solution to the problem. I believe that the model is trying to fit a very peculiar genetic architecture where there are many causal variants within narrow windows (e.g., 100 kb) all interacting with each other. While there is evidence of widespread allelic heterogeneity (e.g., many causal variants at a given locus), evidence of widespread local epistasis is clearly lacking.

Other comments.

a) Table S6: Estimates from LDSC are inconsistent with previously published data. For example, the LDSC estimates for height and BMI are 0.815 and 0.506, respectively. These estimates are much larger than REML estimates obtained with WGS data (~0.7 and ~0.3). This cannot be true. The MELD estimates for the two traits are 0.57 and 0.282 respectively. While the latter estimates make more sense, I'd be curious to see what is the part attributed by MELD to non-additivity. Sorry if that was reported somewhere but I could not see those.

b) Line 88 : Previous methods have been developed to estimate dominance variance using an LD score regression framework.

c) Line 111 (and elsewhere): polygenicity is not a confounding factor like genetic relatedness and population stratification.

d)Lines 137-138: I think this is incorrect. GWAS do not make assumptions (i) and (ii).

e) Lines 145 – 148: V * theta is not a bias because marginal effects and joint effects are different. Equation (3) uses the same notation for both. However, even when theta=0, β_hat would never equal β (unless R = Identity matrix, i.e. all SNPs are independent).

f) Lines 179 – 184: "Instead, …". This is evidence that the additive and non-additive components are not orthogonal. I'm expecting non-additive effects to explain variance on top of additive genetic variance.

g) Line 209: "phenotype they were" – I think that "they" should be deleted.

*Reviewer #2 (Recommendations for the authors):*

The authors present an approach for variance component estimation using summary statistics. In particular, they extend the univariate LDSC regression framework to include a parameter capturing local pairwise epistasis, in addition to the usual intercept and additive components, a method they call "MELD". The authors achieve this by computing a second set of LD scores that index the extent to which a given SNP tags SNP-SNP products within a local window (as opposed to tagging individual SNPs in vanilla LDSC regression). They perform a variety of simulations showing that their method is well callibrated under the standard additive LDSC model as well as their proposed generative model including local pairwise epistatic effects. Finally, they apply their method to a variety of phenotypes in the UK and Japan biobanks, identifying substantial putative non-additive genetic variance.

1. In their analyses comparing LDSC and MELD narrow-sense heritability estimates in the UK biobank (UKB), the LDSC heritability estimates are far higher than I've seen elsewhere. For instance, the authors report h2 estimates of 0.815 and 0.506 for height and BMI, respectively. In the Neale lab UKB h2 browser, which used a fairly liberal set of covariates (nealelab.github.io/UKBB_ldsc/h2_browser.html), these same phenotypes have h2 estimates of 0.485 and 0.249. Evans and colleagues report (doi.org/10.1038/s41588-018-0108-x) LDSC h2 estimates of 0.259 and 0.231, also in the UKB. I can only assume an error in the present work has resulted in such large LDSC h2 values. As such, it is hard to meaningfully compare the MELD and LDSC h2 estimates.

2. The authors' simulations only cover a subset of common generative models that I'd like to see before interpreting their findings. Specifically, they don't present simulations under

additive + GxE effectsthe above + local pairwise epistatic effectsadditive + long range pairwise epistasis (e.g., at loci on different chromosomes)the above + local pairwise epistatic effectsadditive + ancestry-by-G effects (e.g., the product of individual genotypes and a genomic PC)the above + local pairwise epistatic effects

which I would like to see both for their proposed method and for LDSC. They do not need to present a method that will perform well under all of these scenarios, only demonstrate how their method performs under other plausible generative models. To the extent that MELD outperforms LDSC under the MELD generative model, it may perform relatively poorly under alternative architectures.

3. It is believed that much of the signal we see in LDSC h2 estimates reflects the effects of unmeasured variants tagged by, e.g., a million or so HapMap3 SNPs. How does MELD perform when one or both of SNPs with epistatic effects aren't directly measured in the provided data?

4. There is an analogy between LDSC regression and Haseman-Elston (HE) regression (doi.org/10.1214/17-AOAS1052). What would the HE regression equivalent of MELD be? Answering this question will better situate MELD in existing methodological literature.

5. It would be helpful to also present the MELD broad-sense h2 estimates whenever MELD and LDSC narrow-sense h2 estimates are compared.

[Editors’ note: further revisions were suggested prior to acceptance, as described below.]

Thank you for submitting your article "Accounting for statistical non-additive interactions enables the recovery of missing heritability from GWAS summary statistics" for consideration by *eLife*. Your article has been reviewed by 2 peer reviewers, and the evaluation has been overseen by a Reviewing Editor and George Perry as the Senior Editor.

While the reviewers and editors appreciate that the manuscript has improved substantially since the initial submission – and that the method may be picking up something interesting – a key point that was made in the initial review was not addressed adequately: the authors have not clarified the relationship between the parameters their method estimates and the traditional decomposition of the genetic variance into additive and non-additive components. The traditional definition of the additive variance is an orthogonal decomposition, where the non-additive component is defined as the component that is orthogonal to the best linear prediction of the genetic values. This is not the case in the authors' method since they require that the genotype matrix and genetic interaction matrix are correlated in order for their method to pick up any 'epistasis' component. However, the authors claim to recover additional variance beyond the additive variance even from additive GWAS summary statistics. Unless the authors can clarify in detail how their parameters and empirical results relate to traditionally defined additive and non-additive variance components, we will not be able to publish the manuscript.

A further point that was raised during the consultation between the editors and reviewers was whether the authors' method is vulnerable to the artefact that led to a retraction of a Nature paper on cis-epistasis affecting gene expression: https://www.nature.com/articles/s41586-021-03766-y. This article was retracted because they found that what appeared to be epistasis was better explained by interaction genotypes better tagging haplotypes with causal variants.

We would require that the authors address the above points in addition to the point raised by reviewer 2 about whether the s-LDSC results should change the interpretation given to the authors' empirical results about the magnitude of epistasis.

*Reviewer #2 (Recommendations for the authors):*

This paper has a lot of strong points, and I commend the authors for the effort and ingenuity expended in tackling the difficult problem of estimating epistatic (non-additive) genetic variance from GWAS summary statistics. The mere possibility of the estimated univariate regression coefficient containing a contribution from epistasis, as represented in the manuscript's Equation~3 and elsewhere, is intriguing in and of itself.

Is i-LDSC Estimating Epistasis?

Perhaps the issue that has given me the most pause is uncertainty over whether the paper's method is really estimating the non-additive genetic variance, as this has been traditionally defined in quantitative genetics with great consequences for the correlations between relatives and evolutionary theory (Fisher, 1930, 1941; Lynch and Walsh, 1998; Burger, 2000; Ewens, 2004).

Let us call the expected phenotypic value of a given multiple-SNP genotype the total genetic value. If we apply least-squares regression to obtain the coefficients of the SNPs in a simple linear model predicting the total genetic values, then the partial regression coefficients are the average effects of gene substitution and the variance in the predicted values resulting from the model is called the additive genetic variance. (This is all theoretical and definitional, not empirical. We do not actually perform this regression.) The variance in the residuals---the differences between the total genetic values and the additive predicted values---is the non-additive genetic variance. Notice that this is an orthogonal decomposition of the variance in total genetic values. Thus, in order for the variance in Wθ to qualify as the non-additive genetic variance, it must be orthogonal to Xβ.

At first, I very much doubted whether this is generally true. And I was not reassured by the authors' reply to Reviewer~1 on this point, which did not seem to show any grasp of the issue at all. But to my surprise I discovered in elementary simulations of Equation 1 above that for mean-centered X1 and X2, (X1β1+X2β2) is uncorrelated with X1X2θ for seemingly arbitrary correlation between X1 and X2. A partition of the outcome's variance between these two components is thus an orthogonal decomposition after all. Furthermore, the result seems general for any number of independent variables and their pairwise products. I am also encouraged by the report that standard and interaction LD Scores are ‘lowly correlated' (line~179), meaning that the standard LDSC slope is scarcely affected by the inclusion of interaction LD Scores in the regression; this behavior is what we should expect from an orthogonal decomposition.

I have therefore come to the view that the additional variance component estimated by i-LDSC has a close correspondence with the epistatic (non-additive) genetic variance after all.

In order to make this point transparent to all readers, however, I think that the authors should put much more effort into placing their work into the traditional framework of the field. It was certainly not intuitive to multiple reviewers that Xβ is orthogonal to Wθ. There are even contrary suggestions. For if (Xβ)⊺Wθ=β⊺X⊺Wθ is to equal zero, we know that we can't get there by X⊺W equaling zero because then the method has nothing to go on (e.g., line~139). We thus have a quadratic form---each term being the weighted product of an average (additive) effect and an interaction coefficient---needing to cancel out to equal zero. I wonder if the authors can put forth a rigorous argument or compelling intuition for why this should be the case.

In the case of two polymorphic sites, quantitative genetics has traditionally partitioned the total genetic variance into the following orthogonal components:

additive genetic variance, σAA2, the numerator of the narrow-sense heritability;dominance genetic variance, σA2;additive-by-additive genetic variance, σAA2;additive-by-dominance genetic variance, σAD2; anddominance-by-dominance genetic variance, σDD2.

See Lynch and Walsh (1998, pp. 88-92) for a thorough numerical example. This decomposition is not arbitrary or trivial, since each component has a distinct coefficient in the correlations between relatives. Is it possible for the authors to relate the variance associated with their Wθ to this traditional decomposition? Besides justifying the work in this paper, the establishment of a relationship can have the possible practical benefit of allowing i-LDSC estimates of non-additive genetic variance to be checked against empirical correlations between relatives. For example, if we know from other methods that σD2 is negligible but that i-LDSC returns a sizable σAA2, we might predict that the parent-offspring correlation should be equal to the sibling correlation; a sizable σD2 would make the sibling correlation higher. Admittedly, however, such an exercise can get rather complicated for the variance contributed by pairs of SNPs that are close together (Lynch and Walsh, 1998, pp. 146-152).

I would also like the authors to clarify whether LDSC consistently overestimates the narrow-sense heritability in the case that pairwise epistasis is present. The figures seem to show this. I have conflicting intuitions here. On the one hand, if GWAS summary statistics can be inflated by the tagging of epistasis, then it seems that LDSC should overestimate heritability (or at least this should be an upwardly biasing factor; other factors may lead the net bias to be different). On the other hand, if standard and interaction LD Scores are lowly correlated, then I feel that the inclusion of interaction LD Score in the regression should not strongly affect the coefficient of the standard LD Score. Relatedly, I find it rather curious that i-LDSC seems increasingly biased as the proportion of genetic variance that is non-additive goes up---but perhaps this is not too important, since such a high ratio of narrow-sense to broad-sense heritability is not realistic.

How Much Epistasis Is i-LDSC Detecting?

I think the proper conclusion to be drawn from the authors' analyses is that statistically significant epistatic (non-additive) genetic variance was not detected. Specifically, I think that the analysis presented in Supplementary Table~S6 should be treated as a main analysis rather than a supplementary one, and the results here show no statistically significant epistasis. Let me explain.

Most serious researchers, I think, treat LDSC as an unreliable estimator of narrow-sense heritability; it typically returns estimates that are too low. Not even the original LDSC paper pressed strongly to use the method for estimating h2 (Bulik-Sullivan et al., 2015). As a practical matter, when researchers are focused on estimating absolute heritability with high accuracy, they usually turn to GCTA/GREML (Evans et al., 2018; Wainschtein et al., 2022).

One reason for low estimates with LDSC is that if SNPs with higher LD Scores are less likely to be causal or to have large effect sizes, then the slope of univariate LDSC will not rise as much as it ‘should’ with increasing LD Score. This was a scenario actually simulated by the authors and displayed in their Supplementary Figure~S15. [Incidentally, the authors might have acknowledged earlier work in this vein. A simulation inducing a negative correlation between LD Scores and χ^2^ statistics was presented by Bulik-Sullivan et al. (2015, Supplementary Figure 7), and the potentially biasing effect of a correlation over SNPs between LD Scores and contributed genetic variance was a major theme of Lee et al. (2018).] A negative correlation between LD Score and contributed variance does seem to hold for a number of reasons, including the fact that regions of the genome with higher recombination rates tend to be more functional. In short, the authors did very well to carry out this simulation and to show in their Supplementary Figure~S15 that this flaw of LDSC in estimating narrow-sense heritability is also a flaw of i-LDSC in estimating broad-sense heritability. But they should have carried the investigation at least one step further, as I will explain below.

Another reason for LDSC being a downwardly biased estimator of heritability is that it is often applied to meta-analyses of different cohorts, where heterogeneity (and possibly major but undetected errors by individual cohorts) lead to attenuation of the overall heritability (de Vlaming et al., 2017).

The optimal case for using LDSC to estimate heritability, then, is incorporating the LD-related annotation introduced by Gazal et al. (2017) into a stratified-LDSC (s-LDSC) analysis of a single large cohort. This is analogous to the calculation of multiple GRMs defined by MAF and LD in the GCTA/GREML papers cited above. When this was done by Gazal et al. (2017, Supplementary Table 8b), the joint impact of the improvements was to increase the estimated narrow-sense heritability of height from 0.216 to 0.534.

All of this has at least a few ramifications for i-LDSC. First, the authors do not consider whether a relationship between their interaction LD Scores and interaction effect sizes might bias their estimates. (This would be on top of any biasing relationship between standard LD Scores and linear effect sizes, as displayed in Supplementary Figure~S15.) I find some kind of statistical relationship over the whole genome, induced perhaps by evolutionary forces, between cis-acting epistasis and interaction LD Scores to be plausible, albeit without intuition regarding the sign of any resulting bias. The authors should investigate this issue or at least mention it as a matter for future study. Second, it might be that the authors are comparing the estimates of broad-sense heritability in Table~1 to the wrong estimates of narrow-sense heritability. Although the estimates did come from single large cohorts, they seem to have been obtained with simple univariate LDSC rather than s-LDSC. When the estimate of h2 obtained with LDSC is too low, some will suspect that the additional variance detected by i-LDSC is simply additive genetic variance missed by the downward bias of LDSC. Consider that the authors' own Supplementary Table~S6 gives s-LDSC heritability estimates that are consistently higher than the LDSC estimates in Table~1. E.g., the estimated h2 of height goes from 0.37 to 0.43. The latter figure cuts quite a bit into the estimated broad-sense heritability of 0.48 obtained with i-LDSC.

Here we come to a critical point. Lines 282--286 are not entirely clear, but I interpret them to mean that the manuscript's Equation~5 was expanded by stratifying ℓ into the components of s-LDSC and this was how the estimates in Supplementary Table~S6 were obtained. If that interpretation is correct, then the scenario of i-LDSC picking up missed additive genetic variance seems rather plausible. At the very least, the increases in broad-sense heritability reported in Supplementary Table~S6 are smaller in magnitude and not statistically significant. Perhaps what this means is that the headline should be a negligible contribution of pairwise epistasis revealed by this novel and ingenious method, analogous to what has been discovered with respect to dominance (Hivert et al., 2021; Pazokitoroudi et al., 2021; Okbay et al., 2022; Palmer et al., 2023).

References

Bulik-Sullivan, B., Loh, P.-R., Finucane, H. K., Ripke, S., Yang, J., Schizophrenia Working Group of the Psychiatric Genomics Consortium, Patterson, N., Daly, M. J., Price, A. L., and Neale, B. M. (2015). LD Score regression distinguishes confounding from polygenicity in genome-wide association studies. Nature Genetics, 47, 291-295.

Burger, R. (2000). The mathematical theory of selection, recombination, and mutation. Wiley.

de Vlaming, R., Okbay, A., Rietveld, C. A., Johannesson, M., Magnusson, P. K. E., Uitterlinden, A. G., van Rooij, F. J. A., Hofman, A., Groe- nen, P. J. F., Thurik, A. R., and Koellinger, P. D. (2017). Meta-GWAS Accuracy and Power (MetaGAP) calculator shows that hiding heritability is partially due to imperfect genetic correlations across studies. PLoS Genetics, 13, e1006495.

Evans, L. M., Tahmasbi, R., Vrieze, S. I., Abecasis, G. R., Das, S., Gazal, S., Bjelland, D. W., de Candia, T. R., Haplotype Reference Consortium, Goddard, M. E., Neale, B. M., Yang, J., Visscher, P. M., and Keller, M. C. (2018). Comparison of methods that use whole genome data to estimate the heritability and genetic architecture of complex traits. Nature Genetics, 50, 737-745.

Ewens, W. J. (2004). Mathematical population genetics I. Theoretical introduction (2nd ed.). Springer.

Fisher, R. A. (1930). The genetical theory of natural selection. Oxford University Press.

Fisher, R. A. (1941). Average excess and average effect of a gene substitution. Annals of Eugenics, 11, 53-63.

Gazal, S., Finucane, H. K., Furlotte, N. A., Loh, P.-R., Palamara, P. F., Liu, X., Schoech, A., Bulik-Sullivan, B., Neale, B. M., Gusev, A., and Price, A. L. (2017). Linkage disequilibrium-dependent architecture of human complex traits shows action of negative selection. Nature Genetics, 49, 1421-1427.

Hivert, V., Sidorenko, J., Rohart, F., Goddard, M. E., Yang, J., Wray, N. R., Yengo, L., and Visscher, P. M. (2021). Estimation of non-additive genetic variance in human complex traits from a large sample of unrelated individuals. American Journal of Human Genetics, 108, 786- 798.

Lee, J. J., McGue, M., Iacono, W. G., and Chow, C. C. (2018). The accuracy of LD Score regression as an estimator of confounding and genetic correlations in genome-wide association studies. Genetic Epidemiology, 42, 783-795.

Lynch, M., and Walsh, B. (1998). Genetics and the analysis of quantitative traits. Sinauer.

Okbay, A., Wu, Y., Wang, N., Jayashankar, H., Bennett, M., Nehzati, S. M., Sidorenko, J., Kweon, H., Goldman, G., Gjorgjieva, T., Jiang, Y., Hicks, B., Tian, C., Hinds, D. A., Ahlskog, R., Magnusson, P. K. E., Oskarsson, S., Hayward, C., Campbell, A., … Young, A. I. (2022). Polygenic prediction of educational attainment within and between families from genome-wide association analyses in 3 million individu- als. Nature Genetics, 54, 437-449.

Palmer, D. S., Zhou, W., Abbott, L., Wigdor, E. M., Baya, N., Churchhouse, C., Seed, C., Poterba, T., King, D., Kanai, M., Bloemendal, A., and Neale, B. M. (2023). Analysis of genetic dominance in the UK Biobank. Science, 379, 1341-1348.

Pazokitoroudi, A., Chiu, A. M., Burch, K. S., Pasaniuc, B., and Sankararaman, S. (2021). Quantifying the contribution of dominance deviation effects to complex trait variation in biobank-scale data. American Journal of Human Genetics, 108, 799-808.

Wainschtein, P., Jain, D., Zheng, Z., TOPMed Anthropometry Working Group, NHLBI Trans-Omics for Precision Medicine Consoritum, Cupples, L. A., Shadyab, A. H., McKnight, B., Shoemaker, B. M., Mitchell, B. D., Psaty, B. M., Kooperberg, C., Liu, C.-T., Albert, C. M., Roden, D., Chasman, D. I., Darbar, D., Lloyd-Jones, D. M., Arnett, D. K.,... Visscher, P. M. (2022). Assessing the contribution of rare variants to complex trait heritability from whole-genome sequence data. Nature Genetics, 54, 263-273.

Abstract: Here and elsewhere, the term cis is used. Parenthetically it is explained that a cis-interaction score captures interactions between a focal variant and nearby variants. Does this mean that i-LDSC only tries to estimate the variance contributed by pairwise interactions between nearby SNPs? That is, does the method say nothing about statistical interactions between SNPs on different chromosomes, say? The paper should make a clear statement about this point somewhere.

line~54--58: I do not understand what is being asserted here. I think it might clear some things up to point out that the developers of LDSC did not rest content with the assumption that every SNP has an effect. E.g., they conducted simulations showing that the intercept and slope remain unbiased estimators, so along as certain other conditions are met, as the proportion of SNPs with a nonzero effect ranged from one all the way down to 10−4 (Bulik-Sullivan et al., 2015, Supplementary Figures 3 and 4).

line~177: I suggest choosing a term other than ‘infinitesimal model’. Some kind of model where every SNP has an effect does not need to hold in order for standard LDSC to work.

line~505: ‘delete-one jackknife’ is confusing, because each block removed in a jackknife replicate contains more than one SNP. Leaving out the ‘delete-one’ is better.

lines~513--518: The justifications for the shortcuts in the computation of interaction LD Scores seem a bit handwavy to me. What about SNPs with which the *j*-th SNP is highly correlated? Is it possible to explain why these do not matter so much?

line~615: Now here is a variance decomposition which, I think, is truly questionable. Why does it make sense to say that the variance attributable to statistical interaction between genetic and non-genetic variables is part of the broad-sense heritability?

Figure~S8: What makes windows of 25 or 50 SNPs for calculating interaction LD Scores the best for getting an estimate of the broad-sense heritability with i-LDSC? Why does a smaller window lead to inflation? This is not fully explained. I am sometimes fine with trying several values of an adjustable setting and picking whatever seems to work the best, but here I am more than a bit curious.

Figure~S16: Is this a duplication of Figure~4D?

[Editors’ note: further revisions were suggested prior to acceptance, as described below.]

Thank you for resubmitting your work entitled "Discovering non-additive heritability using additive GWAS summary statistics" for further consideration by *eLife*. Your revised article has been evaluated by George Perry (Senior Editor) and a Reviewing Editor.

The manuscript has been improved but there are some remaining issues that need to be addressed, as outlined below. If you think you are unable to fully address the reviewer's comments (which pertain to similar issues identified in the last round of reviews), we would advise you not to submit a revised manuscript.

*Reviewer #1 (Recommendations for the authors):*

1. I don't have issues with the authors' derivations but I do still think the authors have not yet successfully related their variance decomposition to the standard variance decomposition in classical quantitative genetics. Here's a simple example demonstrating this

For simplicity, let the generative model be y=X1β1+X2β2+Wθ+e where X1, X2 are independent and standardized and W=standardized(X1Χ2), denoting r1=cov(X1,W)≠0, r2=cov(Χ2,W)≠0, with cov(X1,Χ2)=0 and *e* independent of everything else and y having zero mean and unit variance.

The classical additive genetic variance will the R squared from regressing y on X1 and X2. I.e.,

Vg=maxbcor(y,X1+X2b)2=maxb[cov(y,X1+X2b)21+b2]=maxb[cov(X1β1+X2β2+Wθ+e,X1+X2b)21+b2]=maxb[(β1+θr1+β2b+θr2b)21+b2]

Doing some calculus, we get Vg=(β12+β22)+2θ(β1r1+β2r2)+θ2(r12+r22). So the classical genetic variance is larger than the additive variance component in the authors model (here β12+β22)! It will include the part of the epistatic component tagged by additive SNPs X, which is nonzero despite Xβ and Wθ theta being orthogonal.

I imagine you should be able to figure out the classical additive genetic variance (and epistatic etc) from the authors' model, though this isn't present in their current parametrization. If you can do that, I think it's far easier to interpret.

2. I'm not satisfied with the authors' response to the editors' comment here:

"If these simulations were conducted for sparse architectures, we would have likely seen a greater impact on i-LDSC; although, we want to note that we have already shown the LD score regression framework to be uncalibrated for traits with sparse genetic architectures (see Figure 3 —figure supplement 8)."

The issue here isn't that the results will be biased; LDSC regression is biased for estimating h2 with missing variants but this bias is easy to understand--missing causal variants lead to lower estimates. The question for your method is what happens to epistatic variance component. It's fine if it's attenuated, it's not fine if leads to false positive signal of epistasis. Simulations to clarify this are necessary

*Reviewer #2 (Recommendations for the authors):*

I focus my review in this round on the issues raised by the editor and also on a point of my own accepting the results at face value, how are we to interpret their statistical significance and importance?

Orthogonal decomposition of the genetic variance

I do not think the authors did much to clarify whether their partition of the genetic variance is an orthogonal decomposition in accord with traditional quantitative-genetic theory. Looking at their line 138 and Equation 8, I see that they make use of the assumption that the additive effects and interaction coefficients are independent random variables with zero means. This kind of assumption has become common in statistical genetics (e.g., Yang et al., 2010), and I'm pretty sure the authors think they are simply following the lead of the original LDSC paper. But, as a few papers have pointed out (e.g., de los Campos et al., 2015), this assumption does not make a lot of sense since it is properties of individuals (genotypes, residuals) that are random rather than the properties of SNPs. And even if we are taking an average over different locations in the genome rather than over a population of individuals, why should the mean of the betas be zero? If we always count the derived allele, then the mean might well be negative if there is any tendency for mutations to have a depressing effect on the trait in question. Also, why independent? The recent report that SNPs with larger main effects tend to exhibit more dominance (Palmer et al., 2023) might lead us to expect that pairs of SNPs with larger main effects might show stronger statistical interactions.

Despite the fact that the ‘symbolic convention’ of E(β) = 0 has often been used to justify surprisingly robust tools, I think that it should no longer be retained in statistical genetics. It is confusing, it might be used in the future in an attempt to justify results that are really far off, and it is known already to prove too much. The supposed proofs of GCTA and LDSC seem to show that they are definitely unbiased, but they are in fact biased under conditions that cannot be explained in terms of this genetic-effects-as-random-variables notion.

My strong suggestion is to justify the orthogonality of Xβ and Wθ in some different way. One such way might to prove that Equation 1 in my first review is an orthogonal decomposition, treating the betas and thetas as fixed constants and the X's as random variables. I can see intuitively why this is the case. Suppose that X1 and Χ2 are highly or even perfectly correlated. If X1 is mean-centered, then it should be uncorrelated with the square of mean-centered Χ2. Roughly speaking, each x1 should be balanced by another of opposite sign, and since they'll both be weighted by the square of χ2 in the sum, they will cancel each other out.

Another way might be to add some caveats. Something like: "The appropriateness of treating genetic effects as random variables in analytical derivations has been questioned. Later, we will justify the theory presented here with simulation results showing that i-LDSC accurately recovers the magnitude of the epistatic variance in Eq. (1) under a broad range of conditions."

A related suggestion is to add a qualifier at some point in the text to the use of the term "additive-by-additive" or the symbol σAA2. The authors do not really explain the relationship between their epistatic variance component associated with product terms and the traditional σAA2, σAD2, and σDD2. Or, instead of a special qualifier, maybe use a different term and symbol altogether. Maybe the term should be "epistatic" or "pairwise-epistatic." Sometimes the symbol σI2 is used to denote epistatic genetic variance, the I standing for "interaction" in the statistical sense (e.g., Falconer and Mackay, 1996). This seems less potentially misleading.

Might interaction LD scores better tag single variants, as in the retracted paper of Hemani et al.

I am satisfied with the response of the authors, based on their new simulations.

The statistical significance and substantive importance of the estimated epistatic variance components

Previously, the authors put forth their heritability estimates with no functional annotations as their main results and treated their estimates with functional annotations as supplementary. I argued that the latter estimates should be treated as the main. In their revision, the authors put both sets of results in their main text. I suppose that this is a reasonable compromise.

The authors argue that the results with functional annotations (i.e., s-LDSC) cannot be dismissed even though they are not statistically significant. They point out in their reply that the standard error calculated by LDSC with the block jackknife is conservative. I had neglected this point, with which I agree. The block-jackknife standard error seems not to vanish with infinite sample size, probably because there are inherent differences between blocks of the genome. The sample size used by the authors (~350,000) might be large enough for this feature of LDSC to be a problem.

Sometimes the LDSC developers in a situation like this will take an average estimate over all traits in an analysis and then use the block jackknife to calculate the significance of it. Perhaps the authors here could calculate the average percentage increase in the heritability; this might turn out to be better powered. But this is just a suggestion, and the authors do not need to take the trouble in order to secure my assent to publication.

References

de los Campos, G., Sorensen, D., and Gianola, D. (2015). Genomic heritability: What is it? PLoS Genetics, 11, e1005048.

Falconer, D.S., and Mackay, T.F.C. (1996). Introduction to quantitative genetics (4th ed.). Longman.

Palmer, D.S. et al. (2023). Analysis of genetic dominance in the UK Biobank. Science, 379, 1341-1349.

Yang, J. et al. (2010). Common SNPs explain a large proportion of the heritability for human height. Nature Genetics, 42, 565-569.

---

## [Author Response]

[Editors’ note: the authors resubmitted a revised version of the paper for consideration. What follows is the authors’ response to the first round of review.]

Reviewer #1 (Recommendations for the authors):In this study, Darnell and colleagues propose a new method to quantify the contribution of cis-epistasis (i.e., interactions between nearby genetic variants) to the heritability of complex traits. Their method, MELD, is an extension of the popular LD score regression methodology. The authors perform simulations conditional on real genotypes to assess the consistency of their estimators of SNP-based heritability and apply their methods to 25 complex traits measured in participants of the UK Biobank and Biobank Japan. The authors conclude that unaccounted epistasis biases estimates of (narrow sense) SNP-based heritability.I have concerns regarding the method and the conclusions of this study.

We thank the reviewer for carefully reading of our manuscript and we appreciate the constructive comments. Our responses and edits to the specific comments are given below.

1. Additive and non-additive effects are not orthogonal in the proposed model. The authors used the generative models in Equations (1) and (2) to define the narrow sense heritability. However, this definition is incorrect because the additive and non-additive components of their model are not orthogonal. A key point to understand is that there is an additive genetic variance even when all genetic effects arise from statistical interactions between loci. This is the core issue discussed in many papers cited by the authors (e.g., Hill, Goddard and Visscher) but this does not seem to be fully grasped in this study. Unfortunately, I think there is an important confusion between what quantitative genetics classically terms as additive genetic variance and what the authors define here. The same distinction applies for dominance variance. Therefore, I don't think that what the authors term as a "bias" is justified here. That being said, it could still be that what MELD estimates (i.e., not the narrow sense heritability) can teach us something interesting about the genetic architecture of complex traits. I'm not just sure what exactly but my next point below may help.2. Observations are consistent with alternative explanations. The linear relationship between (expected) chi-square statistics and LD scores postulated by the LD score regression methodology holds under a number of assumptions: (i) all variants are causal and (ii) explain the same amount of variance. However, if (i) is violated then the expected chi-square statistic at a given SNP k is an affine function of the squared correlation between SNP k and all causal variants (in the vicinity). As consequence, if causal variants are enriched in low or high LD regions of the genome, then extra (non-linear) terms could be needed to represent the relationship between chi-square and LD. I believe that the real data application of MELD is affected by the non-random distribution of causal variants wrt LD. The authors could test this prediction by sampling causal variants as a function of their LD scores, simulate a trait, and apply MELD on their resulting GWAS summary statistics. Another related explanation could be if causal variants are poorly imputed. This should also create a signal detectable by MELD.

We will attempt to address the previous two comments together because they are related. First, we want to thank the reviewer for bringing up these important points and we apologize for not appropriately representing the concept of additive genetic variance as it is classically defined in the quantitative genetics’ literature. If we define additive genetic variance as being any type of effect that can be explained by a linear model, then we agree that the term “bias” is not appropriate.

In the revised manuscript, we follow many of the reviewer’s suggestions as we reframe the contributions of our work. To begin, we now highlight that the original LD score regression (LDSC) relies on the fact that the expected relationship between chi-square test statistics (i.e., the squared magnitude of GWAS allelic effect estimates) and LD scores holds when complex traits are generated under the infinitesimal (or polygenic) model. Importantly, the estimand of the LDSC model is the proportion of phenotypic variance attributable to additive effects of genotyped SNPs. In this paper, we show that the LDSC framework can be extended to estimate greater proportions of genetic variance in complex traits (i.e., beyond the variance that is attributable to additive effects) when a subset of causal variants is involved in a gene-by-gene (G×G) interaction.

The key theoretical insight we now leverage is that SNP-level GWAS summary statistics can provide evidence of non-additive genetic effects contributing to trait architecture if there is a nonzero correlation between individual-level genotypes and their statistical interactions. Our newly named “interaction-LD score” (i-LDSC) model therefore aims to recover missing heritability from GWAS summary statistics by incorporating an additional score that measures the nonadditive genetic variation that is tagged by genotyped SNPs. We then demonstrate how i-LDSC builds upon the original LDSC model through the development of new “*cis*-interaction” LD scores which help to investigate signals stemming from *cis*-acting SNP-by-SNP interactions.

In our new simulations, we show that the *cis*-interaction LD scores that we propose are not always enough to recover missing genetic variation for all types of trait architectures. Following the reviewer’s suggestion, we did a study where we generated synthetic phenotypes with sparse architectures from a spike-and-slab model (Zhou et al. 2013, *PLOS Genetics*). Here, traits were simulated with solely additive effects, but this time only variants with the top or bottom {1, 5, 10, 25, 50, 100 percentile of LD scores were given non-zero coefficients. Breaking the assumed relationship between LD scores and chi-squared test statistics (i.e., that they are generally positively correlated) led to unbounded estimates of heritability in all but the (polygenic) scenario when 100% of SNPs contributed to the phenotypic variance for both LDSC and i-LDSC. Results for these simulations can be found in Figure S15.

3. Estimates of narrow sense heritability from MELD are biased. Figure S8 shows that MELD estimates of narrow sense heritability are biased. The authors propose a model averaging approach, which is claimed to work. However, I remain sceptical that this is the solution to the problem. I believe that the model is trying to fit a very peculiar genetic architecture where there are many causal variants within narrow windows (e.g., 100 kb) all interacting with each other. While there is evidence of widespread allelic heterogeneity (e.g., many causal variants at a given locus), evidence of widespread local epistasis is clearly lacking.

This is a great point, and we appreciate the reviewer for bringing this up. Indeed, different types of interaction scores used within the i-LDSC model will fit to different types of genetic architectures. When the *cis*-interaction scores are computed over genomic window sizes that are too narrow, i-LDSC can yield upwardly biased heritability estimates (e.g., Figure S8). In practice, we find that *cis*-interaction LD scores that are calculated using larger windows lead to the most accurate estimates of heritability while also not over representing the total phenotypic variation explained by tagged non-additive genetic effects. Therefore, unless otherwise stated, we use *cis*interaction LD scores calculated with a ±50 SNP interaction window for all simulations and real data analyses conducted in this work. We now make this explicit point in the revised manuscript (see new lines 225-228 and 549-554).

Other comments.a) Table S6: Estimates from LDSC are inconsistent with previously published data. For example, the LDSC estimates for height and BMI are 0.815 and 0.506, respectively. These estimates are much larger than REML estimates obtained with WGS data (~0.7 and ~0.3). This cannot be true. The MELD estimates for the two traits are 0.57 and 0.282 respectively. While the latter estimates make more sense, I'd be curious to see what is the part attributed by MELD to non-additivity. Sorry if that was reported somewhere but I could not see those.

We really appreciate the reviewer catching this discrepancy in the heritability estimates we report and those commonly reported in the literature. We did an extensive review of the LD score regression framework, the use of certain covariates in the software, and fixed our implementation of the weighted least squares algorithm to estimate model coefficients. This corrected the inflated heritability estimates that we were seeing in our analyses for traits in the UK Biobank and BioBank Japan. For example, for height and body mass index (BMI) in the UK Biobank, we now get heritability estimates around 0.368 and 0.176 for LDSC, respectively. For those same traits, iLDSC produces heritability estimates 0.482 and 0.235, respectively. The updated results for the real data analyses can be found in the new Figures 4 and S16 and the new Tables 1 and S6.

We apologize for not making the estimates σ^ (i.e., the proportion of heritability estimates from iLDSC attributed to non-additivity) more readily available in our initial submission. These now can be found in the newly revised Table 1 along with (1) heritability estimates from LDSC and i-LDSC, and (2) P-values highlighting statistically significant contributions of tagged non-additive genetic variation for 25 traits in the UK Biobank and BioBank Japan.

b) Line 88 : Previous methods have been developed to estimate dominance variance using an LD score regression framework.

We appreciate the reviewer for highlighting this omission in our previous submission. In the newly revised manuscript, we now include a reference to Palmer et al. (2023), *Science* (see line 65) where the LD score regression framework was extended to account for dominance effects.

c) Line 111 (and elsewhere): polygenicity is not a confounding factor like genetic relatedness and population stratification.

We thank the reviewer for pointing out this error and have removed all references to polygenicity as a source of confounding.

d) Lines 137-138: I think this is incorrect. GWAS do not make assumptions (i) and (ii).

We agree with the reviewer that our assumptions about GWA studies was not consistent with the literature. In the revision, we have edited this part of manuscript to read (see new lines 135-141):

“A central objective in GWAS studies is to infer how much phenotypic variation can be explained by genetic effects. To achieve that objective, a key consideration involves incorporating the possibility of non-additive sources of genetic variation to be correlated with and explained by additive effect size estimates obtained from GWAS analyses (Hill et al. 2008, PLOS Genetics). If we assume that the genotype and interaction matrices ***X*** and ***W*** are not completely orthogonal (i.e., such that ***X***^⊺^***W***≠𝟎) then the following relationship between the moment matrix ***X***^⊺^***y***, the observed marginal GWAS summary statistics β^, and the true coefficient values **β** from the generative model in Eq. (1) holds in expectation (see Materials and methods)… “

e) Lines 145 – 148: V * theta is not a bias because marginal effects and joint effects are different. Equation (3) uses the same notation for both. However, even when theta=0, β_hat would never equal β (unless R = Identity matrix, i.e. all SNPs are independent).

We agree with the reviewer, and we apologize again for not using the definitions that are consistent with the quantitative genetics’ literature. We have removed the language we used previously and now write the following (see new lines 145-148):

“Intuitively, the term ***V*θ** can be interpreted as the non-additive effects that are tagged by the additive effect estimates from the GWAS study. Note that, when (i) non-additive genetic effects play a negligible role on the overall architecture of a trait (i.e., such that **θ**=𝟎) or (ii) the genotype and interaction matrices ***X*** and ***W*** do not share the same column space (i.e., such that ***X***^⊺^***W***=𝟎), the equation above simplifies to a relationship between LD and summary statistics that is assumed in many common GWAS studies and methods…”

f) Lines 179 – 184: "Instead, …". This is evidence that the additive and non-additive components are not orthogonal. I'm expecting non-additive effects to explain variance on top of additive genetic variance.

This is a great catch. As we mentioned in a response to a previous comment made by the reviewer, we fixed our implementation of the weighted least squares algorithm to estimate model coefficients. Using the appropriate model fitting strategy resulted in the expected result where heritability estimates for i-LDSC are always larger than those from LDSC --- indicating that the variance explained by the *cis*-interaction LD score is on top of the variance captured by additive LD score alone. The updated results for the real data analyses can be found in the new Figures 4 and S16 and the new Tables 1 and S6.

g) Line 209: "phenotype they were" – I think that "they" should be deleted.

We appreciate the reviewer for reading our manuscript so closely. We have fixed this typo.

Reviewer #2 (Recommendations for the authors):The authors present an approach for variance component estimation using summary statistics. In particular, they extend the univariate LDSC regression framework to include a parameter capturing local pairwise epistasis, in addition to the usual intercept and additive components, a method they call "MELD". The authors achieve this by computing a second set of LD scores that index the extent to which a given SNP tags SNP-SNP products within a local window (as opposed to tagging individual SNPs in vanilla LDSC regression). They perform a variety of simulations showing that their method is well callibrated under the standard additive LDSC model as well as their proposed generative model including local pairwise epistatic effects. Finally, they apply their method to a variety of phenotypes in the UK and Japan biobanks, identifying substantial putative non-additive genetic variance.

We thank the reviewer for their in-depth reading of our manuscript and feel that their suggestions have vastly improved our work. Our responses to the specific comments are given below.

1. In their analyses comparing LDSC and MELD narrow-sense heritability estimates in the UK biobank (UKB), the LDSC heritability estimates are far higher than I've seen elsewhere. For instance, the authors report h2 estimates of 0.815 and 0.506 for height and BMI, respectively. In the Neale lab UKB h2 browser, which used a fairly liberal set of covariates (nealelab.github.io/UKBB_ldsc/h2_browser.html), these same phenotypes have h2 estimates of 0.485 and 0.249. Evans and colleagues report (doi.org/10.1038/s41588-018-0108-x) LDSC h2 estimates of 0.259 and 0.231, also in the UKB. I can only assume an error in the present work has resulted in such large LDSC h2 values. As such, it is hard to meaningfully compare the MELD and LDSC h2 estimates.

We really appreciate the reviewer catching this discrepancy in the heritability estimates we report and those commonly reported in the literature. We did an extensive review of the LD score regression framework, the use of certain covariates in the software, and fixed our implementation of the weighted least squares algorithm to estimate model coefficients. More specifically, we altered the regression weights that were recommended by the original authors of the LDSC model (e.g., HapMap3 SNPs with no MHC region) and we believe that it corrected for correlation in chisquared statistics due to LD and heteroskedasticity between variants with small and large chisquared statistics. This procedure lessened the inflation in the heritability estimates that we were seeing in our initial analyses for traits in the UK Biobank and BioBank Japan. For example, for height and body mass index (BMI) in the UK Biobank, we now get heritability estimates around 0.368 and 0.176 for LDSC, respectively. For those same traits, i-LDSC produces heritability estimates 0.482 and 0.235, respectively. The updated results for the real data analyses can be found in the new Figures 4 and S16 and the new Tables 1 and S6.

We also apologize for not making the estimates σ^ (i.e., the proportion of heritability estimates from i-LDSC attributed to non-additivity) more readily available in our initial submission. These now can be found in the newly revised Table 1 along with (1) heritability estimates from LDSC and iLDSC, and (2) P-values highlighting statistically significant contributions of tagged non-additive genetic variation for 25 traits in the UK Biobank and BioBank Japan.

2. The authors' simulations only cover a subset of common generative models that I'd like to see before interpreting their findings. Specifically, they don't present simulations underadditive + GxE effectsthe above + local pairwise epistatic effectsadditive + long range pairwise epistasis (e.g., at loci on different chromosomes)the above + local pairwise epistatic effectsadditive + ancestry-by-G effects (e.g., the product of individual genotypes and a genomic PC)the above + local pairwise epistatic effectswhich I would like to see both for their proposed method and for LDSC. They do not need to present a method that will perform well under all of these scenarios, only demonstrate how their method performs under other plausible generative models. To the extent that MELD outperforms LDSC under the MELD generative model, it may perform relatively poorly under alternative architectures.

We thank the reviewer for providing us with this suggestion. As suggested by the reviewer, in the revised manuscript, we now compare heritability estimates from both LDSC and i-LDSC in the presence of additive effects, *cis*-acting interactions, and a third source of genetic variance stemming from either gene-by-environment (G×E) or gene-by-ancestry (G×Ancestry) effect. The results for these analyses can be found in the new Figures S9-S14. As we report in new lines 240-254:

“In general, i-LDSC underestimates overall heritability when additive effects and cis-acting interactions are present alongside G×E (Figure S9) and/or G×Ancestry effects when PCs are included as covariates (Figure S10). Notably, when PCs are not included to correct for residual stratification, both LDSC and i-LDSC can yield unbounded heritability estimates greater than 1 (Figure S11). Also interestingly, when we omit cis-interactions from the generative model (i.e., the genetic architecture of simulated traits is only made up of additive and G×E or G×Ancestry effects), i-LDSC will still estimate a nonzero genetic variance component with the cis-interaction LD scores (Figures S12-S14). Collectively, these results empirically show the important point that cis-interaction scores are not enough to recover missing genetic variation for all types of trait architectures; however, they are helpful in recovering phenotypic variation explained by statistical cis-interaction effects. Recall that the linear relationship between (expected) χ^2^ test statistics and LD scores proposed by the LDSC framework holds when complex traits are generated under the polygenic model where all causal variants have the same expected contribution to phenotypic variation. When cis-interactions affect genetic architecture (e.g., in our earlier simulations in Figure 3), these assumptions are violated in LDSC, but the inclusion of the additional nonlinear scores in i-LDSC help recover the relationship between the expectation of χ^2^ test statistics and LD.”

3. It is believed that much of the signal we see in LDSC h2 estimates reflects the effects of unmeasured variants tagged by, e.g., a million or so HapMap3 SNPs. How does MELD perform when one or both of SNPs with epistatic effects aren't directly measured in the provided data?

This is a great suggestion. In the revised version of the manuscript, we performed the new simulations using all the UK Biobank genotype variants to assign causal SNPs contributing additive and nonlinear effects (see new Figures S9-S15). However, some of these are not HapMap3 SNPs and are ultimately not included in the LDSC or i-LDSC regression analyses to produce heritability estimates. This presents a realistic scenario where variants with both additive and non-additive genetic effects are omitted (line 688-698). Exceptions to this setup are the initial set of simulations which we use to illustrate the power of the i-LDSC in a model where the exact effect of all variants are observed and included in the LDSC or i-LDSC regression (these include Figures 1-3 and S1-S8).

4. There is an analogy between LDSC regression and Haseman-Elston (HE) regression (doi.org/10.1214/17-AOAS1052). What would the HE regression equivalent of MELD be? Answering this question will better situate MELD in existing methodological literature.

We appreciate the reviewer for making this connection. Indeed, there is a direct connection between the LDSC framework and variance component approaches such as Haseman-Elston (HE) regression (Zhou 2017, *Annals of Applied Statistics*). While deriving the HE regression equivalent of i-LDSC might be outside the scope of this particular paper, as part of future work, we can think of i-LDSC as a multiple random effect model and explore alternative fitting algorithms such as MQS which is based on a method of moments and produces estimates that are mathematically identical to the HE regression (Zhou 2017, *Annals of Applied Statistics*; Crawford et al. 2017, *PLOS Genetics*; Zhu and Zhou 2020, *Computational and Structural Biotechnology Journal*).

5. It would be helpful to also present the MELD broad-sense h2 estimates whenever MELD and LDSC narrow-sense h2 estimates are compared.

Completely agree. We now report both LDSC and i-LDSC heritability estimates together whenever they are referenced. This can be seen reflected in the new Figures 3-4 and S8-15 and Tables 1 and S6.

[Editors’ note: what follows is the authors’ response to the second round of review.]

The reviewers have discussed their reviews with one another, and the Reviewing Editor has drafted this to help you prepare a revised submission.While the reviewers and editors appreciate that the manuscript has improved substantially since the initial submission – and that the method may be picking up something interesting – a key point that was made in the initial review was not addressed adequately: the authors have not clarified the relationship between the parameters their method estimates and the traditional decomposition of the genetic variance into additive and non-additive components. The traditional definition of the additive variance is an orthogonal decomposition, where the non-additive component is defined as the component that is orthogonal to the best linear prediction of the genetic values. This is not the case in the authors' method since they require that the genotype matrix and genetic interaction matrix are correlated in order for their method to pick up any 'epistasis' component. However, the authors claim to recover additional variance beyond the additive variance even from additive GWAS summary statistics. Unless the authors can clarify in detail how their parameters and empirical results relate to traditionally defined additive and non-additive variance components, we will not be able to publish the manuscript.A further point that was raised during the consultation between the editors and reviewers was whether the authors' method is vulnerable to the artefact that led to a retraction of a Nature paper on cis-epistasis affecting gene expression: https://www.nature.com/articles/s41586-021-03766-y. This article was retracted because they found that what appeared to be epistasis was better explained by interaction genotypes better tagging haplotypes with causal variants.We would require that the authors address the above points in addition to the point raised by reviewer 2 about whether the s-LDSC results should change the interpretation given to the authors' empirical results about the magnitude of epistasis.

We thank the editors for their continued consideration of our manuscript. To address the concern about the lack of relationship between the parameters in i-LDSC and traditional decompositions of genetic variance we have added new derivations around the model formulation. We now fully formalize the interaction component used in the i-LDSC model as an estimate of the phenotypic variance explained by additive-by-additive interactions between genetic variants (see the updated Results and Material and Methods). This provides two key takeaways that also addressed some of the reviewer’s concerns. First, we show that our model does indeed assume that additive, dominance, and additive-by-additive genetic effects are indeed orthogonal to each other. This is important because it means that there is a unique partitioning of genetic variance when studying a trait of interest. The second key takeaway is that the genotype matrix ***X*** and the matrix of genetic interactions ***W*** themselves are correlated despite being linearly independent. This property stems from the fact that the additive-by-additive effects between two SNPs are encoded as the Hadamard product of two genotypic vectors in the form ***w**_m_* = ***x**_j_* ∘ ***x**_k_* (which is a nonlinear function of the genotypes). In the revised manuscript, new edits corresponding to these changes can be found around Eqs. (2)-(4) and Eqs. (6)-(9).

We additionally performed new sets of simulations. In the first, we investigated the possibility of additive-by-additive interaction effects estimated by i-LDSC being inflated by additive effects in a haplotype that are unobserved during the model fitting of i-LDSC (i.e., a scenario highlighted in the paper mentioned by both the editors and the reviewer). Generally, we observed that, across a range of both minor allele frequencies and effect sizes, the omission of causal haplotypes had a negligible effect on the estimated value of the coefficients in i-LDSC (see new Figure 3 —figure supplement 9). We hypothesize this is because the simulations were done for polygenic architectures where all SNPs have at least an additive effect. As a result, not observing a small subset of SNPs does not hinder the ability of i-LDSC to estimate genetic variance because the effect size of each SNP is small. If these simulations were conducted for sparse architectures, we would have likely seen a greater impact on i-LDSC; although, we want to note that we have already shown the LD score regression framework to be uncalibrated for traits with sparse genetic architectures (see Figure 3 —figure supplement 8).

Please find our responses to the reviewer below. We are very grateful for the constructive criticism and do agree that addressing these concerns has made our manuscript even stronger.

Reviewer #2 (Recommendations for the authors):This paper has a lot of strong points, and I commend the authors for the effort and ingenuity expended in tackling the difficult problem of estimating epistatic (non-additive) genetic variance from GWAS summary statistics. The mere possibility of the estimated univariate regression coefficient containing a contribution from epistasis, as represented in the manuscript's Equation~3 and elsewhere, is intriguing in and of itself.Is i-LDSC Estimating Epistasis?Perhaps the issue that has given me the most pause is uncertainty over whether the paper's method is really estimating the non-additive genetic variance, as this has been traditionally defined in quantitative genetics with great consequences for the correlations between relatives and evolutionary theory (Fisher, 1930, 1941; Lynch and Walsh, 1998; Burger, 2000; Ewens, 2004).Let us call the expected phenotypic value of a given multiple-SNP genotype the total genetic value. If we apply least-squares regression to obtain the coefficients of the SNPs in a simple linear model predicting the total genetic values, then the partial regression coefficients are the average effects of gene substitution and the variance in the predicted values resulting from the model is called the additive genetic variance. (This is all theoretical and definitional, not empirical. We do not actually perform this regression.) The variance in the residuals---the differences between the total genetic values and the additive predicted values---is the non-additive genetic variance. Notice that this is an orthogonal decomposition of the variance in total genetic values. Thus, in order for the variance in Wθ to qualify as the non-additive genetic variance, it must be orthogonal to Xβ.At first, I very much doubted whether this is generally true. And I was not reassured by the authors' reply to Reviewer~1 on this point, which did not seem to show any grasp of the issue at all. But to my surprise I discovered in elementary simulations of Equation 1 above that for mean-centered X1 and X2, (X1β1+X2β2) is uncorrelated with X1X2θ for seemingly arbitrary correlation between X1 and X2. A partition of the outcome's variance between these two components is thus an orthogonal decomposition after all. Furthermore, the result seems general for any number of independent variables and their pairwise products. I am also encouraged by the report that standard and interaction LD Scores are ‘lowly correlated’ (line~179), meaning that the standard LDSC slope is scarcely affected by the inclusion of interaction LD Scores in the regression; this behavior is what we should expect from an orthogonal decomposition.I have therefore come to the view that the additional variance component estimated by i-LDSC has a close correspondence with the epistatic (non-additive) genetic variance after all.In order to make this point transparent to all readers, however, I think that the authors should put much more effort into placing their work into the traditional framework of the field. It was certainly not intuitive to multiple reviewers that Xβ is orthogonal to Wθ. There are even contrary suggestions. For if (Xβ)⊺Wθ=β⊺X⊺Wθ is to equal zero, we know that we can't get there by X⊺W equaling zero because then the method has nothing to go on (e.g., line~139). We thus have a quadratic form---each term being the weighted product of an average (additive) effect and an interaction coefficient---needing to cancel out to equal zero. I wonder if the authors can put forth a rigorous argument or compelling intuition for why this should be the case.In the case of two polymorphic sites, quantitative genetics has traditionally partitioned the total genetic variance into the following orthogonal components:additive genetic variance, σA2, the numerator of the narrow-sense heritability;dominance genetic variance, σD2;additive-by-additive genetic variance, σAA2;additive-by-dominance genetic variance, σAD2; anddominance-by-dominance genetic variance, σDD2.See Lynch and Walsh (1998, pp. 88-92) for a thorough numerical example. This decomposition is not arbitrary or trivial, since each component has a distinct coefficient in the correlations between relatives. Is it possible for the authors to relate the variance associated with their Wθ to this traditional decomposition? Besides justifying the work in this paper, the establishment of a relationship can have the possible practical benefit of allowing i-LDSC estimates of non-additive genetic variance to be checked against empirical correlations between relatives. For example, if we know from other methods that σD2 is negligible but that i-LDSC returns a sizable σAA2, we might predict that the parent-offspring correlation should be equal to the sibling correlation; a sizable σD2 would make the sibling correlation higher. Admittedly, however, such an exercise can get rather complicated for the variance contributed by pairs of SNPs that are close together (Lynch and Walsh, 1998, pp. 146-152).I would also like the authors to clarify whether LDSC consistently overestimates the narrow-sense heritability in the case that pairwise epistasis is present. The figures seem to show this. I have conflicting intuitions here. On the one hand, if GWAS summary statistics can be inflated by the tagging of epistasis, then it seems that LDSC should overestimate heritability (or at least this should be an upwardly biasing factor; other factors may lead the net bias to be different). On the other hand, if standard and interaction LD Scores are lowly correlated, then I feel that the inclusion of interaction LD Score in the regression should not strongly affect the coefficient of the standard LD Score. Relatedly, I find it rather curious that i-LDSC seems increasingly biased as the proportion of genetic variance that is non-additive goes up---but perhaps this is not too important, since such a high ratio of narrow-sense to broad-sense heritability is not realistic.

We thank the reviewer for taking the time to thoughtfully offer more context on how we might situate the i-LDSC framework within the greater context of traditional quantitative genetics. We now formalize the interaction component used in the i-LDSC model as an estimate of the phenotypic variance explained by additive-by-additive interactions between genetic variants (which we denote by σ^AA to follow the conventional notation). In the newly revised Material and Methods, we also show how the i-LDSC model can be formulated to include dominance effects in a more general framework. Our updated derivations provide two key takeaways.

First, we assume that the additive and interaction effect sizes in the general model (***β****,****θ***) are each normally distributed with variances proportional to their individual contributions to trait heritability: βj∼N(0,σA2),θm∼N(0,σAA2). This independence assumption implies that the additive and nonadditive components ***Xβ*** and ***Wθ*** are orthogonal where E[βTXTWθ]=E[βT]XTWE[θ]=0. This is important because, as the reviewer points out, it means that there is a unique partitioning of genetic variance when studying a trait of interest. In the revised version of the manuscript, we show this derivation in the main text (see lines 129-143). We also extend this derivation in the Materials and methods where we show the same result even after we include the presence of dominance effects in the generative model (see lines 415-417 and 438-457).

Second, we show that the genotype matrix ***X*** and the matrix of genetic interactions ***W*** are not linearly dependent because the additive-by-additive effects between two SNPs are encoded as the Hadamard product of two genotypic vectors in the form ***w**_m_* = ***x**_j_* ∘ ***x**_k_* (which is a nonlinear function of the genotypes). Linear dependence would have implied that one could find a transformation between a SNP and an interaction term in the form *w_m_* = *c × x_j_* for some constant *c*. However, despite their linear independence, ***X*** and ***W*** are themselves not orthogonal and still have a nonzero correlation. This implies that the inner product between genotypes and their interactions is nonzero XTW≠0. To see this, we focus on a focal SNP ***x***_and_ and consider three different types of interactions:

Scenario I: Interaction between a focal SNP with itself (***x**_j_* ∘ ***x**_j_*).Scenario II: Interaction between a focal SNP with a different SNP (***x**_j_* ∘ ***x**_k_*).Scenario III: Interaction between a focal SNP with a pair of different SNPs (***x**_k_* ∘ ***x**_l_*).

In the Materials and methods of the revised manuscript, we now provide derivations showing when would expect nonzero correlation between ***X*** and ***W*** which rely on the fact that: (1) we assume that genotypes have been mean-centered and scaled to have unit variance, and (2) under Hardy-Weinberg equilibrium, SNPs marginally follow a binomial distribution ***x***_and_ ~*Bin*(2,*p*) where *p* represents the minor allele frequency (MAF) (Wray et al. 2007, *Genome Res*; Lippert et al. 2013, *Sci Rep*). These new additions are given in new lines 460-485.

Lastly, we agree with the reviewer that our results indicate that LDSC inflates estimates of SNPbased narrow-sense heritability. Our intuition for why this happens is largely consistent with the reviewer’s first point: since GWAS summary statistics can be inflated by the tagging of nonadditive genetic variance, then it makes sense that LDSC should overestimate heritability. LDSC uses a univariate regression without the inclusion of *cis-*interaction scores. A simple consequence from “omitted variable bias” is likely happening where, since LDSC does not explicitly account for contributions from the tagged non-additive components which also contribute to the variance in the GWAS summary statistics, the estimate for the coefficient σA2 becomes slightly inflated.

How Much Epistasis Is i-LDSC Detecting?I think the proper conclusion to be drawn from the authors' analyses is that statistically significant epistatic (non-additive) genetic variance was not detected. Specifically, I think that the analysis presented in Supplementary Table S6 should be treated as a main analysis rather than a supplementary one, and the results here show no statistically significant epistasis. Let me explain.Most serious researchers, I think, treat LDSC as an unreliable estimator of narrow-sense heritability; it typically returns estimates that are too low. Not even the original LDSC paper pressed strongly to use the method for estimating h^2^ (Bulik-Sullivan et al., 2015). As a practical matter, when researchers are focused on estimating absolute heritability with high accuracy, they usually turn to GCTA/GREML (Evans et al., 2018; Wainschtein et al., 2022).One reason for low estimates with LDSC is that if SNPs with higher LD Scores are less likely to be causal or to have large effect sizes, then the slope of univariate LDSC will not rise as much as it ‘should’ with increasing LD Score. This was a scenario actually simulated by the authors and displayed in their Supplementary Figure S15. [Incidentally, the authors might have acknowledged earlier work in this vein. A simulation inducing a negative correlation between LD Scores and χ^2^ statistics was presented by Bulik-Sullivan et al. (2015, Supplementary Figure 7), and the potentially biasing effect of a correlation over SNPs between LD Scores and contributed genetic variance was a major theme of Lee et al. (2018).] A negative correlation between LD Score and contributed variance does seem to hold for a number of reasons, including the fact that regions of the genome with higher recombination rates tend to be more functional. In short, the authors did very well to carry out this simulation and to show in their Supplementary Figure S15 that this flaw of LDSC in estimating narrow-sense heritability is also a flaw of i-LDSC in estimating broad-sense heritability. But they should have carried the investigation at least one step further, as I will explain below.Another reason for LDSC being a downwardly biased estimator of heritability is that it is often applied to meta-analyses of different cohorts, where heterogeneity (and possibly major but undetected errors by individual cohorts) lead to attenuation of the overall heritability (de Vlaming et al., 2017).The optimal case for using LDSC to estimate heritability, then, is incorporating the LD-related annotation introduced by Gazal et al. (2017) into a stratified-LDSC (s-LDSC) analysis of a single large cohort. This is analogous to the calculation of multiple GRMs defined by MAF and LD in the GCTA/GREML papers cited above. When this was done by Gazal et al. (2017, Supplementary Table 8b), the joint impact of the improvements was to increase the estimated narrow-sense heritability of height from 0.216 to 0.534.All of this has at least a few ramifications for i-LDSC. First, the authors do not consider whether a relationship between their interaction LD Scores and interaction effect sizes might bias their estimates. (This would be on top of any biasing relationship between standard LD Scores and linear effect sizes, as displayed in Supplementary Figure~S15.) I find some kind of statistical relationship over the whole genome, induced perhaps by evolutionary forces, between cis-acting epistasis and interaction LD Scores to be plausible, albeit without intuition regarding the sign of any resulting bias. The authors should investigate this issue or at least mention it as a matter for future study. Second, it might be that the authors are comparing the estimates of broad-sense heritability in Table~1 to the wrong estimates of narrow-sense heritability. Although the estimates did come from single large cohorts, they seem to have been obtained with simple univariate LDSC rather than s-LDSC. When the estimate of h2 obtained with LDSC is too low, some will suspect that the additional variance detected by i-LDSC is simply additive genetic variance missed by the downward bias of LDSC. Consider that the authors' own Supplementary Table~S6 gives s-LDSC heritability estimates that are consistently higher than the LDSC estimates in Table~1. E.g., the estimated h2 of height goes from 0.37 to 0.43. The latter figure cuts quite a bit into the estimated broad-sense heritability of 0.48 obtained with i-LDSC.Here we come to a critical point. Lines 282--286 are not entirely clear, but I interpret them to mean that the manuscript's Equation~5 was expanded by stratifying ℓ into the components of s-LDSC and this was how the estimates in Supplementary Table~S6 were obtained. If that interpretation is correct, then the scenario of i-LDSC picking up missed additive genetic variance seems rather plausible. At the very least, the increases in broad-sense heritability reported in Supplementary Table~S6 are smaller in magnitude and not statistically significant. Perhaps what this means is that the headline should be a negligible contribution of pairwise epistasis revealed by this novel and ingenious method, analogous to what has been discovered with respect to dominance (Hivert et al., 2021; Pazokitoroudi et al., 2021; Okbay et al., 2022; Palmer et al., 2023).

This is an excellent question raised by the reviewer and, again, we really appreciate such a thoughtful and thorough response. First, we completely agree with the reviewer that the s-LDSC estimates previously included in the Supplementary Material should instead be discussed in the main text of the manuscript. In the revision, we have now moved the old Supplemental Table S6 to be the new Table 2. Second, we also agree that the conclusions about the magnitude of additive-by-additive effects should be based upon variance explained when using the cisinteraction score in addition to scores specific to different biological annotations when available, per s-LDSC.

However, we want to respectfully disagree that the results indicate a negligible contribution of additive-by-additive genetic variance to all the traits we analyzed (see Figure 4D). Although the additive-by-additive genetic variance component is not significant in any trait in the UK Biobank, there is little reason to expect that they would be given the inclusion of 97 other biological annotations from the s-LDSC model. Indeed, in the s-LDSC paper itself the authors look only for enrichment of heritability for a given annotation not a statistically significant test statistic. It also worth noting that jackknife approaches tend to be conservative and yield slightly larger standard errors for hypothesis testing. Taking all the great points that the reviewer mentioned into account, we believe that a moderate stance to the interpretation of our results is one that: (i) emphasizes the importance of using s-LDSC with the *cis*-interaction score to better assess the variance explained by additive-by-additive interaction effects and (ii) allows for the significance of the additive-by-additive component to not be the only factor when determining the importance of the role of non-additive effects in shaping trait architecture.

In the revision, we now write the following in lines 331-343:

“Lastly, we performed an additional analysis in the UK Biobank where the cis-interaction scores are included as an annotation alongside 97 other functional categories in the stratified-LD score regression framework and its software s-LDSC (Materials and methods). Here, s-LDSC heritability estimates still showed an increase with the interaction scores versus when the publicly available functional categories were analyzed alone, but albeit at a much smaller magnitude (Table 2). The contributions from the additive-by-additive component to the overall estimate of genetic variance ranged from 0.005 for MCHC (P = 0.373) to 0.055 for HDL (P = 0.575) (Figures 4C and 4D). Furthermore, in this analysis, the estimates of the additive-by-additive components were no longer statistically significant for any of the traits in the UK Biobank (Table 2). Despite this, these results highlight the ability of the i-LDSC framework to identify sources of “missing” phenotypic variance explained in heritability estimation. Importantly, moving forward, we suggest using the cisinteraction scores with additional annotations whenever they are available as it provides more conservative estimates of the role of additive-by-additive effects on trait architecture.”

Lastly, in the Discussion, we now mention an area of future work would be to explore how the relationship between *cis*-interaction LD scores and interaction effect sizes might bias heritability estimates from i-LDSC (e.g., similar to the relationship explored standard LD scores and linear effect sizes in Figure 3 —figure supplement 8). See new lines 364-367.

Abstract: Here and elsewhere, the term cis is used. Parenthetically it is explained that a cis-interaction score captures interactions between a focal variant and nearby variants. Does this mean that i-LDSC only tries to estimate the variance contributed by pairwise interactions between nearby SNPs? That is, does the method say nothing about statistical interactions between SNPs on different chromosomes, say? The paper should make a clear statement about this point somewhere.

This is a great suggestion. We believe that *cis* in the genetics context is commonly used to refer to physically colocated loci on the same chromosome but have updated the description of the *cis*interaction in the abstract (lines 29-30) to make this explicit.

line~54--58: I do not understand what is being asserted here. I think it might clear some things up to point out that the developers of LDSC did not rest content with the assumption that every SNP has an effect. E.g., they conducted simulations showing that the intercept and slope remain unbiased estimators, so along as certain other conditions are met, as the proportion of SNPs with a nonzero effect ranged from one all the way down to 10^-4^ (Bulik-Sullivan et al., 2015, Supplementary Figures 3 and 4).

We apologize for not being clearer here in the previous submission. We now highlight the initial set of simulations performed by Bulik-Sullivan et al. (2015) in testing the tolerance of the LDSC framework to violations of assumptions of the infinitesimal model in new lines 58-60.

Line 177: I suggest choosing a term other than ‘infinitesimal model’. Some kind of model where every SNP has an effect does not need to hold in order for standard LDSC to work.

We agree and have added that if variance in a trait is only determined by additive effect that the model may also be polygenic (see new lines 185-186).

line~505: `delete-one jackknife' is confusing, because each block removed in a jackknife replicate contains more than one SNP. Leaving out the `delete-one' is better.

We agree that this was confusing and have removed ‘delete-one’ from the manuscript.

lines~513--518: The justifications for the shortcuts in the computation of interaction LD Scores seem a bit handwavy to me. What about SNPs with which the *j*-th SNP is highly correlated? Is it possible to explain why these do not matter so much?

This is a great catch. The previous description of this step was not correct. In the revised version of the manuscript, this procedure is not described as the following (see new lines 589-593):

“In practice, cis-interaction LD scores in i-LDSC can be computed efficiently through realizing two key opportunities for optimization. First, given *J* SNPs, the full matrix of genome-wide interaction effects ***W*** contains on the order of *J*(*J* – 1) /2 total pairwise interactions. However, to compute the cis-interaction score for each SNP, we simply can replace the full ***W*** matrix with a subsetted matrix ***W***_and_ which includes only interactions involving the *j*-th SNP.”

line~615: Now here is a variance decomposition which, I think, is truly questionable. Why does it make sense to say that the variance attributable to statistical interaction between genetic and non-genetic variables is part of the broad-sense heritability?

We define the broad-sense heritability here to be the phenotypic variance explained by any genetic effects, even if there is a genetic effect with an environmental variable. The initial set of revisions suggested a model where a proportion of the phenotypic variation was explained by G×Ancestry or G×E effects. That previous reviewer argued that, although these are not explicitly included in foundational models of variance decomposition, they do offer a useful insight into how i-LDSC estimates might be affected by the presence of genetic interactions with environmental variables.

Figure~S8: What makes windows of 25 or 50 SNPs for calculating interaction LD Scores the best for getting an estimate of the broad-sense heritability with i-LDSC? Why does a smaller window lead to inflation? This is not fully explained. I am sometimes fine with trying several values of an adjustable setting and picking whatever seems to work the best, but here I am more than a bit curious.

We selected the 50 SNP windows for our analysis of empirical traits in the UK Biobank and Biobank Japan because it mitigated any upward bias in the estimates of total heritability explained. We believe that smaller windows provide noisier estimates because they consider only a few interactions and are thus prone to over-estimate the impact of additive-by-additive effects between a focal variant and the rest of the genome.

Figure~S16: Is this a duplication of Figure~4D?

Great catch. Indeed, these two figures were duplicated. Since we have updated Figure 4D to illustrate a different set of parameter estimates with s-LDSC, this comparison of the intercepts in BioBank Japan and the UK Biobank is now only included as the new Figure 4 —figure supplement 1B.

[Editors’ note: what follows is the authors’ response to the third round of review.]

The manuscript has been improved but there are some remaining issues that need to be addressed, as outlined below. If you think you are unable to fully address the reviewer's comments (which pertain to similar issues identified in the last round of reviews), we would advise you not to submit a revised manuscript.Reviewer #1 (Recommendations for the authors):1. I don't have issues with the authors' derivations but I do still think the authors have not yet successfully related their variance decomposition to the standard variance decomposition in classical quantitative genetics. Here's a simple example demonstrating thisFor simplicity, let the generative model be y=X1β1+X2β2+Wθ+e where X1 and X2 are independent and standardized and W=standardize(X1X2) denoting r1=cov(X1,W)≠0, r2=cov(X2,W)≠0, with cov(X1,X2)=0 and e independent of everything else and y Y having zero mean and unit variance.The classical additive genetic variance will the R-squared from regressing y on X1 and X2:Vg=maxbcor(y,X1+X2b)2=maxb[cov(y,X1+X2b)21+b2]=maxb[cov(X1β1+X2β2+Wθ+e,X1+X2b)21+b2]=maxb[(β1+θr1+β2b+θr2b)21+b2]Doing some calculus, we get Vg=(β12+β22)+2θ(β1r1+β2r2)+θ2(r12+r22). So the classical genetic variance is larger than the additive variance component in the authors model (here β12+β22)! It will include the part of the epistatic component tagged by additive SNPs X, which is nonzero despite Xβ and Wθ being orthogonal.I imagine you should be able to figure out the classical additive genetic variance (and epistatic etc) from the authors' model, though this isn't present in their current parametrization. If you can do that, I think it's far easier to interpret.

We thank the reviewer for continuing to encourage us to connect our random effect (linear mixed model) parameterization with the classical form of additive genetic variance. As the reviewer points out, the concept of additive genetic effects partially explaining non-additive variation has also described in classical quantitative genetics (Hill et al. 2008, *PLOS Genetics*, Hivert et al. 2021, *Am J Hum Genet*; Mäki-Tanila and Hill 2014, *Genetics*). In the revised version of the manuscript, we now include the following connection between our model and classical quantitative genetics (see subsection “Connection to quantitative genetics theory”, subsection “Full derivation of interaction LD score regression” and subsection “Overview of the interaction-LD score regression model”).

We hope that the combination of these new derivations better relate our variance decomposition to the standard variance decomposition in classical quantitative genetics.

2. I'm not satisfied with the authors' response to the editors' comment here:"If these simulations were conducted for sparse architectures, we would have likely seen a greater impact on i-LDSC; although, we want to note that we have already shown the LD score regression framework to be uncalibrated for traits with sparse genetic architectures (see Figure 3 —figure supplement 8)."The issue here isn't that the results will be biased; LDSC regression is biased for estimating h2 with missing variants but this bias is easy to understand--missing causal variants lead to lower estimates. The question for your method is what happens to epistatic variance component. It's fine if it's attenuated, it's not fine if leads to false positive signal of epistasis. Simulations to clarify this are necessary

This is a good point brought up by the reviewer. We apologize for not considering this in the previous submission. In the newly revised manuscript, we now include an additional simulation scenario where we generate synthetic traits but run i-LDSC where some number of causal SNPs involved in pairwise interactions are not observed (see new lines 282, 292-296, and 861-879, and the new Figure 3 -—figure supplement 10 and 11). In the text, Materials and methods, we describe this simulation scenario as the following:

“In this set of simulations, we extend the polygenic case to a setting where a portion of the variants involved in genetic interactions are unobserved. Similar to the case with unobserved additive effects, the purpose of these simulations is to assess whether the i-LDSC framework is prone to false discovery of non-additive genetic variance when causal interacting SNPs are not included during the estimation of GWAS summary statistics. In each simulation, we generated haplotypes that each contain 5,000 variants. Traits were simulated using the generative model in Eq. (35) with both additive and interaction effects such that V[Xβ]+V[Wθ]=H2. Here, every SNP in the genome had at least a small additive effect with a corresponding effect size that was drawn to be inversely proportional to its MAF. Only 1% or 5% of variants within each haplotype had causal non-zero interaction effects. However, when running i-LDSC, only a percentage of the interacting SNPs {1%, 5%, 10%, 25%, or 50%} were included in the estimation of ϑ^. We once again generate traits with heritability H2={0.3,0.6} such that the proportion of genetic variance explained by additive effects was equal to ρ={0.5,0.8}. As with the other simulation scenarios, all synthetic traits were generated using UK Biobank genotyped variants that passed initial preprocessing and quality control (see next section). Since not all of these SNPs are HapMap3 SNPs, some variants were omitted from the i-LDSC regression analyses. Overall, as discussed in the main text with results taken over 100 replicates, i-LDSC underestimated values of ϑ^ when there were unobserved interacting variants (see Figure 3 -—figure supplement 10 and 11). As expected, estimates of the additive variance component τ^, on the other hand, were not affected.”

Reviewer #2 (Recommendations for the authors):I focus my review in this round on the issues raised by the editor and also on a point of my own accepting the results at face value, how are we to interpret their statistical significance and importance?Orthogonal decomposition of the genetic varianceI do not think the authors did much to clarify whether their partition of the genetic variance is an orthogonal decomposition in accord with traditional quantitative-genetic theory. Looking at their line 138 and Equation 8, I see that they make use of the assumption that the additive effects and interaction coefficients are independent random variables with zero means. This kind of assumption has become common in statistical genetics (e.g., Yang et al., 2010), and I'm pretty sure the authors think they are simply following the lead of the original LDSC paper. But, as a few papers have pointed out (e.g., de los Campos et al., 2015), this assumption does not make a lot of sense since it is properties of individuals (genotypes, residuals) that are random rather than the properties of SNPs. And even if we are taking an average over different locations in the genome rather than over a population of individuals, why should the mean of the betas be zero? If we always count the derived allele, then the mean might well be negative if there is any tendency for mutations to have a depressing effect on the trait in question. Also, why independent? The recent report that SNPs with larger main effects tend to exhibit more dominance (Palmer et al., 2023) might lead us to expect that pairs of SNPs with larger main effects might show stronger statistical interactions.Despite the fact that the ‘symbolic convention’ of E[β]=0 has often been used to justify surprisingly robust tools, I think that it should no longer be retained in statistical genetics. It is confusing, it might be used in the future in an attempt to justify results that are really far off, and it is known already to prove too much. The supposed proofs of GCTA and LDSC seem to show that they are definitely unbiased, but they are in fact biased under conditions that cannot be explained in terms of this genetic-effects-as-random-variables notion.My strong suggestion is to justify the orthogonality of Xβ and Wθ in some different way. One such way might to prove that Equation 1 in my first review is an orthogonal decomposition, treating the betas and thetas as fixed constants and the X's as random variables. I can see intuitively why this is the case. Suppose that X1 and X2 are highly or even perfectly correlated. If X1 is mean-centered, then it should be uncorrelated with the square of mean-centered X2. Roughly speaking, each x1 should be balanced by another of opposite sign, and since they'll both be weighted by the square of X2 in the sum, they will cancel each other out.Another way might be to add some caveats. Something like: "The appropriateness of treating genetic effects as random variables in analytical derivations has been questioned. Later, we will justify the theory presented here with simulation results showing that i-LDSC accurately recovers the magnitude of the epistatic variance in Eq. (1) under a broad range of conditions."

We appreciate the reviewer for bringing up these important points. Indeed, when formulating the i-LDSC model, we wanted to follow the lead of the original LDSC paper to draw more explicit parallels with our approach. Taking this into account and following the suggestion of the reviewer, we now include caveats in the text when assuming that the additive effects and interaction coefficients are independent random variables with zero means in the generative model. In the revision, we now write the following in lines 131-134:

While the appropriateness of treating genetic effects as random variables in analytical derivations has been questioned (de los Campos et al. 2015, PLOS Genet), later, we will justify the theory presented here with simulation results showing that i-LDSC accurately recovers non-additive genetic variance in Eq. (1) under a broad range of conditions.

We also include a similar caveat in the Materials and methods (see new lines 456-459):

“As we mentioned in the main text, we recognize that the appropriateness of treating genetic effects as random variables in analytical derivations has been questioned (de los Campos et al. 2015, PLOS Genet), but our simulation studies show that i-LDSC accurately recovers nonadditive genetic variance in Eq. (7) under a broad range of conditions.”

In reference to the reviewer’s first point about needing to clarify how our partition of variance relates to theory in traditional quantitative genetics theory, we also want to highlight our response to Reviewer #1 above where we now derive a formal connection with our model (also see subsection “Connection to quantitative genetics theory”, subsection “Full derivation of interaction LD score regression” and subsection “Overview of the interaction-LD score regression model” in the newly revised manuscript).

A related suggestion is to add a qualifier at some point in the text to the use of the term "additive-by-additive" or the symbol σAA2. The authors do not really explain the relationship between their epistatic variance component associated with product terms and the traditional σAA2, σAD2, and σDD2. Or, instead of a special qualifier, maybe use a different term and symbol altogether. Maybe the term should be "epistatic" or "pairwise-epistatic." Sometimes the symbol σI2 is used to denote epistatic genetic variance, the I standing for "interaction" in the statistical sense (e.g., Falconer and Mackay, 1996). This seems less potentially misleading.

This is a great suggestion. In the newly revised manuscript, we now reserve the “additive-by-additive” and the notation σA2 and σAA2 for when we make connections between our random effect model and theoretical concepts in classical quantitative genetics (again see see subsection “Connection to quantitative genetics theory”, subsection “Full derivation of interaction LD score regression” and subsection “Overview of the interaction-LD score regression model”). Following the reviewer’s points, we also use a different variable for the regression coefficient in the i-LDSC model changing the old σ^AA2 to a new symbol ϑ^. The i-LDSC model is now formulated as the following (e.g., see new Eqs. (5), (27), and (28)):E[X2]≈lτ+fϑ+1

We describe the coefficient ϑ^ as an estimate of the proportion of phenotypic variation explained by tagged “pairwise interaction” effects. We hope that the combination of these changes is less misleading for readers.

Might interaction LD scores better tag single variants, as in the retracted paper of Hemani et al.I am satisfied with the response of the authors, based on their new simulations.

We really appreciate the suggestions for the additional simulation studies during the previous round of review.

The statistical significance and substantive importance of the estimated epistatic variance componentsPreviously, the authors put forth their heritability estimates with no functional annotations as their main results and treated their estimates with functional annotations as supplementary. I argued that the latter estimates should be treated as the main. In their revision, the authors put both sets of results in their main text. I suppose that this is a reasonable compromise.The authors argue that the results with functional annotations (i.e., s-LDSC) cannot be dismissed even though they are not statistically significant. They point out in their reply that the standard error calculated by LDSC with the block jackknife is conservative. I had neglected this point, with which I agree. The block-jackknife standard error seems not to vanish with infinite sample size, probably because there are inherent differences between blocks of the genome. The sample size used by the authors (~350,000) might be large enough for this feature of LDSC to be a problem.Sometimes the LDSC developers in a situation like this will take an average estimate over all traits in an analysis and then use the block jackknife to calculate the significance of it. Perhaps the authors here could calculate the average percentage increase in the heritability; this might turn out to be better powered. But this is just a suggestion, and the authors do not need to take the trouble in order to secure my assent to publication.

We thank the reviewer for pointing this out. We were unaware that sometimes the LDSC developers will take an average estimate over all traits in an analysis and then use the block jackknife to calculate statistical significance. While we elected not to do that here, we will keep this strategy in mind for future work. We also added the following statement referencing the possibility of this approach in new lines 639-643:

“It is worth noting that the block-jackknife approach tends to be conservative and yield larger standard errors for hypothesis testing (Efron 1982). As an alternative, we could first run i-LDSC using the block-jackknife procedure over all traits in a study and then use the average of the standard errors to calculate the statistical significance of coefficient estimates; but we do not explore this strategy here and leave that for future work.”